# Modelling the water isotopes distribution in the Mediterranean Sea using a high-resolution oceanic model (NEMO-MED12-watiso-v1.0): Evaluation of model results against in-situ observations

Mohamed Ayache[1], Jean-Claude Dutay[1], Anne Mouchet[2], Kazuyo Tachikawa[3], Camille Risi[4], and Gilles Ramstein[1]

[1]Laboratoire des Sciences du Climat et de l'Environnement, CEA-CNRS-Université Paris Saclay, 91191, Gif-sur-Yvette, France
[2]Freshwater and OCeanic science Unit of reSearch (FOCUS), Université de Liège, B-4000 Liège
[3]Aix Marseille Univ, CNRS, IRD, INRAE, Coll France, CEREGE, 13545, Aix-en-Provence, France
[4]Laboratoire de Météorologie Dynamique, IPSL, CNRS, Sorbonne Université, Paris, France

**Correspondence:** Mohamed Ayache (mohamed.ayache@lsce.ipsl.fr)

**Abstract.**

Stable water isotopes ($\delta^{18}O_w$ and $\delta D_w$) have been successfully implemented for the first time in a high-resolution model of the Mediterranean Sea (NEMO-MED12). In this numerical study, model results are compared with available in-situ observations to evaluate the model performance of the present-day distribution of stable water isotopes and their relationship with salinity on a sub-basin scale. There is good agreement between the modelled and observed distributions of $\delta^{18}O_w$ in the surface water. The model successfully simulates the observed east-west gradient of $\delta^{18}O_w$ characterising surface, intermediate and deep waters. The results also show good agreement between the simulated $\delta D_w$ and the in-situ data. The $\delta D_w$ shows a strong linear relationship with $\delta^{18}O_w$ ($r^2$ = 0.98) and salinity ($r^2$ = 0.94) for the whole Mediterranean Sea. Moreover, the modelled relationships between $\delta^{18}O_w$ and salinity agree well with observations, with a weaker slope in the eastern basin than in the western basin. We investigate the relationship of the isotopic signature of the planktonic foraminifera shells ($\delta^{18}O_c$) with temperature and the influence of seasonality. Our results suggest a more quantitative use of $\delta^{18}O$ records, combining reconstruction with modelling approaches.

## 1 Introduction

Because of their conservative behaviour, stable water isotopes ($\delta^{18}O$ [1] and $\delta D_w$ [2]) provide a unique opportunity to assess hydrological processes and study the hydrological cycle in climate system variability. The isotopic composition of seawater ($\delta^{18}O$) is globally linked to salinity because $\delta^{18}O$ and salinity are affected by common physical processes (i.e. freshwater fluxes or precipitation-evaporation balance). However, the variation of $\delta^{18}O$ is more complex because the water isotopes are subjected to additional fractionation and transport in the atmosphere (Craig and Gordon, 1965). The driving factors mainly

---

[1]equation 1
[2]equation 2

include surface fractionation in relation to atmospheric exchange and oceanic mixing processes, but also continental runoff in coastal areas and ice processes (sea ice formation and iceberg runoff) in polar regions. The evaporation process preferentially extracts lighter water molecules, and the remaining evaporated seawater becomes rich in heavier isotopes. In contrast, the input of freshwater-rich in lighter isotopes by precipitation or river runoff leads to a decrease in the $\delta^{18}O$ and $\delta D_w$ values of seawater. Thus, the salinity and the isotopic compositions of oceanic waters are acquired at the surface; the sinking of surface waters to intermediate or deeper layers does not change these parameters, which can remain stable over long distances until they mix with waters with different properties.

Although water isotopes are among the most widely used proxies in climate research, there are still gaps in our understanding of the processes that control their marine distribution. General circulation models (GCMs) allow us to better understand the past variability of water isotopes documented in various archives and to investigate the relationship between water isotopes and different climate variables. The heavy stable isotopes of water (i.e. deuterium and oxygen-18) have been incorporated into both atmospheric models (e.g., Joussaume et al., 1984; Jouzel et al., 1987; Hoffmann et al., 1998; Brown et al., 2006; Risi et al., 2010a, b; Werner et al., 2011) and oceanic models (Schmidt, 1998, 1999; Paul et al., 1999; Delaygue et al., 2000, 2001; Wadley et al., 2002; Xu et al., 2012), and in coupled ocean-atmosphere models (Schmidt et al., 2007; Tindall et al., 2010; Roche et al., 2004; Roche, 2013; Werner et al., 2016; Cauquoin et al., 2019; Shi et al., 2023). In recent decades, $\delta^{18}O_w$ and $\delta D_w$ data have become increasingly important in paleoclimate modelling studies and have been incorporated into global climate models. The isotopic signals are explicitly simulated to compare with observations, to quantify processes affecting reconstructed seawater isotopic compositions (Roche, 2013; Schmidt et al., 2007). Previous reviews of water isotope measurements and modelling studies (Galewsky et al., 2016; Jones and Dee, 2018; Bowen et al., 2019) have highlighted the importance of understanding spatial and temporal isotopic variability for a quantitative interpretation of its relationship with climate change, and have also shown the potential of $\delta^{18}O$ to characterize individual water masses. However, water isotopes have not been incorporated in a high-resolution regional ocean model, yet. Here, we present the first results of a high-resolution regional dynamical model (at 1/12° horizontal resolution) developed for the Mediterranean Sea (Beuvier et al., 2012a).

In the Mediterranean region, net freshwater fluxes at the sea surface, *i.e.*, the difference between evaporation and precipitation, are the main driving factor of the hydrological cycle (Mariotti et al., 2002), and there is no effect of sea ice formation or melting (*i. e.* no freshwater inflow from ice sheets during the recent "present situation" period). This condition provides a unique opportunity to better understand the spatial and temporal variations of water isotopes in a semi-enclosed basin, away from the interference of sea ice which is currently poorly represented in models. The negative balance between net freshwater input and evaporation (P + R - E < 0) leads to an anti-estuarine pattern in the Mediterranean thermohaline circulation system, with a surface inflow of less saline Atlantic Water (AW) through the Strait of Gibraltar, which is then gradually transformed into saltier water, eventually sinking in the Levantine sub-basin to form Levantine Intermediate Water (LIW) that spreads across the eastern Mediterranean at water depths of between 150 and 700 m until it reaches the Strait of Gibraltar to form the Mediterranean Outflow Water (MOW) (Millot and Taupier-Letage, 2005; Lascaratos et al., 1999). The LIW is one of the main water masses in the Mediterranean Sea (Pinardi and Masetti, 2000), contributing to the formation of the Eastern Mediterranean Deep Water (EMDW) in the Adriatic sub-basin and the Western Mediterranean Deep Water (WMDW) in the Gulf of Lion.

The Mediterranean Outflow Water (MOW) plays an important role in the North Atlantic overturning circulation because the excess salt transported by the water mass contributes to increasing the density of the water masses in the convection zones of the deep water formation (Bigg et al., 2003). In the past, the Mediterranean thermohaline circulation has been profoundly altered, notably during sapropel events, when deep-water ventilation was strongly reduced in the eastern basin, which are well documented by water isotopes observations (Rohling et al., 2015 and references therein). Major changes are also possible in the future as a result of global warming (e.g., Somot et al., 2006; Adloff et al., 2015; Pagès et al., 2020). Understanding the processes that control the circulation of the Mediterranean Sea is therefore a major challenge for understanding climate variability in the Mediterranean basin (e.g., Soto-Navarro et al., 2020).

Compared to other large ocean basins, the Mediterranean Sea can be considered ideal to improve our understanding of the processes that influence and drive oxygen isotope variations, and to further develop the existing modelling approach, because (i) the water residence time is relatively short ($\sim$ 100 years; Millot and Taupier-Letage (2005)); (ii) all major forcing mechanisms are present, including air-sea interaction, buoyancy fluxes and wind forcing, with a well-studied salinity and water isotope structure (e.g., Pierre, 1999); (iii) a well marked $\delta^{18}O_w$ of the surface waters of the eastern Mediterranean basin (value up to 2.2 ‰, Gat et al., 1996) has the potential to trace the process of deep water formation and the thermohaline circulation variability; (iv) a high spatial resolution regional model (NEMO-MED12) is available, which is essential for the simulation of realistic ocean dynamics, and which can then be used for past climate simulation with the adapted coupled regional model (Vadsaria et al., 2020). Over the last decades, considerable progress has been achieved in our understanding of the processes and mechanisms governing the distribution of water isotopes in the Mediterranean Sea, through high-quality sampling and measurements (e.g., Gat et al., 1996; Pierre, 1999; LeGrande and Schmidt, 2006). Nonetheless, no specific modelling focused on water isotopes is yet available for the Mediterranean Sea. This study aims to implement water isotopes as passive tracers in the high-resolution dynamical model (NEMO-MED12) to prepare a direct evaluation of paleoclimate simulation that will then be performed using this modelling platform. We use isotope fluxes from the atmospheric general circulation model (LMDZ-iso, Risi et al., 2010b). Our paper focuses on the simulation of the present-day oceanic distribution of $\delta^{18}O_w$ and $\delta D_w$. We compare model results with existing observations to assess the model's ability to capture the main features of water isotopes distribution in the Mediterranean Sea, as well as the relationship between salinity as a function of $\delta^{18}O_w$ and $\delta D_w$. By combining $\delta^{18}O_w$ and temperature, we can calculate equilibrated calcite $\delta^{18}O_c$ values using paleotemperature equations to compare model results with recent biogenic carbonate data. The results are analysed for the eastern (EMed) and western (WMed) basins to investigate the processes leading to the isotopic distribution of $\delta^{18}O$ and $\delta D_w$ in the Mediterranean Sea. The knowledge of the present-day variability of the isotopic composition of Mediterranean waters should help further studies dedicated to Mediterranean paleoceanography.

## 2 Method

### 2.1 Circulation and ocean dynamic using the NEMO-MED12 model

The dynamical model is the NEMO (Nucleus for European Modelling of the Ocean) free surface ocean circulation model (Madec and NEMO-Team., 2008) in a regional high-resolution configuration called NEMO-MED12 (Beuvier et al., 2012b). The NEMO-MED12 grid is an extraction from the global ORCA-1/12° grid. This corresponds to a grid cell size between 6 to 7.5km from 46°N to 30°N and represents a grid size of 567 × 264 points. The NEMO-MED12 domain covers the entire Mediterranean Sea and includes the west of Gibraltar in the Atlantic Ocean (buffer zone) from 30–47º N in latitude and from 11ºW–36º E in longitude, where salinity and temperature (3-D fields) are relaxed to the observed climatology (Beuvier et al., 2012a). Water exchange with the Black Sea is represented as a two-layer flow with net budget estimates from Stanev and Peneva (2002). The dynamical simulation (the circulation fields, i.e. U, V and W) has been forced with atmospheric fluxes from the high-resolution (50 km) ARPERA dataset (Herrmann and Somot, 2008; Herrmann et al., 2010). NEMO-MED12 is forced by ARPERA daily fields of momentum, evaporation and heat fluxes over the period 1958-2013. For the surface temperature condition, a relaxation term to sea surface temperature (SST) from ERA40 is applied for the heat flux (Beuvier et al., 2012b). This term acts as a first-order coupling between the ocean model's SST and the atmospheric heat flux (Barnier et al., 1995), ensuring consistency between these two terms. The value of the relaxation coefficient is spatially constant and is taken to be -40 W m$^{-2}$ K$^{-1}$, following the CLIPPER Project Team (1999). It corresponds to a 1.2-day restoring timescale for a surface layer of 1 m thickness (Beuvier et al., 2012a).

Numerous studies on ocean dynamics and biogeochemical cycles in the Mediterranean have been carried out using the NEMO-MED12 model (e.g., Brossier et al., 2011; Beuvier et al., 2012b; Soto-Navarro et al., 2014; Ayache et al., 2015a, b, 2016, 2017, 2023; Palmiéri et al., 2015; Guyennon et al., 2015; Richon et al., 2018, 2019). The NEMO-MED12 model represents well the main structures of the Mediterranean thermohaline circulation, with mechanisms having a realistic timescale compared to observations (Ayache et al., 2015a). However, some features of the simulation still need to be improved: for example, the weak formation of the Adriatic deep water (AdDW) as shown using anthropogenic tritium (Ayache et al., 2015a) and CFC simulations (Palmiéri et al., 2015). In the western basin, the WMDW is generally well simulated, but the propagation of the recently ventilated deep water to the south of the basin is underestimated (Ayache et al., 2015a; Palmiéri et al., 2015). All the details of the model and its parameterisations are described separately in (Beuvier et al., 2012b, a; Palmiéri et al., 2015; Ayache et al., 2015a).

### 2.2 Implementing water isotopes in the NEMO model

$\delta^{18}O_w$ and $\delta D_w$ were implemented in the regional high-resolution model NEMO-MED12 (release 3.4 and 3.6 of the NEMO model). A detailed description of the source code of the water isotopes package, with a user's guide, is available in the Supplementary Material (cf. Text S1 in the Supplement). The exact version of the model used to produce the results reported in this paper is archived on Zenodo (https://doi.org/10.5281/zenodo.10453745, Ayache et al., 2024, see supplement). All the abbreviations used in this paper are presented in Table. 1.

The Hydrogen and oxygen isotope compositions are reported as isotopic ratio anomalies to the Vienna Standard Mean Ocean Water reference value (VSMOW):

$$\delta^{18}O = \left( \frac{^{18}R}{^{18}R_{VSMOW}} - 1 \right) \cdot 10^3, \quad where \quad ^{18}R = \frac{^{18}O}{^{16}O} \tag{1}$$

$$\delta D = \left( \frac{^{D}R}{^{D}R_{VSMOW}} - 1 \right) \cdot 10^3, \quad where \quad ^{D}R = \frac{^2H}{^1H} \tag{2}$$

where $^{18}R_{VSMOW}$ and $^{D}R_{VSMOW}$ are the SMOW standard ratios for $^{18}O$ and D respectively. The natural abundances of the oxygen and hydrogen isotopes are $^{16}O$:$^{17}O$: $^{18}O$=0.9976: 0.00038: 0.00205, and $^1H$: $^2H$ (D)= 99.985: 0.00015 (Mook et al. (1974), IAEA ; Gat, 1996).

For simplicity, we explain the implementation of the water isotope in the NEMO-MED12 model using $\delta^{18}O_w$. Equations
for $\delta D_w$ are readily obtained by replacing the isotopic ratio where relevant. Water isotopes behave as conservative tracers in the ocean; they are only modified by fluxes across open boundaries (Craig and Gordon, 1965; Schmidt, 1998; Delaygue et al., 2000; Roche et al., 2004). The isotopic composition is determined on post-processing because here we transport the isotopic ratio (see equation 1), which allows us to carry a single tracer "$^{18}R$" instead of two tracers "$^{18}O$ and $^{16}O$". This reduces the computation time on the machine, which is a crucial factor in the performance of the model, especially in a very long palaeo-
simulation. It is a common practice to transport the isotopic ratio rather than the individual species. For example, radiocarbon distribution ($^{14}C/C$) in the Mediterranean Sea (Ayache et al., 2017) and $^{18}O/^{16}O$ of precipitation (Risi et al., 2010b). Therefore, the equation governing the transport of the isotopic ratio in the ocean is:

$$\frac{\delta}{\delta t} ^{18}r + \nabla \cdot (u^{18}r - K \cdot \nabla^{18}r) = 0 \tag{3}$$

where u is the 3-D velocity field, and K is the diffusivity tensor. It should be noted that the isotopic ratio $^{18}r$ in equation 3
is relative to the total of all isotopic forms. If we neglect the low abundant $^{17}O$ then the relationship between $^{18}r = ^{18}O/O$ and $^{18}R = ^{18}O/^{16}O$ is straightforward.

$$^{18}r = \frac{^{18}R}{(1 + ^{18}R)} \quad and \quad ^{18}R = \frac{^{18}r}{(1 - ^{18}r)} \tag{4}$$

The water isotopes are implemented using the passive tracer engine "TOP: Tracers in Ocean Paradigm" of the NEMO-MED12 ocean model by providing all physical constraints/boundaries of $\delta^{18}O$ and $\delta D_w$ and pseudo-salinity tracers (see Text
S1 in Supplement). Here, we used the offline coupling mode. In this method, the physical variables i.e., the circulation fields (U, V, W) and mixing coefficients (Kz) are previously computed by the NEMO-MED12 dynamical model (Beuvier et al., 2012a) and used to propagate the tracers in the ocean. The physical forcing fields are readed and interpolated at each model time step, i.e., the circulation fields (U, V, W) previously computed by the dynamical model are read daily and interpolated to

give values for each 20-minute time step. NEMO-related forcings are provided at a day frequency while isotopic-related fluxes are given monthly (see below for the atmospheric forcing).

The same approach has been used to simulate the neodymium budget in the present Mediterranean Sea (Ayache et al., 2023) and the past isotopic distribution of Nd (Vadsaria et al., 2019), the anthropogenic tritium invasion (Ayache et al., 2015a), the distribution of CFCs (Palmiéri et al., 2015) and anthropogenic carbon (Ayache et al., 2017). The ocean isotopic ratios are initially set to an average value for the Mediterranean basin of: $\delta^{18}O_w$ = 1.5 ‰ and $\delta D_w$ =8 ‰, and the pseudo-salinity tracer is set to 37 (we have initialised the simulations with these values to save a little computing time on the machine). The simulation was conducted over 30 years following a 44-year spin-up period (1958–1980 repeated twice), ensuring model stability for over 75 years. The years of hydrodynamic forcing were randomly selected from precalculated circulation fields spanning 1958 to 2013 (Beuvier et al., 2012a). The objective of this method is to minimize the impact of extreme variability effects, such as the Eastern Mediterranean Transient (EMT) or the Western Mediterranean Transition (WMT), on the simulated circulation (Roether et al., 2006; Schroeder et al., 2008). The spin-up strategy was adapted from previous passive tracer simulations, such as neodymium and tritium studies (Ayache et al., 2015a, 2016). All output fields in Tab S2 are routinely calculated.

## 2.3 Atmospheric fluxes and river runoff in un-coupled mode

The boundary conditions at the ocean-atmosphere interface over the Mediterranean regions for the water isotope simulation ($\delta^{18}O$ and $\delta D_w$) are given by the isotopic version of the atmospheric model with a comprehensive representation of water isotopes (LMDZ-iso GCM; Risi et al. (2010b)). They consist of climatological gross fluxes of evaporation and precipitation with their isotopic composition (Fig. 1). This ensures consistency between water (evaporation and precipitation) and isotopic fluxes, which is of primary importance here, since their balance generates our tracer distribution, as discussed in Delaygue et al. (2000) and Juillet-Leclerc et al. (1997).

Here we force the simulations from the global isotopic atmospheric model LMDZ-iso (Risi et al., 2010b), which is available with two horizontal resolutions; on a coarse latitude–longitude grid R96 (2.5°×3.75°), the vertical grid of LMDZ-iso extends over 39 layers. The LMDZ-iso Atmospheric simulation was conducted following the Atmospheric Model Intercomparison Project (AMIP) protocol, as presented in Risi et al. (2010b), utilising prescribed monthly and interannually varying SST and sea ice, in addition to a constant $CO_2$ value of 348 ppm for the present-day situation. The impact of these low $pCO_2$ values in comparison to the current value of 421 ppm is constrained by the fact that the model has been evaluated against in-situ data sampled primarily in the 1980s (see Risi et al. (2010b, 2013) for more details on the atmospheric simulation). The aim is to assess the model's performance in the present climate and against in-situ data observed between 1982 and 2022. Therefore, we have opted to use the climatological mean of the LMDZ-iso 1990-2020 simulation as boundary conditions. This choice was made to minimize the warming trend during this period and to ensure that the precipitation and evaporation simulated by the LMDZ-iso model for the current climate situation are as close to the average state as possible, with minimal impact from inter-annual variability. LMDZ-iso simulates reasonably well the spatial and seasonal variations of both $\delta^{18}O$ and deuterium excess (d-excess= $\delta D_w - 8* \delta^{18}O_w$, Dansgaard, 1964). These fluxes were carefully interpolated onto the NEMO-MED12 grid (see Fig. 1). It must be acknowledged that the spatial resolution of LMDZ-iso is relatively coarse for the Mediterranean Sea. It

was necessary to use low-resolution forcing on the simulated isotopic composition concentration because no higher resolution atmospheric isotopic model simulations similar to the dynamical forcing of NEMO-MED12 dynamical simulation (50km) are available at the moment. We therefore performed some sensitivity tests of the results by changing the horizontal resolution of LMDZ-iso between R96 and R144, the results of these experiences are shown in Appendix C. The impact of this low resolution on the simulated isotopic composition is limited because we used pre-calculated dynamical fields of the NEMO-MED12 model (in off-line mode) forced by a higher resolution atmospheric model (50 km) ARPERA dataset (Beuvier et al., 2012b; Herrmann and Somot, 2008; Herrmann et al., 2010).

Isotopes are included in the river discharge of the land surface model ORCHIDEE (Risi et al., 2016) but the isotopic version of ORCHIDEE is too old to be coupled with LMDZ-iso. Therefore, as previously done in Delaygue et al. (2000) for the global ocean, we used river discharge estimation from observations and attributed the isotopic composition of precipitation at the river mouth. River inputs are introduced as freshwater sources at river mouths in the surface layer (Fig. 1g, h, i). We used the climatological mean of the interannual dataset of Ludwig et al. (2009) to compute monthly runoff values of the 33 main river mouths covering the entire Mediterranean draining basin (RivDis dataset Vörösmarty et al., 1996). The Nile played a crucial role in freshening surface water during sapropel events. However, since the construction of the Aswan High Dam in 1965, its influence has decreased (ElElla, 1993; Nixon, 2003). As a result, the Nile is no longer a major contributor to the current state of the Mediterranean Sea.

The values of the inputs of the other rivers are averaged in each Mediterranean sub-basin and placed as coastal runoff in each MED12 coastal grid point of these sub-basins (Fig. 1g, h, i), as done in Beuvier et al. (2012a) and in Palmiéri et al. (2015). Similarly, since it is difficult to couple the old isotopic version of ORCHIDEE with the current version of LMDZ, we adopt an alternative solution to represent the isotopic flux carried by rivers to the ocean: this flux is calculated as $\mathcal{R}_R = \mathcal{R}_P \times R$ where R is the runoff prepared from the data of Ludwig et al. (2009) and Vörösmarty et al. (1996) (see above) and $\mathcal{R}_P$ is the isotopic ratio in precipitations at the same time and location (Fig. 1) as adapted from Delaygue et al. (2000). We have performed some sensitivity simulations to better assess the effect of the $\partial^{18}O_{river}$. The results of these experiences are included in the Appendix to further clarify this point (see Appendix E). The exchange with the Atlantic Ocean is performed through a buffer zone between 11°W and the Strait of Gibraltar, where 3-D water isotopes ($\delta^{18}O_w$ and $\delta D_w$) and salinity model fields are relaxed to the observations from the Global gridded data set of oxygen isotopic composition in seawater (LeGrande and Schmidt, 2006) and using global model outputs after multiple sensitivity simulations (not shown here).

Let $\mathcal{E}$, $\mathcal{P}$, $\mathcal{R}$ represent evaporation, precipitation, and run-off, respectively, then the following boundary condition is relevant at the sea surface.

$$\mathcal{F}^{18}O = \mathcal{E}(\mathcal{R}_s - \mathcal{R}_E) - \mathcal{P}(\mathcal{R}_S - \mathcal{R}_P) - \mathcal{R}(\mathcal{R}_S - \mathcal{R}_R) \tag{5}$$

where $\mathcal{R}_S$ is the isotopic ratio of the oceanic surface, $\mathcal{R}_E$, $\mathcal{R}_P$ and $\mathcal{R}_R$ are the isotopic ratios of evaporation (E), precipitation (P) and run-off (R). In our study, we utilized the offline uncoupled mode of NEMO, which employs pre-calculated dynamics. This mode operates with a fixed volume and explicit fluxes of evaporation, precipitation, and runoff. Alternatively, the online

coupled mode of NEMO can be employed to compute dynamic variables (such as circulation fields U, V, and W) in real time. The sea surface elevation and model layer thicknesses are adjusted by the freshwater flux (E-P-R), consequently affecting the model volume. It is essential to ensure that total volume variations accurately correspond to the E-P forcing used to drive the isotopic module, thus maintaining the perfect conservation of tracer content.

### 2.4 Pseudo salinity in stand-alone ocean model

The water fluxes from the stand-alone experiments with LMDZ-iso are not identical to those constraining NEMO-MED12. Therefore, $\delta^{18}O_w$ or $\delta D_w$ computed with the water fluxes obtained with LMDZ-iso would not be consistent with the salinity predicted by NEMO-MED12. For this reason, we compute a "pseudo-salinity" $S_w$ (Delaygue et al., 2000; Roche et al., 2004). This additional passive tracer does not affect ocean dynamics. Its sole purpose is to provide a coherent assessment of the isotopic fields generated by the model. The evolution equation for $S_w$ is given by equation D2 where we replace $\mathcal{R}_s$ by $S_w$ and where a zero salinity is associated to the water fluxes (i.e. $\mathcal{R}_E$, $\mathcal{R}_P$ and $\mathcal{R}_R = 0$ when solving equation D2 for $S_w$). This passive tracer, hereafter called 'pseudo-salinity', is calculated "offline". The basic understanding of these atmospheric fluxes, $\mathcal{F}^{18}O$ and $\mathcal{F}^S$, is that evaporation tends to increase the surface salinity, and the $^{18}O/^{16}O$ ratio, in contrast to precipitation and runoff. See Appendix D for more details on the concept of pseudo-salinity.

### 2.5 Datasets of $\delta^{18}O_w$ and $\delta D_w$ to evaluate the simulation

For comparison with our model results, we used published in-situ data in the Mediterranean Sea (https://data.giss.nasa.gov/cgi-bin/o18data/geto18.cgi) including Epstein and Mayeda (1953), Stahl and Rinow (1973), Pierre et al. (1986), Gat et al. (1996), Pierre (1999), Voelker (2017), and Reverdin et al. (2022). We also used the global gridded data set of oxygen isotopic composition in seawater from (LeGrande and Schmidt, 2006) to compare the observed and modelled large-scale oceanic $\delta^{18}O_w$ distribution (i.e., the east-west gradient). While $\delta D_w$ observations in Mediterranean waters are not as widespread as $\delta^{18}O_w$, there are some data available in the eastern basin from Gat et al. (1996), and from Reverdin et al. (2022) in the western basin to validate our simulations.

## 3 Results

### 3.1 Simulated present-day distribution of $\delta^{18}O_w$

As a preliminary assessment of our model results, we evaluated the spatial distribution of $\delta^{18}O_w$ in surface waters, zonal vertical sections and basin average vertical profiles (see Table 2 and Fig. 2) forced by the coarse-resolution version (R96) of the LMDZ-iso model. The EMed is enriched by more than 0.45 ‰ (see Table 2) compared to the WMed. The largest variation of $\delta^{18}O_w$ of the water is simulated in the surface waters with a strong east-west gradient (Fig. 2a); the $\delta^{18}O_w$ value is up to 2 ‰ in the EMed, but only 1.55‰ in the WMed. This trend reflects the east-west gradient of oceanic evaporation, which distinguishes the higher evaporation in the EMed than in the WMed (Fig. 1). The $\delta^{18}O_w$ distribution shows a north-south

enhancement in the eastern basin (Fig. 2a) with less enriched surface water in the Aegean and Adriatic basins, two regions characterised by active vertical mixing homogenising the water column, and a relatively high contribution of river discharge to this region (e.g. the Po River in the Adriatic basin). The vertical $\delta^{18}O_w$ distributions are well captured by the model as shown in the west-to-east section and the vertical profile across the Mediterranean (Fig. 2). The intermediate waters (200-800 m depth) form a more homogeneous layer relative to the surface waters. However, the $\delta^{18}O_w$ values decrease towards the west by 0.35‰ at most, which is due to gradual dilution by mixing with the deeper water masses and the Atlantic water. The deep water exhibits homogeneous $\delta^{18}O_w$ values similar to the simulated values in the intermediate water, indicating well-ventilated conditions due to active winter convection.

Comparison of the model output with in situ data shows that the model reproduces well the observed east-west gradient that characterises the surface waters (Fig. 2a Table 2), and correctly reproduces the zonal gradients observed in the intermediate and deep waters (Fig. 2b and 2c). The simulated mean vertical profile of $\delta^{18}O_w$ is consistent with the observations in the western basin of $\delta^{18}O_w$ values (Fig. 2d). The spreading of Atlantic water in the surface of the Alboran basin is well reproduced in the simulation (Fig. 2a, 2d). In the eastern basin, the highest value of $\delta^{18}O_w$ is well reproduced in the simulation, but the model largely underestimates the mean values of the observations in the intermediate and deep waters (Fig. 2e, 2e). This offset is related to the weak formation of the simulated EMDW in the Adriatic sub-basin, as already noted by Ayache et al. (2015a); Palmiéri et al. (2015). To further evaluate the relationship between the in situ data and the simulated $\delta^{18}O_w$, the longitudinal distribution of $\delta^{18}O_w$ is examined for each basin (Fig. 3b, 3c). A pronounced longitudinal gradient is found for simulated and observed $\delta^{18}O_w$ values, with more enriched values in the EMed (between 27°E and 36°E) and more depleted values in the WMed (between -6°E and 11°E) with an intermediate value in the central basin (Fig. 3b, 3c). The observed salinity agrees well with the simulated pseudo-salinity results (Fig. 3), in contrast to the highly variable in situ $\delta^{18}O_w$ values.

## 3.2  The $\delta^{18}O_w$-salinity relationship in the Mediterranean waters

The lower two panels in Fig. 3 show the depth profiles of salinity in relation to $\delta^{18}O_w$ from in-situ data (Fig. 3d) and model output (Fig. 3e) forced by the coarse-resolution version (R96) of the LMDZ-iso model. The more evaporated water in the eastern basin (pseudo-salinity up to 38.9) matches well with more enriched water ($\delta^{18}O_w$ above 1.98 ‰), especially at intermediate depths (300-700 m) corresponding to the LIW layer. The decrease in $\delta^{18}O_w$ and salinity in the deep water is well captured by the model (Fig. 3c, 3d, 3e). However, the model tends to overestimate the value of $\delta^{18}O_w$ associated with a lower salinity in the WMed (salinity = ∼36.4), i.e. the salinity of the inflowing Atlantic waters (Fig. 3c and Fig. 3e).

To further analyse the relationship between $\delta^{18}O_w$ and salinity, we plot the regression slope of $\delta^{18}O_w$ versus salinity for the available in situ data (Fig.4a, b, c) and from the model output (Fig.4 d, e, f). There is a significant positive correlation between salinity and $\delta^{18}O_w$ from the model results (r$^2$ = 0.82) and from the in-situ data (r$^2$= 0.60) for the whole Mediterranean Sea. EMed shows the weakest correlation between salinity and $\delta^{18}O_w$ (r$^2$ = 0.20, Fig. 4c). The whole set of in-situ data values measured in the Mediterranean waters defines the following linear equation: $\delta^{18}O_w = 0.29S - 9.46$ (Fig. 4a); the equation becomes: $\delta^{18}O_w = 0.26S - 8.60$ in the WMed and $\delta^{18}O_w = 0.25S - 8.19$ in the EMed (Fig. 4b, 4c). The difference between the two equations remains fairly small, with a similar slope in the EMed basin and different intercepts. The model simulated

a similar slope to in-situ data throughout the basin ($\delta^{18}O_w$ = 0.25S - 8.01) and the zonal trend is comparable to observation (0.25 and 0.26 for the WMed and the EMed, respectively; Fig. 4e and 4f). Pierre (1999) estimated a similar slope (0.25) for the whole Mediterranean water and 0.27 in the Alboran basin (western basin). Fig. 5a displays the temporal distribution of the $\delta^{18}O_w$-salinity slope in Mediterranean surface water, computed using simulated climatology over last 30 years. Low values (around 0.3, Fig. 5) as well as a weak correlation (0.24, Fig. 4f) were calculated in the eastern basin. The lower slopes reflect the impact of the evaporation surplus in the EMed (Voelker et al., 2015). High values of the slope are simulated in the western basin (> 0.5, Fig. 5a) especially in the Alboran basin which is influenced by Atlantic water characterised by a $\delta^{18}O_w$-S slope of 0.48 (Laube-Lenfant, 1996; Pierre, 1999), and 0.32 obtained by Voelker et al. (2015) in the North East Atlantic with a strong bias towards subtropical waters. While this simulated longitudinal trend appears to agree with observations (Fig. 4 and 5), it is important to note that there are some additional longitudinal variations in slope, particularly in the Aegean Sea and south-easternmost part of the Levantine basin. Fig. 5b displays the spatial $\delta^{18}O$-salinity slope from the model outputs. For each grid point, it is computed as the slope of the $\delta^{18}O_w$ to salinity linear regression, based on the simulated surface values from the 12 surrounding grid points. The mean slope of the spatial regression ($\sim$ 0.3) is relatively similar to the mean value of temporal regression (Fig. 5a). However, the slope based on spatial regression shows greater variation, mainly due to the oceanic circulation, particularly in areas of high mesoscale activity (i.e. the Algerian and Levantine basins), with potentially greater transport/change in salt and water content in the water column caused by the oceanic mesoscale eddies.

### 3.3  Present-day distribution of deuterium ($\delta D_w$) and d-excess

Since identical boundary fluxes (precipitation, evaporation, and river runoff) drive both $\delta^{18}O_w$ and $\delta D_w$ isotopes in the surface water, the zonal gradient patterns between EMed and WMed are strikingly similar (Fig. 6), with the most enriched areas ($\delta D_w$ values >= 8 ‰) located in the more evaporated EMed basin and the most depleted areas in the WMed basin (especially the Alboran basin with $\delta D_w$ values <= 6 ‰). As for the $\delta^{18}O_w$, the $\delta D_w$ values are lower in the Aegean basin, which may be related to a relatively high freshwater contribution (P and R) and an active vertical mixing. The spatial structures of $\delta D_w$ simulated by the model are consistent with the observations available in the EMed and in the WMed surface water (Fig. 6) with values slightly lower than in situ data in the surface and intermediate waters. The distributions are more uniform in the deep water. Simulated $\delta D_w$ exhibit a linear relationship with $\delta^{18}O_w$ (Fig. 7a), and salinity (Fig.7b) with a significant correlation ($r^2$=0.98 and 0.94, respectively). $\delta D_w$ observations in the Mediterranean are not as extensive as those of $\delta^{18}O_w$. Therefore, there is currently not enough data to constrain and validate our $\delta D_w$ simulation as shown in Fig. 7c, where a weak correlation ($r^2$ = 0.25) was found between the few data available in the eastern basin and $\delta D_w$ simulated in the same data location.

The deuterium excess "d-excess" reflects the relationship between the isotopic ratios of hydrogen and oxygen. This indicates the kinetic (non-equilibrium) fractionation effects that occur when water is evaporated from oceanic regions (Dansgaard, 1964). The simulated mean surface water d-excess values range from -4.4‰ to -1.5 ‰, with relatively small variations (variance = -0.27 ‰), and a clear negative shift in simulated d-excess values was observed across the basin (Fig. 8). The WMed basin is enriched in d-excess compared to the EMed basin, and the regions with the lowest d-excess are located in the Levantine sub-basin. In-situ observations of d-excess from Gat et al. (1996) and Reverdin et al. (2022) show an important E-W gradient,

with higher values recorded in the Western Mediterranean (WMed) and lower values in the Eastern Mediterranean (EMed). The simulated D-excess values closely match the in-situ data from the EMed, whereas the model significantly underestimates observed $\delta D_w$ values in the WMed (Fig. 8). The model results show an increase in d-excess for water masses with higher $\delta^{18}O_w$ depletion, as suggested by Xu et al. (2012) using the MPI-OM model simulating water isotope variation on a global scale. These negative values are generally in accordance with the positive values of deuterium excess in atmospheric water vapor and precipitation observed and simulated in this region, associated with the dryness of near-surface air (Pfahl and Wernli, 2008). In a more recent study, Benetti et al. (2014) observed a d-excess ranging from -1.56 to -1.72 in the surface waters of the eastern subtropical Atlantic. Their findings reveal a contrasting trend between increasing $\delta^{18}O_w$, $\delta D_w$, and decreasing d-excess, which corresponds closely with our simulated values. The authors suggest that d-excess variations are predominantly influenced by humidity and wind speed rather than mixing effects.

### 3.4 Variations of $\delta^{18}O_{calcite}$ in the Mediterranean Sea

A useful tool for reconstructing past climate is the isotopic composition of foraminiferal shells from sediment cores (Shackleton, 1967). However, due to temperature-dependent fractionation, the isotopic signature of the CaCO$_3$ shell ($\delta^{18}O_c$) differs from $\delta^{18}O_w$. The $\delta^{18}O_c$ values depend on both $\delta^{18}O_w$ and seawater temperature at calcification depth. For planktonic foraminifera, the isotopic fractionation relationships during calcification can be assumed to be represented by an equation for equilibrated calcite. We used a paleotemperature equation for inorganic calcite by Kim and O'Neil (1997) modified by Bemis et al. (1998), with the use of the 0.27 ‰ correction from VSMOW (Vienna Standard Mean Ocean Water) to VPDB (Vienna Pee Dee Belemnite) conversion. The equation was applied to both the model output and the available in-situ data, as presented in Section 2.5.

$$\mathcal{T} = 16.1 - 4.64(\delta^{18}O_c - \delta^{18}O_{sw}) + 0.09(\delta^{18}O_c - \delta^{18}O_{sw})^2 \tag{6}$$

$$\delta^{18}O_{c(VPDB_{‰})} = (\delta^{18}O_{sw(VSMOW_{‰})} - 0.27) + \frac{4.64 - \sqrt{21.53 - 0.36(16.1 - \mathcal{T}^\circ_C)}}{0.18} \tag{7}$$

The relationship of $\delta^{18}O_c$ with temperature and the influence of seasonality are shown in Figure 9. The mean annual $\delta^{18}O_c$ values for surface seawater were computed using the model outputs of surface $\delta^{18}O_w$ and surface water temperature. The simulated annual mean $\delta^{18}O_c$ vary from -0.8 to 2 ‰ in the surface (Fig. 9b), with higher values in the northern part of the Mediterranean and lower values near the southern coast. This latitudinal gradient of $\delta^{18}O_c$ is different from the zonal pattern of $\delta^{18}O_w$ (Fig. 2a) and is related to the effect of temperature (Fig. 9c). Considering this strong temperature dependency, the seasonal variability of $\delta^{18}O_c$ was examined (Fig. 9d). The highest simulated $\delta^{18}O_c$ values were obtained for winter (February, March) and the lowest values for summer/autumn. The $\delta^{18}O_c$ calculated from in situ $\delta^{18}O_w$ and measured seawater temperature show the same seasonal trend (Fig. 9e). Even if the available observational data do not cover all the months of the year, our results indicate the importance of temperature effects on $\delta^{18}O_c$ in the Mediterranean Sea. We note here that seasonal variation of $\delta^{18}O_w$ is small in the surface layer of both the eastern and the western basins (see Fig. A1 in appendix). The $\delta^{18}O_c$

variation is mainly localised in surface and intermediate waters (first 300m depth) (Fig. A2 in appendix). The comparison of simulated and observed $\delta^{18}O_c$ (0-50 m depth) shows a strong positive correlation with a similar range of variability (between -0.22 and 1.91 ‰ from the model output, and between -0.82 and 1.97 from the in-situ data) as shown in Fig. 9f. In this study, we analysed the impact of temperature on $\delta^{18}O_c$ calculations, both in a global model and at high regional resolution. Please refer to Appendix B for further details.

## 4    Discussion

This study provides the first simulation of the water isotopes ($\delta^{18}O_w$ and $\delta D_w$) in the Mediterranean Sea covering the entire basin. These two tracers were implemented in the high-resolution regional model NEMO-MED12. New insights into the distribution of water isotopes and their relation to salinity in the Mediterranean Sea were obtained by comparing this numerical study with in-situ data. Analysis of the results from an oceanic point of view shows good agreement with the in-situ data, opening up a range of possibilities for long-term palaeoclimate simulations in this basin and the use of this modelling approach in coupled ocean-atmosphere models. The inputs and boundary conditions $\delta^{18}O_w$ and $\delta D_w$ were taken from a global atmospheric model with a low resolution and have been tested for the first time in this study with a regional model at a high resolution. Both observed $\delta^{18}O_w$ and $\delta D_w$ show a pronounced east-west gradient, characterised by more enriched water in the eastern basin than in the western basin. This gradient is well captured by the model, and is in good agreement with the available in-situ data. It is not possible to constrain and validate the $\delta D_w$ simulation due to the limited number of $\delta D_w$ observations. Thus, our discussion below focuses on the results of the $\delta^{18}O_w$ simulation.

     A significant correlation between model output and in-situ data ($r^2 = 0.68$) was obtained over the whole basin, with a higher correlation in the WMed basin than in the EMed basin. Our model also successfully simulates the observed vertical distribution of the water isotope composition of the Mediterranean water masses (Fig. 2, Fig. 6). Despite a slight bias in EMed due to the previously reported weak formation of AdDW, the vertical distribution compares favourably with the available in-situ data. Some improvements are still needed in certain aspects of the simulation. The model largely underestimated the mean $\delta^{18}O_w$ values of observations in intermediate and deep waters, and failed to simulate the highly enriched water in the eastern basin (up to 2.4, by ‰ Gat et al. (1996)). This inconsistency should be investigated in a fully coupled ocean-atmosphere model with a higher horizontal/vertical resolution of the atmospheric model. The advantage of using a coupled model lies in the consistent simulation of changes in the different components of the model (for instance between the precipitation over land, the ocean variability and runoff input from the land), and more realistic ocean-atmosphere feedbacks in the coupled model (Bretherton and Battisti, 2000; Schmidt et al., 2007).

     The main difference between the data and the numerical simulations is the smaller amplitude of $\delta^{18}O_w$, particularly in the eastern basin. This discrepancy can be explained in two ways. First, this may be due to the low spatial resolution of the isotope forcing. Vadsaria et al. (2020) showed that high resolution ($\sim$ 30 km of the atmospheric model) is critical to accurately capture the synoptic variability needed to initiate the formation of the intermediate and deep waters of the Mediterranean thermohaline circulation. Due to the peculiarities of the atmospheric circulation (high wind gusts in winter) and the oceanic circulation

(deep convection) in this intercontinental basin, high spatial resolution forcings are needed (Li et al., 2006). Nevertheless, a change in the horizontal resolution of the LMDZ-iso atmospheric model (from R96 to R144) does not improve the model results, and the model does not simulate the highest values of $\delta^{18}O_w$ observed by Gat et al. (1996) in the eastern basin at either resolution. There may be a certain threshold of spatial resolution below which the simulation is improved by a finer resolution. Unfortunately, LMDZ-iso simulations at resolutions finer than R144 are not yet available to test this hypothesis. Sensitivity tests were performed to investigate the effect of changing the resolution of the LMDZiso atmospheric model (between R96 and R144) and the oceanic model (between ORCA2 and NEMO-MED12), the results of which are presented in the supplementary material of this paper (see Appendix C).

The second hypothesis is that the discrepancy is due to the physics of the atmosphere, which is independent of the horizontal resolution. In parallel with the too low $\delta^{18}O_w$ in the western part of the basin, LMDZ-iso underestimates the depletion and d-excess of precipitation and vapour in this region (Risi et al., 2010a). These discrepancies in LMDZ-iso are consistent with the insufficient near-surface air dryness. The underestimated dryness would lead to a lower surface evaporative flux in LMDZ-iso, leading NEMO-MED12 to underestimate evaporative enrichment of surface water. In addition, the underestimation of water vapour depletion in the LMDZ-iso leads to an overestimation of evaporative flux (Craig and Gordon, 1965), which in turn leads to a further underestimation of evaporative enrichment of surface water by NEMO-MED12. The underestimation of the dryness in LMDZ-iso could be due to insufficient vertical resolution (Risi et al., 2012) or to a misrepresentation of shallow convection in this region (Hourdin et al., 2015). Such a discrepancy is not observed in the salinity data. To obtain larger spatial coverage, we used $\delta^{18}O_w$ obtained in the 1971 to 1990 period in addition to two data points acquired in 1949. Therefore, it is not impossible that temporal variation of $\delta^{18}O_w$ and different data quality with time could induce further scatter. Despite the smaller range of the $\delta^{18}O_w$, our parameterisation produced realistic general features of spatial distribution, particularly zonal trends in surface water. The results suggest that this approach can be used to generate water isotopic simulations with adequate validity at decadal time scales (i.e. 50 years of simulation), opening up the prospect of simulations at longer time scales in the context of palaeoclimate studies.

Mediterranean regional climatic conditions (i.e. excess of evaporation over precipitation) shape a specific relationship in surface waters between observed salinity and $\delta^{18}O_w$ values, characterised by a $\delta^{18}O_w$-S slope of 0.25, much lower than the slope value of 0.45 obtained in Atlantic surface waters (Pierre, 1999) and 0.32 calculated by Voelker et al. (2015) in the NE Atlantic. Our results are consistent with these findings: $\delta^{18}O_w$ shows a linear relationship with salinity, and the simulated slope of $\delta^{18}O_w$ with salinity (0.28) is very similar to that calculated by Pierre (1999) using in situ observations. The model simulated similar differences between the EMed and the WMed, with a steeper slope in WMed as computed using in-situ data. It is not surprising that there is a high correlation between these two fields, since the processes that affect $\delta^{18}O_w$ at the surface are also those that affect surface salinity. Nevertheless, especially in areas of high mesoscale activity (e.g. the Algerian Basin), the spatial slope $\delta^{18}O_w$-S of our simulation shows strong variations. The slope is therefore not homogeneous but depends on the local climate conditions (wind speed, temperature, etc.). It is well documented that mesoscale eddies can transport water, heat, salt and other tracers as they spread in the ocean, influencing water column properties and biological activities (Chelton et al., 2011; Dong et al., 2014). In summary, our simulation results indicate a significant deviation in the slope of the $^{18}O_w$-salinity

relationship compared to the global slope (Pierre, 1999; Voelker et al., 2015). The calculated slopes are consistently lower within this basin, reflecting the influence of evaporation surplus, as highlighted by Gat et al. (1996) in their study of the eastern Mediterranean basin.

The simulated $\delta D_w$- $\delta^{18}O_w$ relationship provides a realistic d-excess surface field. A comparison between data and model is not possible due to the lack of d-excess field data for the Mediterranean. However, the modelled d-excess is consistent with other modelling studies (e.g. Xu et al., 2012) which show an increase in d-excess for water masses more depleted in $\delta^{18}O_w$. Our simulations show similar negative d-excess values for the whole Mediterranean basin. Thus, assuming that the atmospheric d-excess signature is largely dominated by non-equilibrium isotope fractionation during evaporative processes of marine surface waters (Dansgaard, 1964; Gat et al., 1994), the remaining surface waters should have a negative d-excess value as simulated by our model. More recently, Benetti et al. (2014) observed a contrasting trend between increasing $^{18}O_w$, $\delta D_w$ and decreasing d-excess, suggesting that d-excess variations are predominantly influenced by humidity and wind speed rather than mixing effects. Simulating both $\delta D_w$ and $^{18}O_w$ is useful for paleoclimate applications involving both $\delta D$ and $^{18}O$ of natural archives, particularly when using this modelling approach in a fully coupled configuration. Notably, $\delta D_w$ in leaf waxes (Sachse et al., 2012) and speleothem fluid inclusions (van Breukelen et al., 2008) are useful for paleoclimate reconstructions.

An interesting tool for mapping potential changes in the oceanic circulation over time could be a data-model comparison exercise for the $\delta^{18}O$ of calcite in past climates. We can calculate $\delta^{18}O_c$ and compare our model results with $\delta^{18}O_c$ calculated from in-situ data, since water temperatures and $\delta^{18}O_w$ are explicitly simulated by our model (see Sec 3.4). The results show that the surface $\delta^{18}O_c$ distributions derived from the model results are consistent with the general spatial pattern of $\delta^{18}O_c$ measurements in the present-day situation. Higher values of $\delta^{18}O_c$ are simulated mainly in the northern part of the Mediterranean as compared to the southern part. The difference between $\delta^{18}O_w$ and $\delta^{18}O_c$ is related to the Mediterranean temperature pattern, with a high negative correlation, especially in the surface layer. Calcite $\delta^{18}O_c$ is widely used in paleoclimate research. Understanding its seasonal variability is crucial for reconstructing past climates. The influence of seasonal temperature variability on $\delta^{18}O_c$ (equation 6) is important, particularly in the Mediterranean Sea because of marked seasonal thermal contrast. The $\delta^{18}O_c$ values are determined by both $\delta^{18}O_w$ and the seawater temperature at the calcification depth. For planktonic foraminifera such as Globigerinoides ruber and Globigerina bulloides, the calcification depth typically ranges from 0 to 100 meters, though variations exist depending on the basin (Coppa et al., 1980; Grazzini et al., 1986). The season of maximal foraminiferal production can be estimated by data from sediment traps. For instance, G. ruber and G. bulloides have been associated with calcification seasons in October-November and April-May according to Kallel and Labeyrie (1997), while others suggest January-March (Avnaim-Katav et al., 2019) and February-April (Rigual-Hernández et al., 2012). In this context, we used our model results to explore the relationship between the $\delta^{18}O_c$ and temperature. We employed a paleotemperature equation for inorganic calcite by Kim and O'Neil (1997), modified by Bemis et al. (1998), as shown in Fig. 9. Our simulations indicate that the highest $\delta^{18}O_c$ values occur during winter (February, March), while the lowest values are observed during summer/autumn. Although the available observational data do not cover all months of the year, our results align with existing data, highlighting the significant influence of temperature on $\delta^{18}O_c$ in the Mediterranean Sea. Nonetheless, a dedicated study should be conducted to further elucidate the seasonal aspect.

To extend this study, certain sensitivity tests/modelling developments must be performed. For the present-day situation, it would be useful to evaluate the influence of different forcing factors on the distribution of water isotopes in the Mediterranean (e.g. the influence of the inflow/outflow from the Atlantic at the Strait of Gibraltar, the influence of surface runoff, etc.). In our experimental set-up, river runoff is computed by considering the isotopic signature of precipitation. This assumption can lead to an unrealistic isotopic composition of the river runoff. Future studies will improve the representation of water isotopes in river runoff using a coupled ocean-atmosphere-land model. For past climates such as the Holocene (i.e. sapropels events), appropriate oceanic circulation and atmospheric fluxes could be combined to estimate differences with the present-day situation. This could help to test and better understand the reconstructed past data. The use of transient simulation offers an interesting test-bed to progress on this issue, especially to evaluate the Mediterranean circulation sensitivity to hydrological/thermal perturbation during the most recent Holocene sapropel S1 (10.5 to 6.1 cal ka BP) and the last interglacial sapropel S5 (128-122 ka, Grant et al., 2016) which occurred under warm conditions with strong seasonality and a high sea level stand. Regional climate models can bridge the gap between the coarse resolution of global climate models and the regional-to-local scales. They provide a more realistic representation of physical processes and climate feedback compared to global climate models. This is especially true for the Mediterranean region with its complex geology (Li et al., 2006). The water isotope modelling package presented in this study can be used in coupled regional configurations, such as regIPSL (Drobinski et al., 2012), which may assist in the preparation of a global-scale coupled version. Additionally, a sequential architecture of a global-regional modelling platform has been developed by Vadsaria et al. (2020) using the same dynamical model NEMO-MED. This platform can be used sequentially in a wide range of paleoclimate contexts, from the Quaternary to the Pliocene, with a regional model that is forced by a global model.

## 5   Summary and conclusions

Here, for the first time, stable water isotopes were successfully implemented in a high-resolution regional model of the Mediterranean (called NEMO-MED12-watiso-v1.0) forced by the atmospheric model LMDZ-iso. The isotopic composition of seawater $\delta^{18}O_w$ and $\delta D_w$ is simulated explicitly by the oceanic model. The model successfully simulates the observed basin-scale pattern of $\delta^{18}O_w$ and E-W gradients in surface water, evidencing the larger degree of evaporation of surface waters in the eastern basin. It also successfully reproduces the vertical distribution of $\delta^{18}O_w$ in Mediterranean waters masses. Furthermore, the simulated $\delta^{18}O_w$-salinity relationships are also in good agreement with the data, with a smaller slope in the EMed than in the WMed, and a slope of 0.25 across the basin. The modelled d-excess values are in good agreement with other modelling studies, with an enhancement of d-excess for water masses depleted in $\delta^{18}O_w$. Such negative d-excess values are found throughout the Mediterranean Sea in our simulation results. We examine the relationship of $\delta^{18}O_c$ with temperature and the influence of seasonality. The gradient of $\delta^{18}O_c$ is different from the pattern of $\delta^{18}O_w$ due to the effect of temperature, with the highest values obtained in winter and the lowest values in summer/autumn.

Improvements are needed in certain aspects of the simulation. A global atmospheric model simulation (LMDZ-iso) with a relatively coarse resolution was used for the isotopic forcing fluxes of precipitation and evaporation. In order to generate

steeper gradients in the hydrological cycle variables over the Mediterranean basin (evaporation, precipitation) and to improve the isotopic simulation of the present study, a higher spatial resolution (< 50 km) may be required. Here we calculate the isotopic composition of rivers based on the isotopic composition of precipitation, which means that the enriched $\delta^{18}O$ in rivers due to evaporation is not included in our simulation. It is recommended that a future study better represents the $\delta^{18}O_{river}$

480 (see Appendix E). It would be interesting to compare how NEMO-MED12 responds to inputs from different isotope-enabled atmospheric GCMs, as documented in SWING2 (Risi et al., 2012). In addition, an intercomparison of results from different coupled models could be valuable as an extension of SWING2. The use of a coupled system would provide more physical coherence between atmosphere, land, and ocean components and could allow a more reliable simulation of Mediterranean water isotopes. Present-day climate conditions were the focus of this first evaluation of the new stable water isotope package

485 implemented in the NEMO-MED12 model. The model will then be used for different palaeoclimatic conditions to improve our knowledge of past marine isotopic changes and to use it in palaeoclimate reconstructions.

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

*Author contributions.* MA, JCD, AM contributed to the model development, simulations, and diagnostics. MA, JCD, KT, CA, GR were involved in the writing and revision of the manuscript.

*Competing interests.* The authors declare that they have no conflict of interest

**Table 1.** Abbreviations and Units

| Abbreviation | Presentation | Unit |
|---|---|---|
| $\delta^{18}O_w$ | Delta-Oxygen-18 in seawater (see equation 1) | ‰ |
| $\delta D_w$ | Delta-Deuterium in seawater (see equation 2) | ‰ |
| $\delta^{18}O_c$ | Delta-Oxygen-18 in sea calcite (see equation 7) | ‰ |
| d-excess | The deuterium excess "d-excess$= \delta D - 8 \times \delta^{18}Ow$" | ‰ |
| P | Precipitation | kg/m$^2$/s |
| E | Evaporation | kg/m$^2$/s |
| R | River runoff | kg/m$^2$/s |
| $^{18}r$ | isotopic ratio $^{18}r = \dfrac{^{18}O}{Q}$ (see equation 4) | |
| $^{18}R$ | isotopic ratio $^{18}R = \dfrac{^{18}O}{^{16}O}$ (see equation 4) | |
| $R_P$ | isotopic ratio in precipitations | |
| $R_E$ | isotopic ratio in evaporation | |
| $R_R$ | isotopic ratio in river runoff | |
| R96 | coarse-resolution grid (2.5°×3.75°) of LMDZiso atmospheric model | |
| R144 | Medium resolution grid (1.27°× 2.5°) of LMDZiso atmospheric model | |
| $^{18}R_{VSMOW}$ | Standard Mean Ocean Water standard ratios for $^{18}O$ | |
| $^{D}R_{VSMOW}$ | Standard Mean Ocean Water standard ratios for D | |
| VSMOW | Vienna Standard Mean Ocean Water reference value | |
| VPDB | Vienna Pee Dee Belemnite | |
| LIW | Levantine Intermediate Water | |
| AW | Atlantic Water | |
| MOW | Mediterranean Outflow Water | |
| WMDW | Western Mediterranean Deep Water | |
| AdDW | Adriatic deep water | |
| SST | Sea Surface Temperature | °C |
| IAEA | International Atomic Energy Agency | |
| WMed | Western Mediterranean basin | |
| EMed | Eastern Mediterranean basin | |
| CaCO$_3$ shell | Planktonic foraminifera shells | |

**Table 2.** Mean, standard deviation, minimum and maximum values from the model outputs and available in-situ data from (Epstein and Mayeda, 1953; Stahl and Rinow, 1973; Pierre et al., 1986; Gat et al., 1996; Pierre, 1999) calculated in the surface water (0-100 m depth) of the whole basin, eastern, and western basins.

|             |      | Model | In-situ data |
|-------------|------|-------|--------------|
| Whole basin | Mean | 1.55  | 1.46         |
|             | Min  | 0.74  | 0.7          |
|             | Max  | 1.82  | 2.19         |
|             | Std  | 0.2   | 0.25         |
| WMed        | Mean | 1.4   | 1.2          |
|             | Min  | 0.74  | 0.7          |
|             | Max  | 1.71  | 1.67         |
|             | Std  | 0.15  | 0.2          |
| EMed        | Mean | 1.68  | 1.57         |
|             | Min  | 0.8   | 1.19         |
|             | Max  | 1.82  | 2.19         |
|             | Std  | 0.12  | 0.18         |

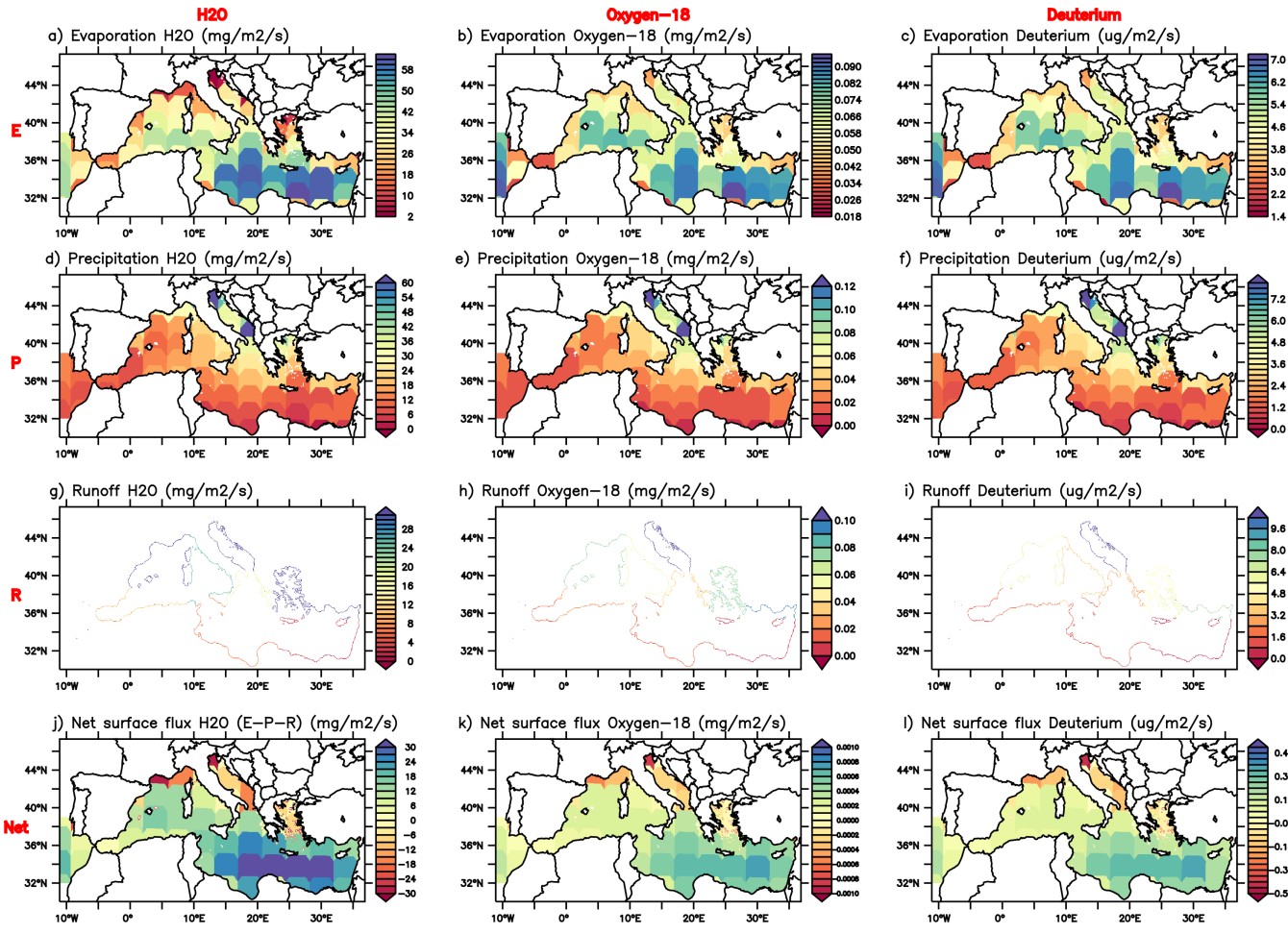

**Figure 1.** Boundary conditions and input (evaporation and precipitation) maps applied to NEMO that originate from the LMDZ-iso atmospheric model (Risi et al., 2010b).**a)** Evaporation, **b)** Precipitation, **c)** River runoff, **J)** Net surface flux (E - P - R) for $H_2O$, **(b, e, h, k)** the same but for $\delta^{18}O_w$, **(c, f, i, l)** for $\delta D_w$. The isotopic composition of river runoff is not available from the LMDZ-iso model: this flux is computed as $^{18}RP \times R$ where R is prepared from the data ofLudwig et al. (2009) and Vörösmarty et al. (1996) and $^{18}RP$ is the isotopic ratio in precipitations at the same time and location

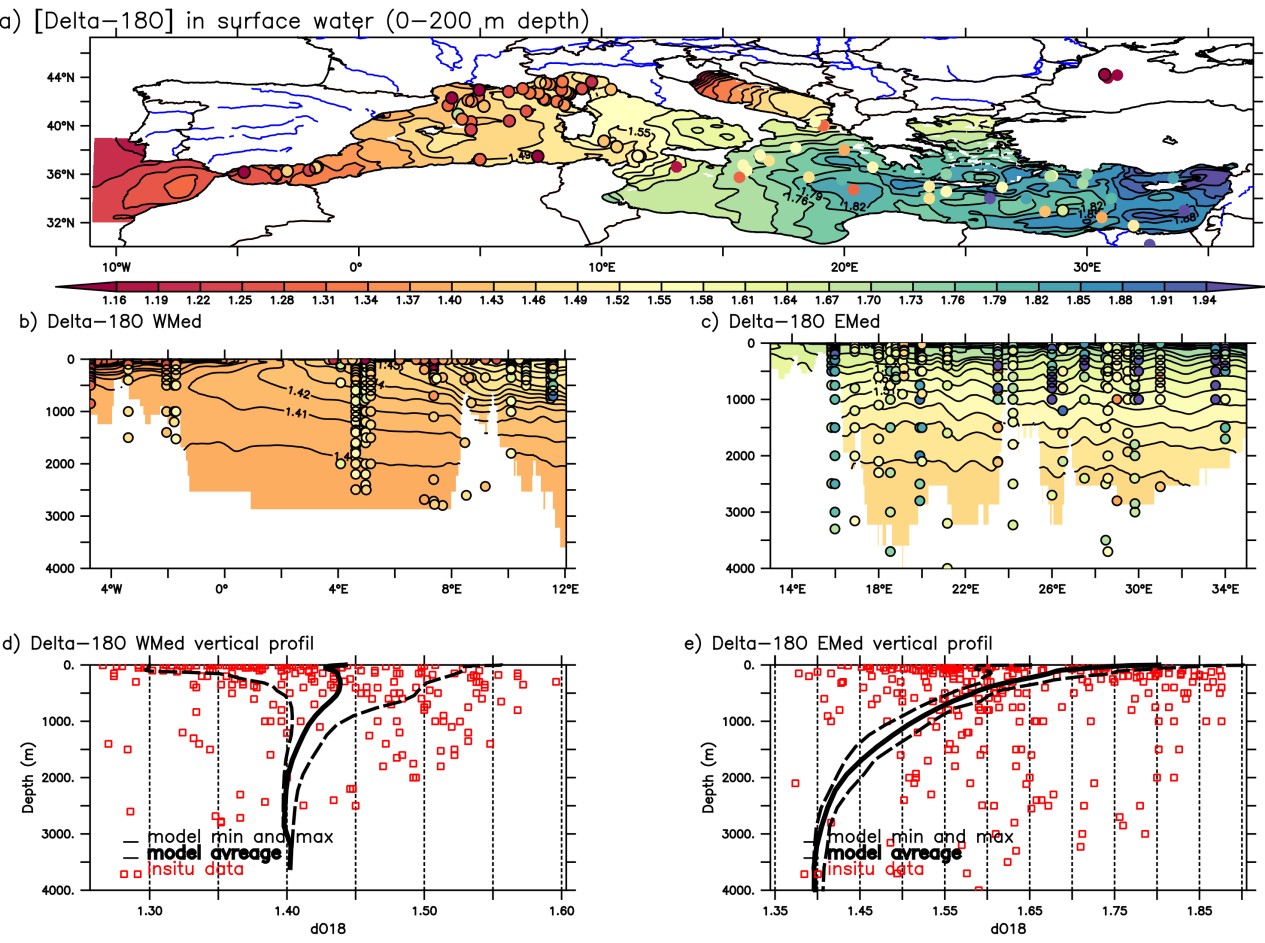

**Figure 2.** The model outputs against in-situ data for the present-day situation. **a)** $\delta^{18}O_w$ (in ‰) distribution in the surface water (50 m depth). **b)** E-W vertical section of $\delta^{18}O_w$ (in ‰) in the western Mediterranean basin d) Zonal mean comparison of $\delta^{18}O_w$ (in ‰) average vertical profiles in the western basin presenting model results against in-situ data. **c)** and **e)** the same as **b)** and **d)** but for the eastern basin. Colour-filled dots represent in-situ observations from (Epstein and Mayeda, 1953; Stahl and Rinow, 1973; Pierre et al., 1986; Gat et al., 1996; Pierre, 1999; Voelker, 2017; Reverdin et al., 2022). Both model and in-situ data use the same color scale.

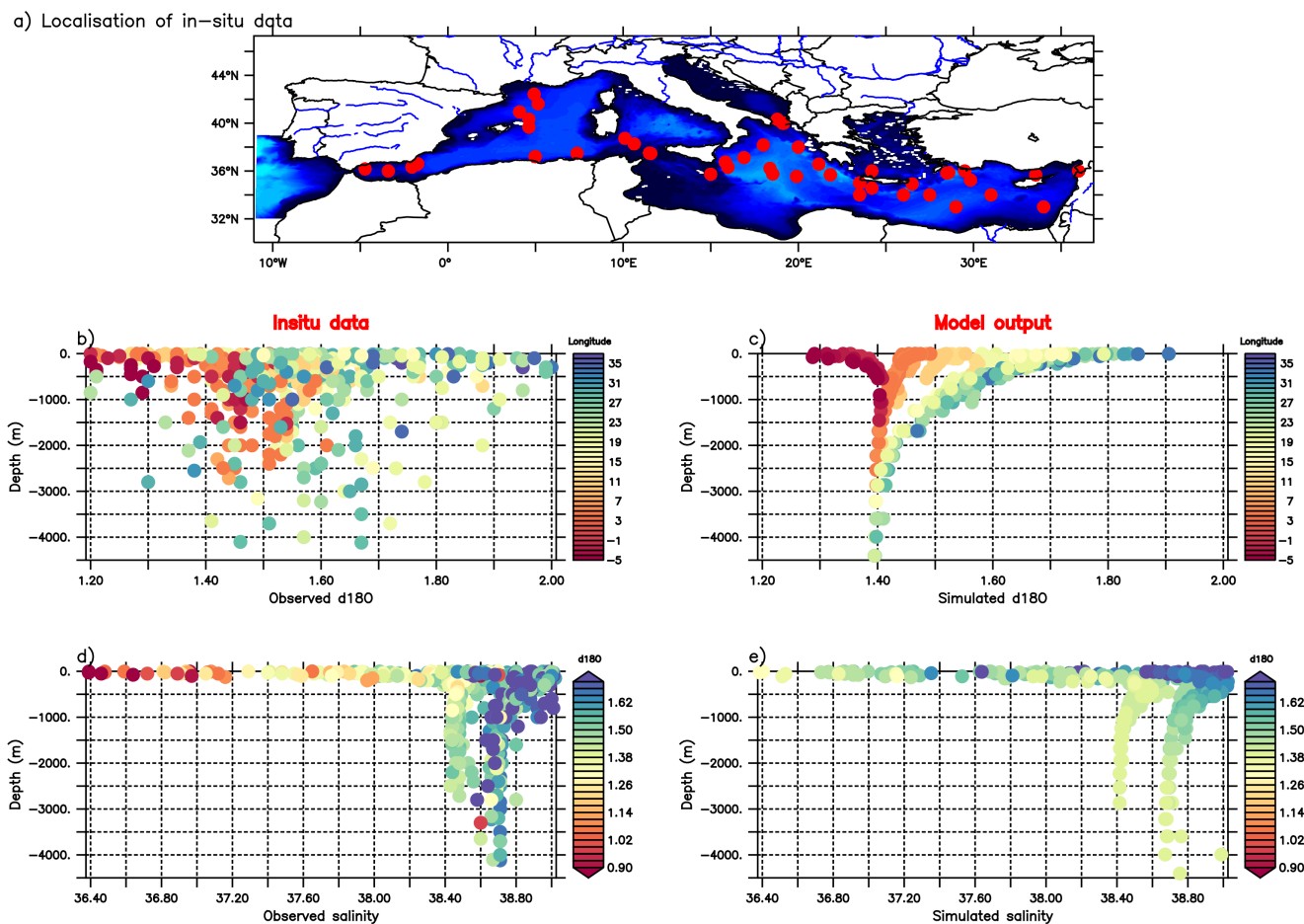

**Figure 3. a)** Location map of all stations of in-situ data (Epstein and Mayeda, 1953; Stahl and Rinow, 1973; Pierre et al., 1986; Gat et al., 1996; Pierre, 1999). **b)** depth profiles of the $\delta^{18}O_w$ (in ‰) from in-situ data (the color code indicates the latitudes of the data in °E). **c)** The same as in **b)** but from the model output. **d)** Depth profiles of salinity from in-situ observations (the color code indicates the $\delta^{18}O_w$ for each data station). **e)** The same as in **d)** but from the model output.

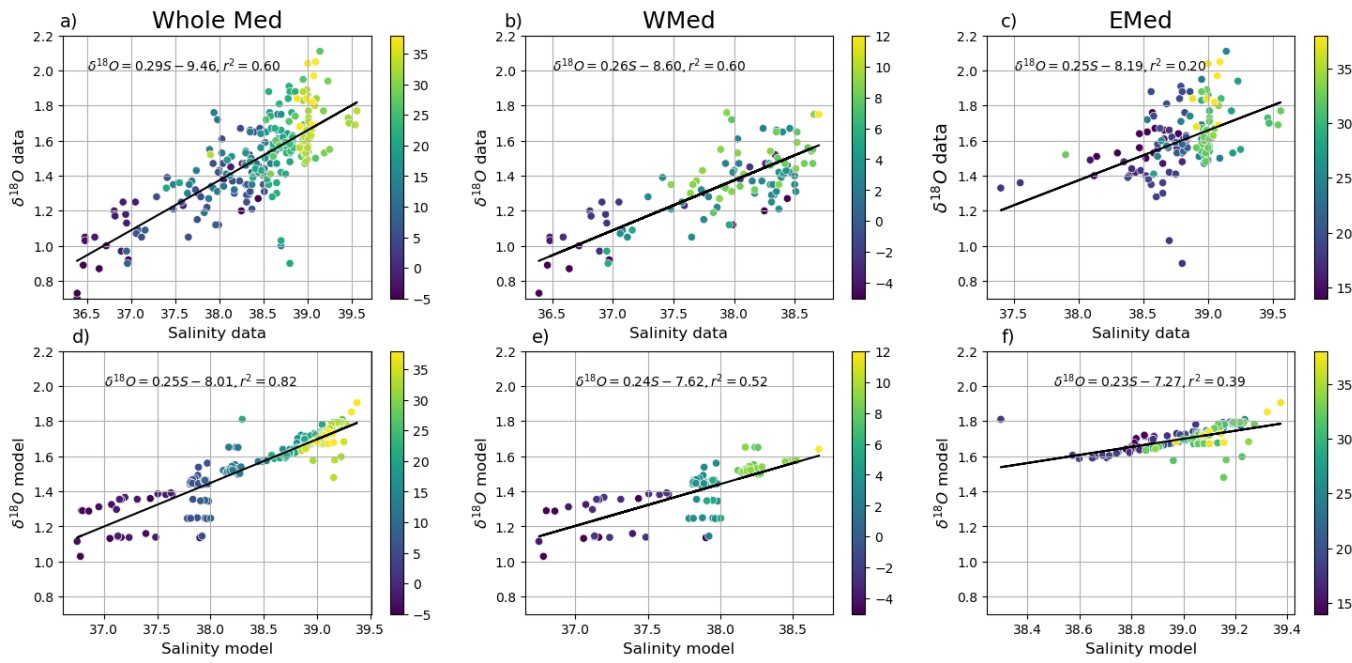

**Figure 4.** $\delta^{18}O_w$–salinity relationship in the surface water (Average 0-200 m depth of the whole basin (left column), western basin (middle column), and of the eastern Mediterranean basin (right column) calculated from available in-situ data (Epstein and Mayeda, 1953; Stahl and Rinow, 1973; Pierre et al., 1986; Gat et al., 1996; Pierre, 1999) in the upper panel (in a, b, c), and from model outputs extracted in the same positions of in-situ data (in d, e, f). The color code indicates the latitudes of the data in °E.

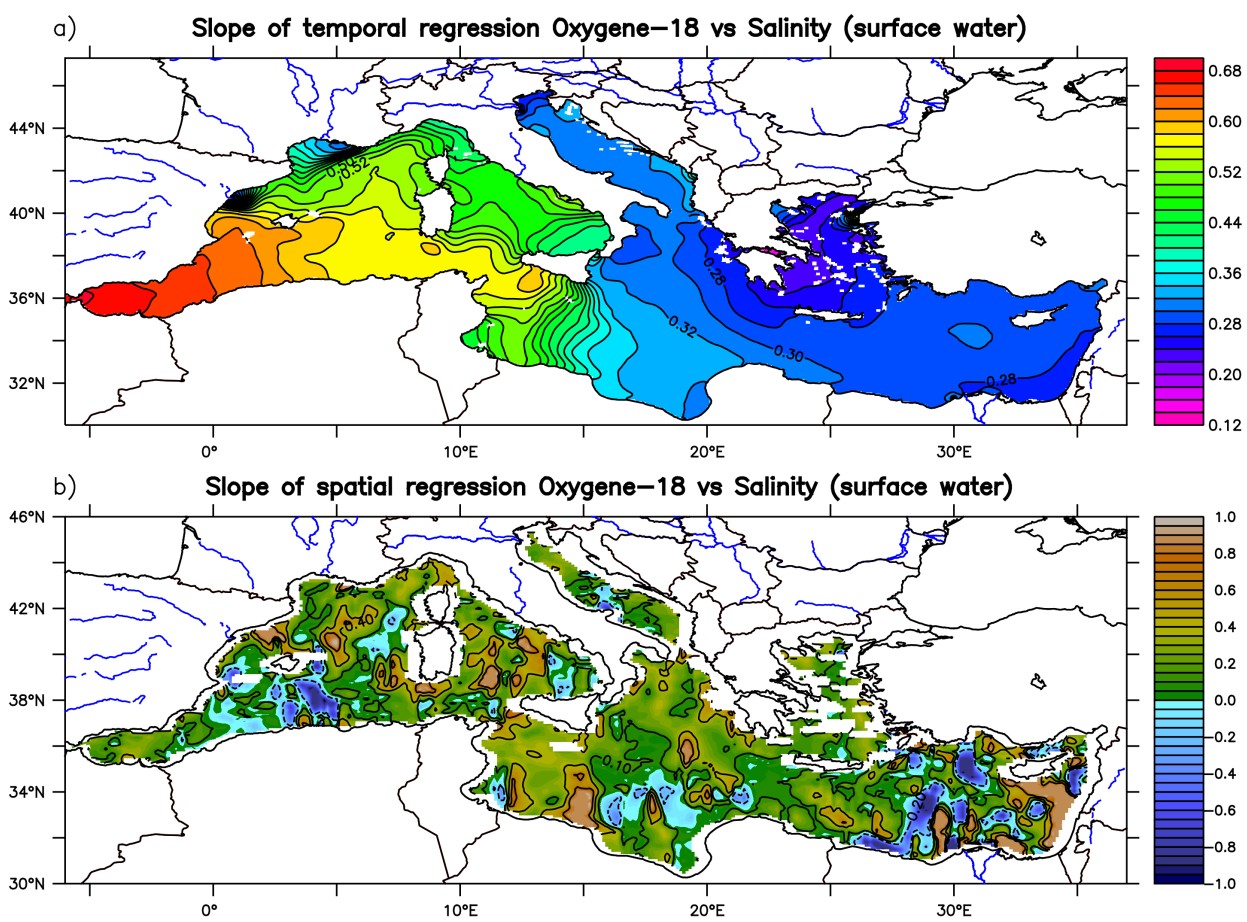

**Figure 5.** a) Horizontal map of the slope of temporal regression between the $\delta^{18}O_w$–salinity in the surface water computed using simulated last 30 years climatology, b) Spatial $\delta^{18}O_w$-salinity slope from the model outputs calculated for each grid point using simulated surface values from the 12 surrounding grid points. The non-significant zones for the regression at 95% level are marked with a black cross.

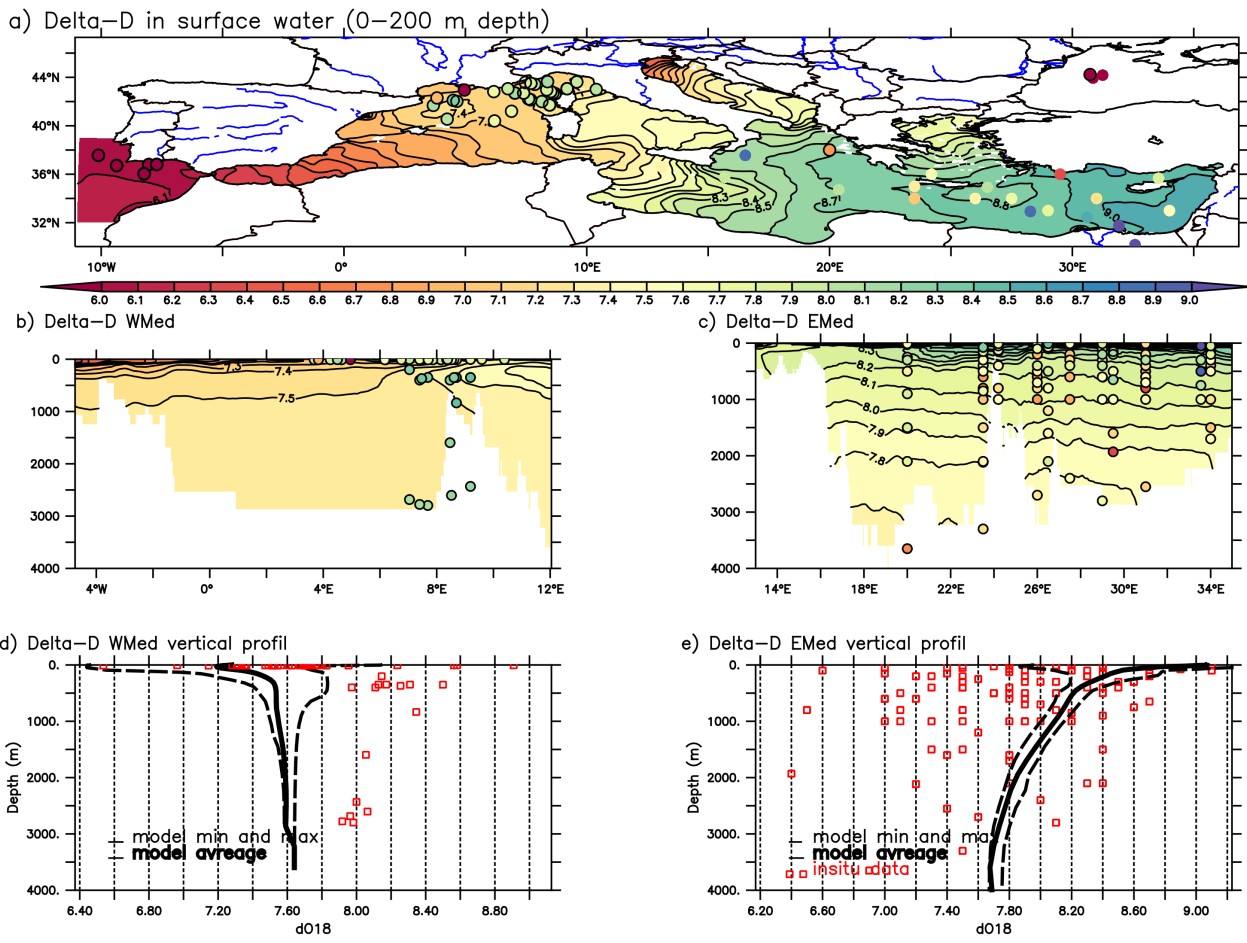

**Figure 6.** The same as in Fig. 2 but for Deuterium isotope (in ‰). In-situ data from Gat et al. (1996) and from Reverdin et al. (2022)

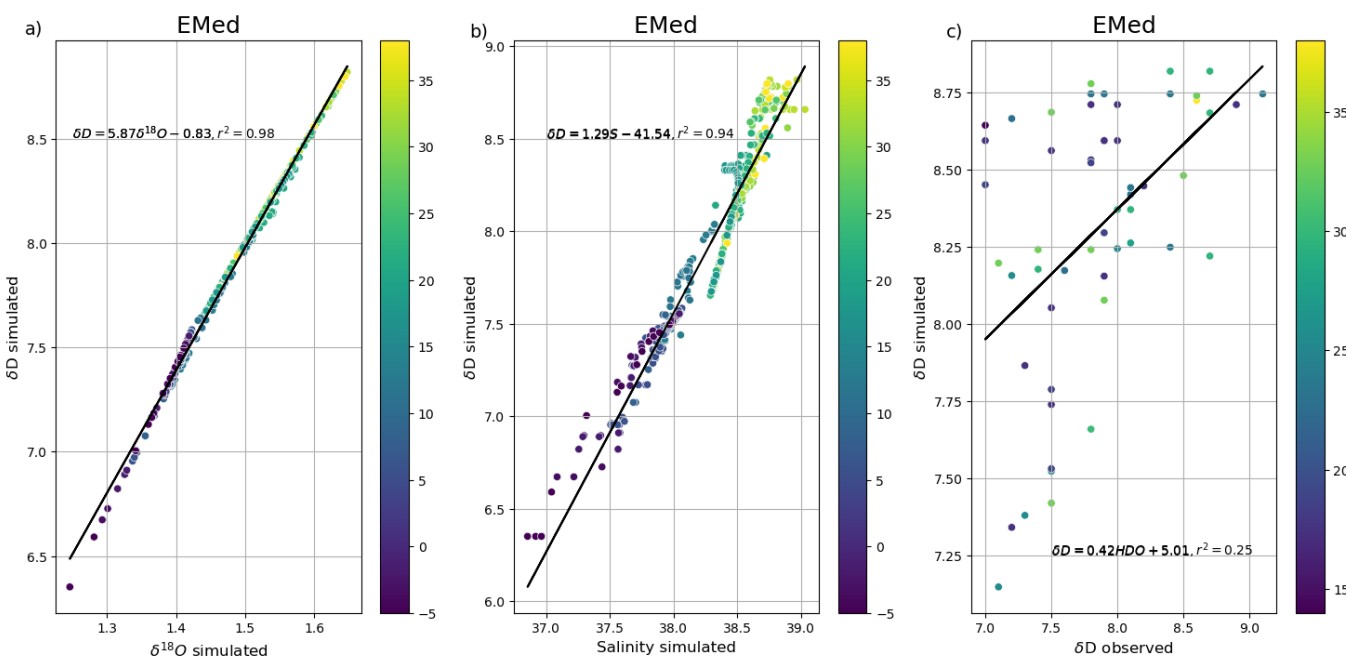

**Figure 7.** a) Multi-scatter plot of simulated $\delta^{18}O_w$ (averaged over the last 30 years of the simulation) versus simulated $\delta D_w$ at the same location of in-situ data in the eastern basin (left panel), b) simulated salinity versus simulated $\delta D_w$ (middle panel), and observed $\delta D_w$ (from Gat et al. (1996) against simulated $\delta D_w$ in c) (right panel). The color code indicates the latitudes of the data in °E.

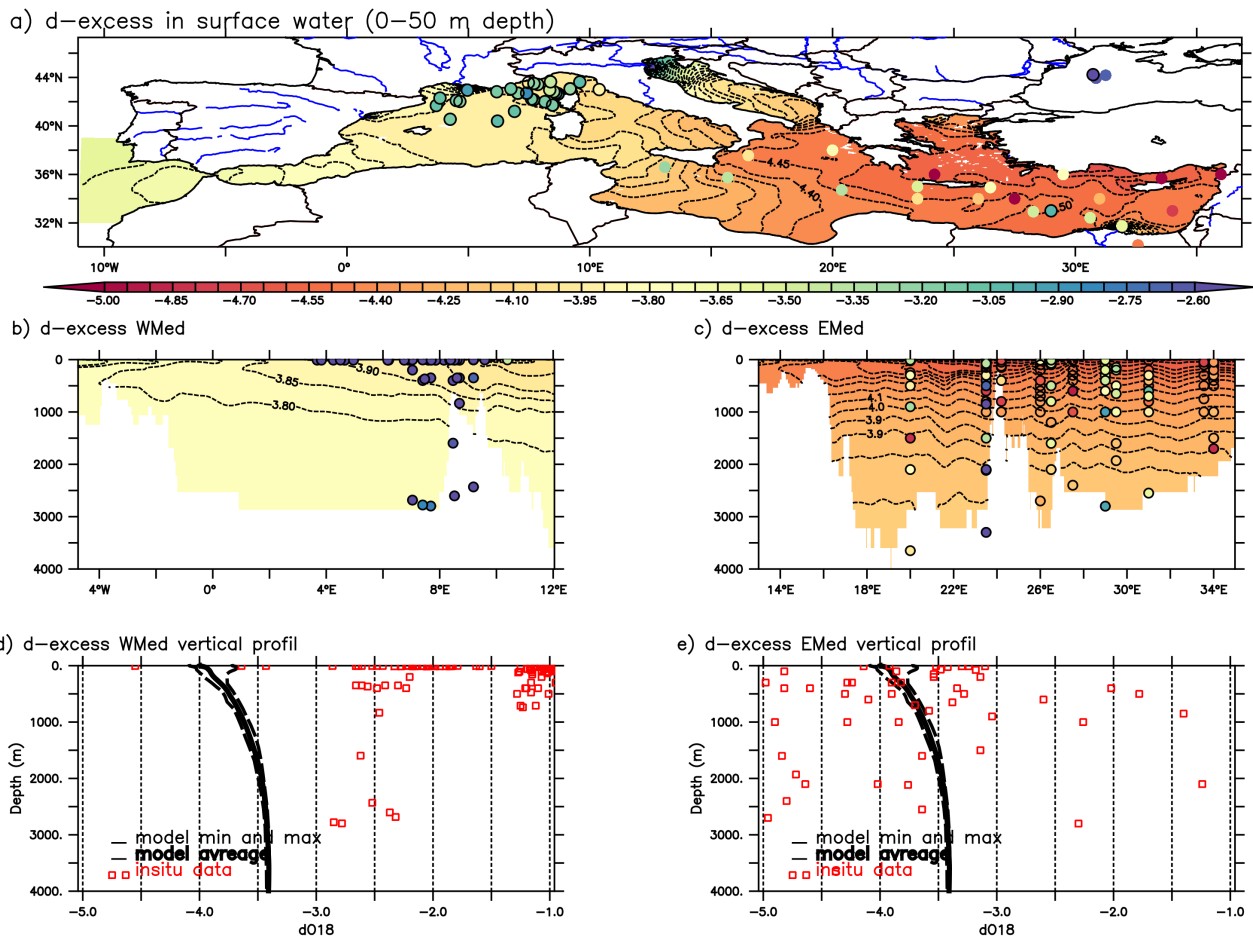

**Figure 8.** Horizontal map of d-excess in the surface water defined as deuterium excess (d-excess= $\delta D_w - 8* \delta^{18}O_w$, Dansgaard (1964)). Color-filled dots represent in-situ observations from (Gat et al., 1996; Reverdin et al., 2022). Both model and in-situ data use the same color scale.

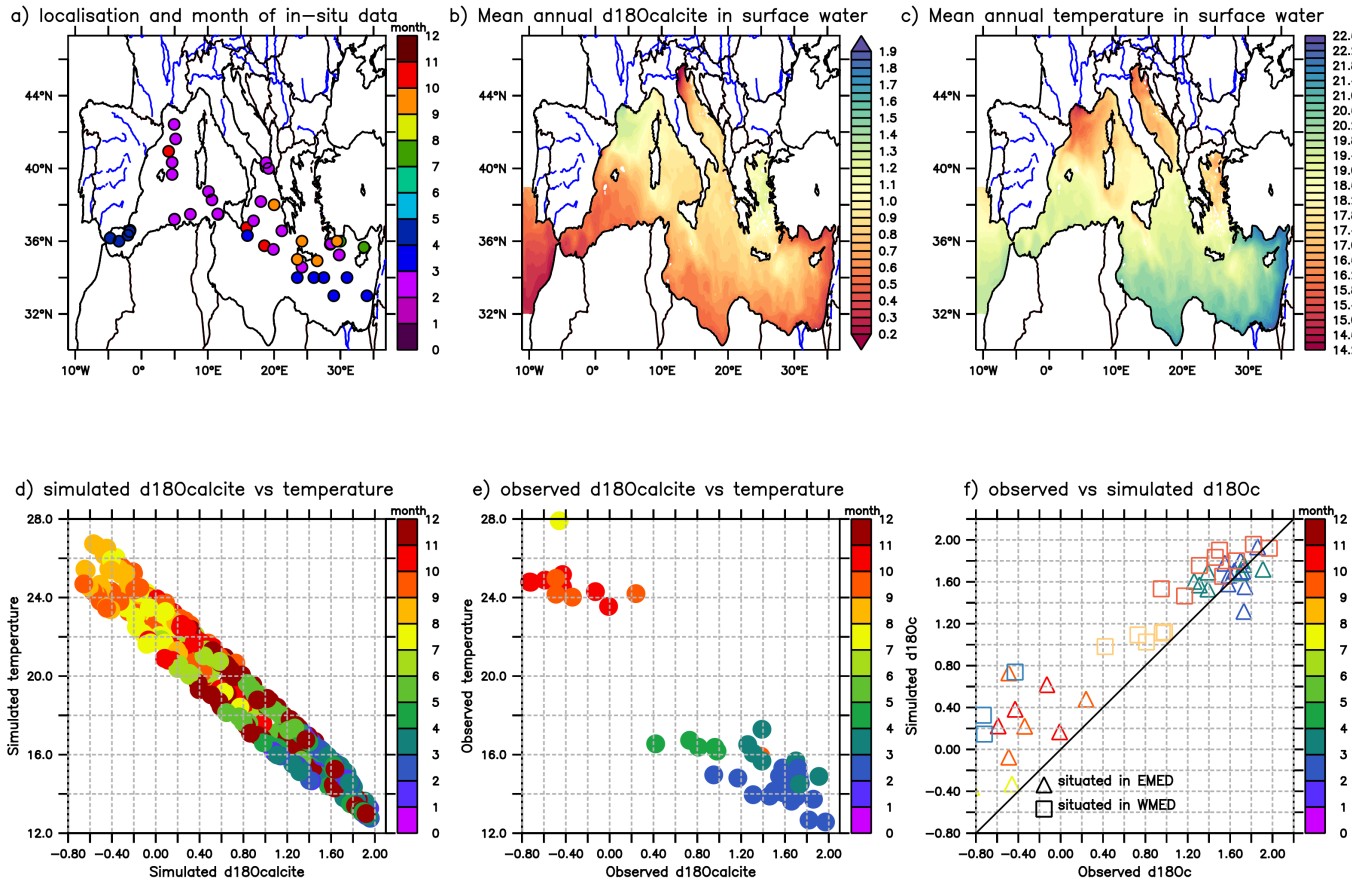

**Figure 9.** a) Localisation and the month of available in-situ data. b) Annual mean $\delta^{18}O_c$ (in ‰) distribution in calcite (surface layer 0-100 m depth), calculated using the method of Bemis et al. (1998). c) Horizontal maps of surface mean annual temperature in °C. d) Multi-scatter plots of simulated $\delta^{18}O_c$ against simulated temperature in the same location; the colour code shows months. e) the same as d) but from in-situ data. f) Comparison of the simulated and observed $\delta^{18}O_c$ (in ‰) in the surface layer averaged in the two basins (boxes from the WMed and triangles from the EMed dashed line model). The color code shows months.

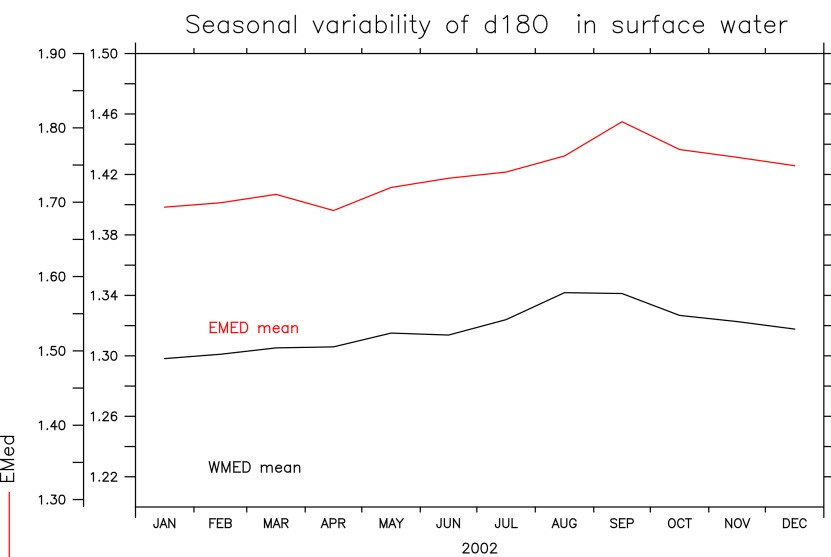

**Figure A1.** Seasonal variation of $\delta^{18}O_w$ (in ‰) in eastern and western basin surface waters

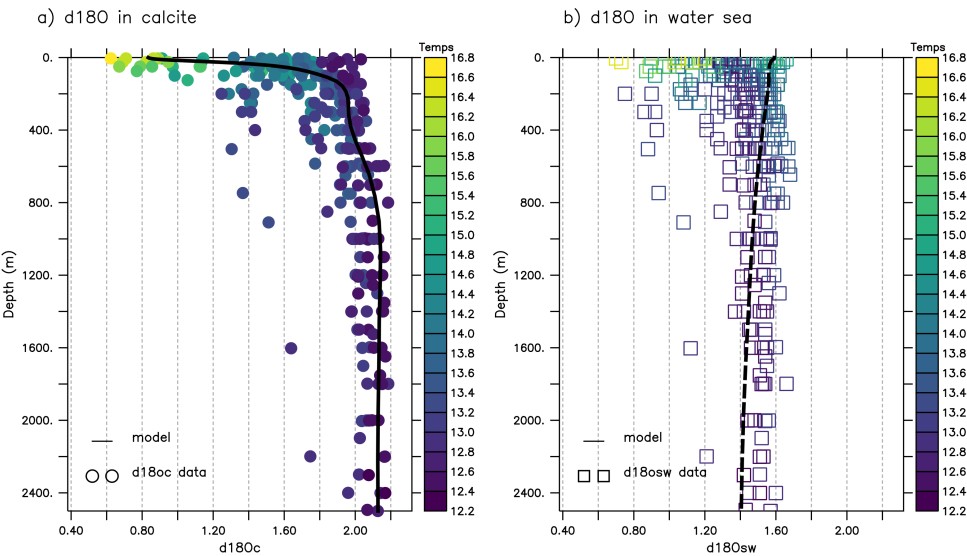

**Figure A2.** a) Comparison of the simulated average vertical profiles of $\delta^{18}O_c$ (in ‰) (circle data and line from model outputs). b) same as a) but for $\delta^{18}O_w$

## Appendix B: Sensitivity to temperature employed for computing $\delta^{18}O_c$

The forcing of surface temperature used in the calculation of $\delta^{18}O_c$ does not come from LMDZ-iso but from an ERA-40 relaxation term applied to the ARPERA heat flux. This is certainly among the limitations of the OFFLINE coupling mode with the use of a pre-calculated dynamical field.

We presented horizontal temperature maps used in calculating $\delta^{18}O_c$ (refer to Fig. B1c). We judged this simulation to produce reasonable temperature patterns in the Mediterranean Sea. A notable difference arises when comparing the $\delta^{18}O_c$

calculated with high-resolution simulated temperatures (cf. Fig. B1a and B1b below) to that derived from a global model using temperature data from LMDZ (cf. Fig. B1c and B1d). The global model shows a significant bias in $\delta^{18}O_c$ as a consequence of low temperatures simulated in the Mediterranean Sea (cf. Fig. B1c and B1d).

Additionally, in this simulation, we employed the same freshwater forcing (from Ludwig et al. (2009), and the RivDis dataset, Vörösmarty et al. (1996)) as that used in the dynamical simulation (in Beuvier et al. (2012a)) where the temperature

was simulated, ensuring complete consistency between freshwater flux and temperature. This validates our choice to utilize temperatures simulated by the MED12 model and forced by ERA5 rather than LMDZiso (see the figure below). However, this inconsistency requires further investigation within a fully coupled ocean-atmosphere model to ensure consistent simulation of changes across various model components.

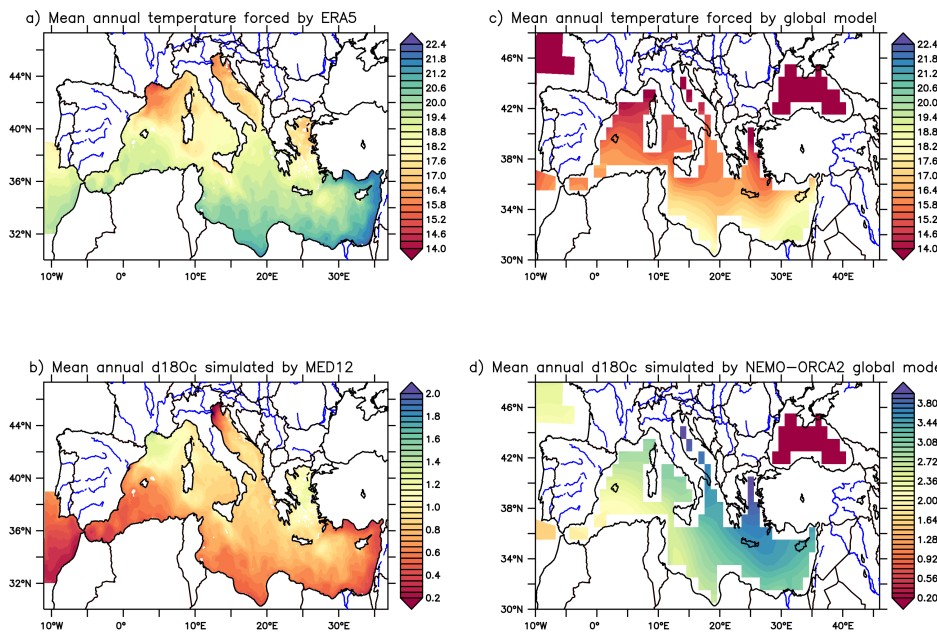

**Figure B1.** Comparing the $\delta^{18}O_c$ calculated with high-resolution simulated temperature (a and b) to that derived from a global model using temperature data from LMDZ (c and d)

**Appendix C: Assessing the impact of changing the resolution of atmospheric and oceanic models**

Sensitivity tests were conducted on the results by altering the horizontal resolution of LMDZ-iso from R96 to R144. A significant correlation was obtained: $r^2$ values are 0.66 and 0.68 for the whole basin with the LMDZ-iso "96" and "R144", respectively. This shows that the $\delta^{18}O_w$ distribution is globally well simulated by the model. However, neither LMDZ-iso "96" nor "R144" reproduce the highest values of $\delta^{18}O_w$ observed in the Mediterranean Sea (up to 2.4 ‰ as measured by Gat et al. (1996)). Hence, the results of the model are very close between these two horizontal resolutions (R96 and R144) of the

LMDZ-iso atmospheric model (Fig.C1), so, there may be a certain threshold of spatial resolution below which the simulation is improved by a finer resolution. Vadsaria et al. (2020) demonstrated that high resolution ( 30 km of the atmospheric model) is critical to accurately capture the synoptic variability needed to initiate the formation of the intermediate and deep waters of the Mediterranean thermohaline circulation (Li et al., 2006). Therefore, we chose to use the R96 resolution, which is the least expensive. The figure below displays the $\delta^{18}O_w$ anomaly map between the two simulations R144 and R96. The simulations

show a very small difference, ranging between -0.2 and +0.2 ‰. One possible reason for this slight variation is the use of runoff forcing. As described in Section 2.2 of the paper, the runoff forcing is based on data from Ludwig et al. (2009) instead of ORCHIDEEiso. This is because the water flows simulated by ORCHIDEEiso are unrealistic in the Mediterranean basin. For example, ORCHIDEEiso significantly overestimates the Nile River discharge. In summary, the change of horizontal resolution between R144 and R96 is not sufficient to generate drastic changes in evaporation and precipitation (as suggested by Vadsaria

et al. (2020)), and also the fact that the same runoff forcing was used in both the R96 and R144 simulations explain the small difference between these two simulations despite the change in model resolution. Nonetheless, a dedicated study should be conducted to further elucidate the resolution impact on the tracer distribution.

The model's high resolution presents a unique opportunity to represent a realistic thermohaline circulation in the Mediterranean basin, thus enabling a better understanding of the processes governing water isotopic distribution within this inter-

continental basin. Figure C2 illustrates a comparison between the results of the global model (ORCA2, with a 2° horizontal resolution) and the NEMO-MED12 model, both employing the same water isotopes modeling approach and driven by the identical atmospheric model, LMDZiso (at R96 resolution). The comparison reveals that the global model produces unrealistically high values of $\delta^{18}O_w$ in the Mediterranean Sea, especially in the eastern basin ($\delta^{18}O_w > 2$ ‰, maximum 3.3 ‰), whereas in-situ data show maximum values of around 2.1 ‰ (Gat et al., 1996). Overall, high-resolution models can bridge the

gap between the coarse resolution of global climate models and the regional-to-local scales. They can provide a more realistic representation of physical processes and climate feedback compared to global climate models. This is particularly true for the Mediterranean region with complex geology. In particular, atmospheric circulation (high wind gusts in winter) and oceanic circulation (deep convection) are better represented in regional models (Ludwig et al., 2009).

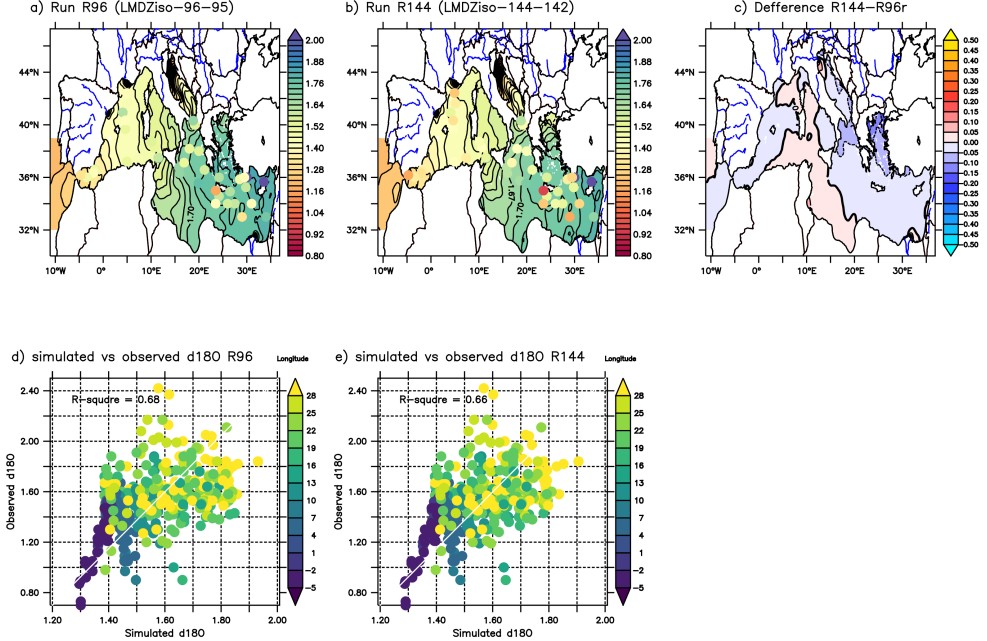

**Figure C1.** "Distribution of $\delta^{18}O_w$ (in ‰) in surface water (at a depth of 50 m) from the R96 simulation. Colored dots represent in-situ observations compiled from Epstein and Mayeda (1953), Stahl and Rinow (1973), Pierre et al. (1986), Gat et al. (1996), and Pierre (1999). Panel d) presents a multi-scatter plot comparing simulated $\delta^{18}O_w$ (averaged over the last 30 years of the simulation) from the R96 simulation with in-situ data from the mentioned sources across the entire basin. The color code indicates the latitudes of the data in °E. Panels b) and e) depict the same as panels a) and d), respectively, but from the R144 simulation. Panel c) illustrates the $\delta^{18}O_w$ anomaly map between the R144 and R96 simulations in surface water."

## Appendix D: Pseudo-Salinity against standard simulated Salinity

The water fluxes from the stand-alone (non-coupled) experiments with LMDZiso are not identical to those constraining NEMO-Med12. Hence $\delta^{18}O_w$ or $\delta D_w$ computed with the water fluxes obtained with LMDZiso would not be consistent with the salinity predicted by NEMO-Med12. For this reason, we compute a "pseudo salinity" Sw (Delaygue et al., 2000; Roche et al., 2004). This additional passive tracer does not affect the ocean dynamics. Its sole purpose is to allow a coherent assessment of the relation of the isotopic fields predicted by the model with salinity since they are computed with the same fresh-water forcing.

The evolution equation for Sw is given by:

$$\rho K \bigtriangledown \mathcal{S}_w = (\mathcal{E} - \mathcal{P} - \mathcal{R} + |\mathcal{I}|)\mathcal{S}_w - (S_w \mathcal{I}) \tag{D1}$$

Let $\mathcal{E}, \mathcal{P}, \mathcal{R}$ represent evaporation, precipitation, and run-off, respectively, $S_w$ is the salinity of water, $\mathcal{I}$ is the net freshwater flux associated to sea-ice formation. With the further assumption that the salinity associated with evaporation, precipitation,

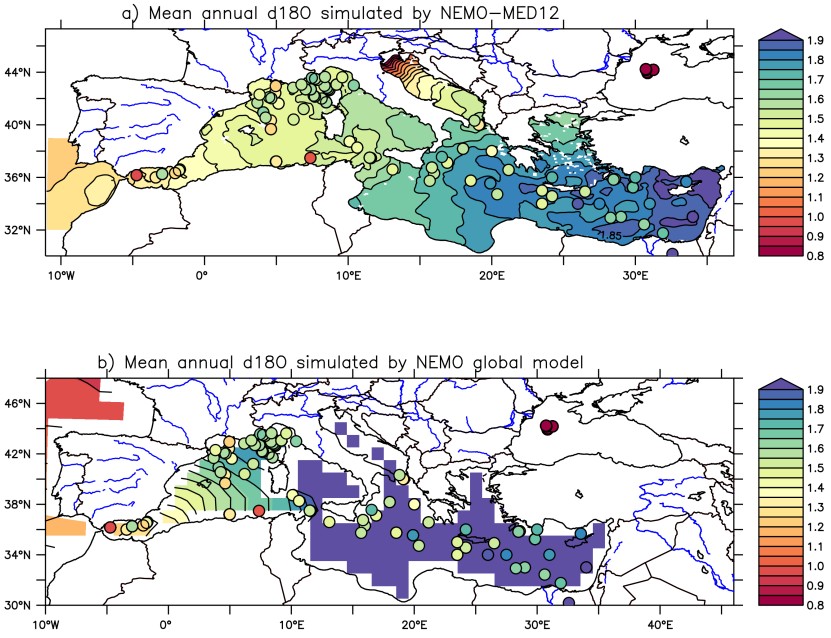

**Figure C2.** Comparison between the δ18Ow results of the global model (ORCA2 2° of resolution) and NEMO-MED12 model, using the same water isotope modeling approach and forced by the same atmospheric model LMDZiso (R96)

and run-off is zero (no effect of freezing/melting on the concentration/dilution of pseudo-salinity in the Mediterranean Sea), the boundary condition for salinity reads.

$$\rho K \bigtriangledown \mathcal{S}_w = (\mathcal{E} - \mathcal{P} - \mathcal{R})\mathcal{S}_w \tag{D2}$$

The basic understanding of these atmospheric fluxes, is that evaporation tends to increase the surface salinity, and the 18O/16O ratio, in contrast to precipitation and runoff.

In Fig. D1 below, we have plotted the anomaly in salinity-pseudo-salinity to assess the correspondence between pseudo-salinity results and standard modeled salinity. The well-known east-west gradient is effectively captured by recalculated pseudo-salinity, showing very similar values to those of standard salinity. Minor deviations are noticed in the Gulf of Lions and the Algerian Basin, attributed to overlooked meso-activity impacts in the global LMDZiso simulation. Overall, the pseudo-salinity globally yields values highly comparable to standard simulated salinity."

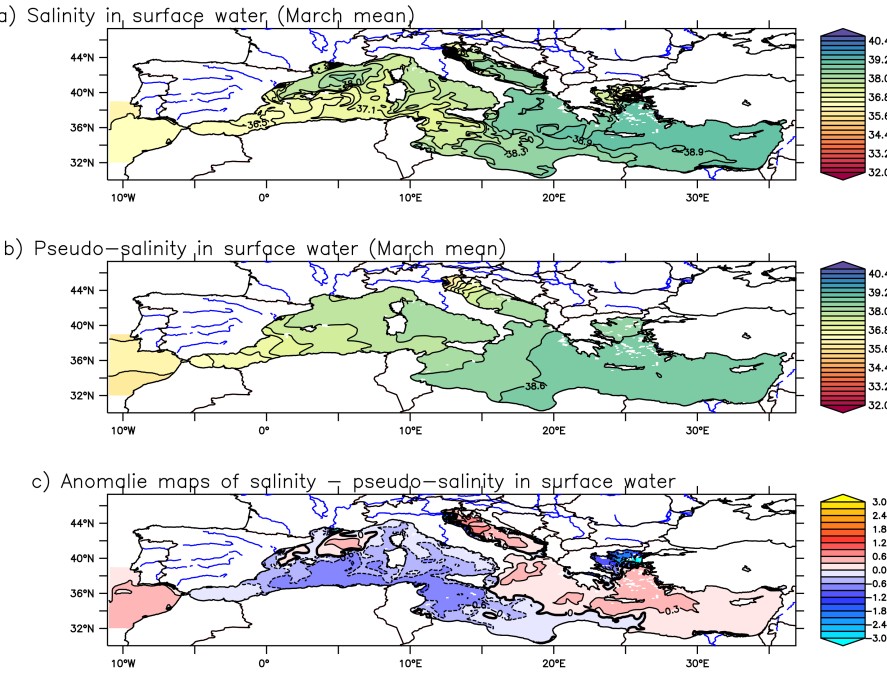

**Figure D1.** a) Standart simulated salinity from NEMO-MED12 in the surface model. b) Pseudo-salinity simulated in the surface water. c) the anomaly a) - b)

## Appendix E: Sensitivity to isotopic composition input from river runoff

The simulation of surface water isotope fluxes is carried out using the land surface model ORCHIDEE. Isotopes are incorporated into the river discharge of ORCHIDEEiso, as described by Risi et al. (2016). However, the isotopic version of ORCHIDEEiso is outdated and cannot be coupled with the current version of LMDZ-iso. A joint project is currently underway to reintroduce water isotopes in the new versions of ORCHIDEE and to couple with LMDZ-iso. Already performed LMDz-ORCHIDEE numerical experiments Risi et al. (2010b) provide monthly mean isotopic and freshwater fluxes except for

run-offs.

    We have conducted sensitivity simulations to assess the impact of computing the isotopic composition of rivers based on the isotopic composition of precipitation (as explained in the paper, see section 2.3). Two new experiments (EXP1 and EXP2) were conducted using output from an earlier version of LMDZiso coupled to ORCHIDEE-iso (cf. Risi et al., 2016) at a lower resolution of R96x71.

– EXP1: Employed the approach described in section 2.3 of the paper, where $^{18}R_{river}= {}^{18}R_{precip} \times {}^{18}R_{runoff}$

    – EXP2: Integrated the simulated $\delta^{18}O$ of rivers from the older version of LMDZiso at R96x71 resolution, where $^{18}R_{river}={}^{18}R_{riverLM}$ $\times {}^{18}R_{runoff}$.

– Here, $^{18}R_{precip}$ and $^{18}R_{riverLMDZiso}$ are derived from LMDZiso (R96x71, Risi et al., 2016), while $^{18}R_{runoff}$ is from the interannual dataset of Ludwig et al. (2009) and the RivDis dataset from Vörösmarty et al. (1996).

The results of these sensitivity simulations are shown in Fig. E1 below. In EXP1, the model reproduces a reasonable east-west gradient similar to our results using a higher version of LMDZiso (R96), as shown in Fig. 2a. In EXP2, the addition of the $\delta^{18}O$ of rivers simulated by LMDZiso reveals a more enriched isotopic composition ($^{18}O_{river}$) compared to $\delta^{18}O_{precip}$. Indeed, evaporation can enrich heavier isotopes in the remaining water, including rivers, which is particularly evident for the Po River, exhibiting a clear positive anomaly around 0.5‰ near the coast and dispersed over the Adriatic Sea. The impact of other main rivers (e.g., Rhone and Po) remains very close to the coast, rapidly dispersed by circulation.

However, the impact of the Nile significantly influences the $\delta^{18}O_w$ signal simulated in EXP2, highlighting a well-known issue in ORCHIDEEiso concerning the simulation of Nile discharge, where ORCHIDEEiso tends to largely overestimate the discharge, as depicted in the figure.E1 below.

Consequently, we opted not to utilize the global version of LMDZiso due to the complex hydrology of the Mediterranean region. Instead, we employed a combination of model outputs and in-situ data to estimate the runoffs entering the Mediterranean Sea. For the isotopic composition, we adopted the same approach used by Delaygue et al. (2000).

In conclusion, these sensitivity simulations (EXP1 and EXP2) showed an enrichment of $^{18}O_{river}$ in the rivers due to evaporation, especially for the Po. The influence of the Nile significantly affects the signals, which has prevented the use of this version of LMDZiso (R71) and we are unable to couple this old version of ORCHIDEE (outdated) with the current version of LMDZiso. Therefore, the approach of Delaygue et al. (2000) was chosen over the data for its reproducibility and usability in paleo simulations.

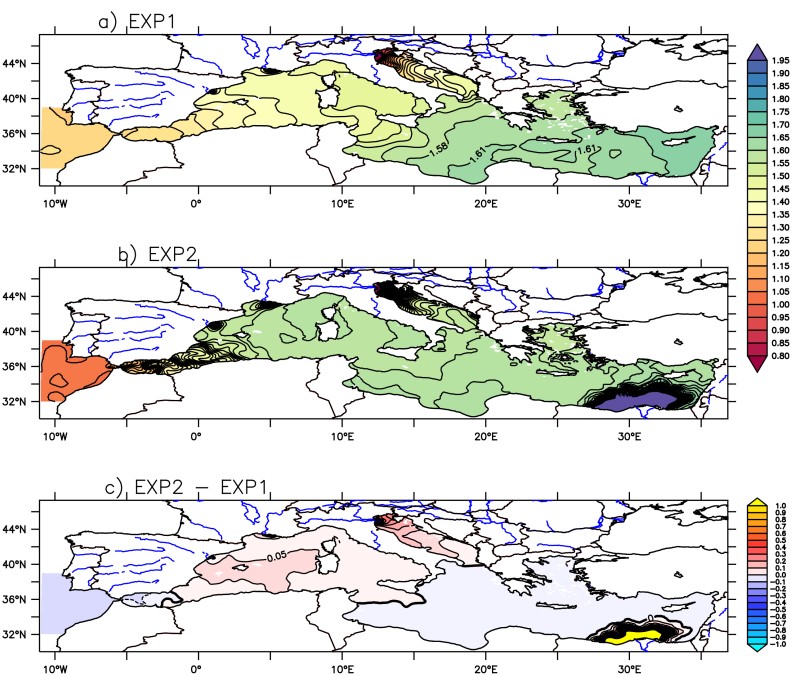

**Figure E1.** a) EXP1: we use the same approach as described in our submitted paper, i.e., $^{18}R_{river} = {}^{18}R_{precip} \times {}^{18}R_{runoff}$.b) EXP2: we added the $^{18}O_{river}$ simulated by the old version of LMDZiso at a lower resolution R96x71. $^{18}R_{river} = {}^{18}R_{riverLMDZiso} \times {}^{18}R_{runoff}$. C) the difference EXP2 – EXP1