# Peer review of "Modelling the water isotopes distribution in the Mediterranean Sea using a high-resolution oceanic model (NEMO-MED12-watiso-v1.0): Evaluation of model results against in-situ observations"

_Geoscientific Model Development, 2023_

## Referee Comment (RC3)

Ayache et al NEMO iso review

Water isotope tracers are indeed a useful way to track the water cycle, and this study seeks to provide for high resolution insight into the Mediterranean ocean.

The authors of this study include expert isotope modelers, so the work is on the whole very solid.

I have Mostly few questions about the specifics of implementation and the write up.

1) for someone who is *not* a water isotope modeler, the casual inclusion of shorthand / jargon without explanation needs to be expressly defined. I.E., $\delta^{18}O_{sw}$ or $\delta D$ (also— shouldn't you write $\delta D_{sw}$ to be consistent?) or $CaCO_3$ or $\delta^{18}O_c$ all need to be defined – what does the delta mean. What do the subscripts mean. Some of the equation rendering has broken down maybe on the author's side, maybe on the Copernicus side.

2) In the write up of previous work in the med for isotopes, the authors may (not?) be aware that there is almost certainly a mistake in the $\delta D$ values of Gat as they vary much much less than $\delta^{18}O_{sw}$ – probably the original source should be sought out for that validation.

3) When it is said that 'we use fluxes' from LMDZiso – that is surface water isotope fluxes? How are fluxes from rivers handled? Do you use observed isotope values or simulated ones? (Do the simulated river values closely approximate the measured ones?) If no measurements are available, what was done instead?

4) On page 5, they say "it is common to transport the isotopic ratio rather than the individual isotope..." then later "and pseudo-salinity fluxes". I don't know NEMO that well, but I am going to guess they are saying in a round about way that this ocean model has a rigid lid instead of a free surface. They should say either way. Because most isotope models do *not* in fact transport around concentrations of isotopes, they transport around mass. Sure – some models do not actually conserve mass – they are forever having to reimplement water isotopes in their code because they have virtual moisture or salt fluxes. Anyhow, those who *can* do indeed transport around mass not concentration. The per mil isotopic composition is determined on post-processing. Why? This is done so that the isotope / tracer code can have an exact replica of 'water' from the non-tracer code and this tracer can be 1:1 compared throughout the entire model to made sure mass isn't being gained/lost anywhere spuriously. Isotopic composition comes into play because SMOW is defined and fractionation at phase changes is defined. This is, in general, simpler for an ocean model where the mass of water is simply (MO – S), but if you have a rigid lid, then you have virtual mass fluxes of isotopes. Clarity for this point is required.

5) The 'interpolated to 20 min time step'—does this mean that actual rainfall and weather systems otherwise are regressed and then passed to the model at this finer time step, or is the daily value simply applied/scaled at the 20 minute interval. I would guess that if you are using some sort of nudged version of LMDZiso that there is useful information at a finer timescale (i.e., if its been nudged at 3 hour timesteps,

why not interpolate from 3hr->20min) – otherwise you'll miss the finer temporal resolution features. You wouldn't need to store *all* of LMDZiso values at that timestep—just those in your domain.

6) I'm still confused about the pseudo-salinity tracer. Please explain
7) Page 6:: the present day values seem awfully low. CO2 of 348ppm – I rarely encounter PhD students anymore born in a world with CO2 this low.
8) NEMO-MED12 grid is jargon that I don't understand
9) Still confused on L165-170 how the isotopic composition for the rivers was determined. It sounds like you are saying that the isotopic composition of river discharge = local grid box precipitation isotopic composition (which would be wrong of course). Can't you use observations *or* use d18Oriver from LMDZiso (or another isotope enabled model).  Since you have already established that the Med is an evaporative basin, you might expect that d18Oriver to be a bit enriched compared to d18Oprec… (Places downriver or downhill in a P>E location you would expect d18Oriver  to be a bit depleted compared to d18Oprec… ) But the Med, and particular places like the Nile, you definitely should expect some evaporation to strip out the light isotopes of the river.

**H2O18 in River Outflow – H2O18 in Precipitation**

[Figure]

10) Can you write up the E-W surface d18Osw context from obs ? Maybe putting observed d18Oriver would make for a better gradient. (The baseline composition is set by your SMOW definition—I'd worry less about that.)
11) For deriving d18O-S relationships – can you put yours in context of the LMDZiso? Would you expect NEMOiso to differ that much given that you are prescribing your end member from the coupled model? Is this a useful section?

12) For section 3.3 – can you please check the Gat96 comparison. Does it make sense?
13) For the d18Ocalcite discussion, what is the correlation between d18Oc and temperature temporally and spatially. For interannual variability, does the inclusion of d18Osw confound the correlation. Also—you are presuming surface dwelling foraminifera. Maybe its interesting to look at species specific d18Oc.
14) There are *some* existing SWING comparisons of different isotopic compositions for different groups. Maybe for your next paper you could pull those in, but for this one, you should at least mention and speculate if it would be useful.

---

## Author Comment (AC1)

**Manuscript " Modelling the water isotopes distribution in the Mediterranean Sea using a high-resolution oceanic model (NEMO-MED12-watiso-v1.0): Evaluation of model results against in-situ observations"**

Mohamed Ayache[1], Jean-Claude Dutay[1], Anne Mouchet[2], Kazuyo Tachikawa[3], Camille Risi[4], and Gilles Ramstein[1]

[1]Laboratoire des Sciences du Climat et de l'Environnement, CEA-CNRS-Université Paris Saclay, 91191, Gif-sur-Yvette, France
[2]Freshwater and OCeanic science Unit of reSearch (FOCUS), Université de Liège, B-4000 Liège
[3]Aix Marseille Univ, CNRS, IRD, INRAE, Coll France, CEREGE, 13545, Aix-en-Provence, France
[4]Laboratoire de Météorologie Dynamique, IPSL, CNRS, Sorbonne Université, Paris, France Correspondence: Mohamed Ayache (mohamed.ayache@lsce.ipsl.fr)

**Reply to reviewers' comments**

Dear Pr. I., Andrew Yool

We would like to thank you for providing us the opportunity to revise our manuscript, and we are extremely grateful to Pr. Antje Voelker, Pr. Allegra N. LeGrande and the anonymous reviewer for their careful reading and comments that helped to improve our manuscript significantly.

We have revised our manuscript and provided a detailed response to each reviewer's comment and request below.

**Color code**
Reviewer comments
Authors response
The modifications performed in the manuscript appear in red above and in the revised manuscript with Changes Marked.

**#1: Review by Antje Voelker:**

Ayache and co-authors present the first high-resolution modeling study for water isotopes in the Mediterranean Sea. As a first attempt to relate their results to future paleoceanographic applications, they apply their water isotope model outcomes to calculate $\partial^{18}O$ in marine carbonate. Overall, this is an interesting and novel study and, in my opinion, fits well into GMD. As someone working with water isotopes in sea water, I am very happy to see such studies advancing our knowledge. The manuscript is well written and the figures all informative and needed. The results are relevant and future attempts to go towards a fully coupled ocean-atmosphere model should be of great interest for the scientific communities interpreting speleothem and lacustrine paleo-records in the Mediterranean region.

The science presented is sound, although I am not an expert in climate models and therefore cannot fully judge if the model description is sufficient and can be reproduced based on the information given. From my reading I would say both criteria are sufficiently fulfilled.

I do not have major comments for the manuscript and believe minor revision will address the points I am making below. Some relevant changes might arise from the additional in-situ data I am pointing out in the specific comments, but those will not change the overall outcome of the study. One caveat I see in the manuscript is that Nile river run-off is never mentioned and discussed. For the sapropel research (mentioned in the manuscript) and tracing influences of NW African monsoon rainfall in paleoclimate studies, but also in the modern hydrological cycle that is an important process.

We thank Pr. Antje Voelker for the summary of our paper, and the positive assessment of its significance. Historically, the Nile played a crucial role in freshening surface water during sapropel events. However, following the construction of the Aswan High Dam in 1965, its influence has decreased (ElElla, 1993; Nixon, 2003). As a result, the Nile is no longer a primary factor contributing to the present-day state of the Mediterranean Sea.

Therefore, in the revised manuscript, we have included the following sentences to clarify this point.

"The Nile played a crucial role in freshening surface water during sapropel events. However, since the construction of the Aswan High Dam in 1965, its influence has decreased (ElElla, 1993; Nixon, 2003). As a result, the Nile is no longer a major contributor to the current state of the Mediterranean Sea."

(see section 2.3, lines 186-189 in the track changes version).

Moving forward, we will address each of the reviewer's comments in detail.

**Specific comments:**
Line 10: as a paleoceanographer I understand where you want to go with the phrase "CaCO$_3$ shell" but not every reader will be aware that you referring to planktonic foraminifera shells here. So, the text needs to be amended here to be understandable for every reader.

Thanks! Changed to "planktonic foraminifera shells ($\delta^{18}$Oc)". A table containing all abbreviations used in this manuscript has been added to the revised manuscript (see new Table. 1).

Line 83: if you just want to focus on paleoceanography, you need to add Sea after Mediterranean. However, I believe you can go further and say Mediterranean (region) paleoclimate as the modeling results should also be relevant for studies of speleothems and lacustrine sediments, besides paleoceanographic studies (that would also go beyond foraminiferal calcite shells). You actually hint to the broader potential impact in line 319!

Thank you for this synthesis and we fully agree. We are currently working with other teams on the IPSL coupled climate model, including the land surface model 'ORCHIDEEiso' and the atmospheric model 'LMDZiso', to implement water isotopes. This will enable further paleoclimate applications in the future across the entire Mediterranean region.

Line 121: verify bouquin AIEA; this reads like a placeholder text for a missing reference. It might also be IAEA.

Corrected (it was the French abbreviation).

Line 141: please provide reference for the standard isotopic values.

In models, the standard isotopic value is set arbitrarily, usually motivated by practical or computational constraints. Previous model-intercomparison projects of isotope-enabled models have shown that standard isotopic values could widely vary across models (Risi et al 2012).
"In reality, this value isn't standard; rather, it represents an average for the Mediterranean basin. We've set the simulations with these values to expedite computation time on the machine, as opposed to using the VSMOW value (i.e. The Mediterranean basin is largely more enriched as compared to the global scale)."
Changed in the revised ms (see section 2.2, lines 147-149 in the track changes version): "The ocean isotopic ratios are initially set to an average value for the Mediterranean basin of $\delta^{18}O_w$ = 1.5 ‰, $\delta D_w$ =8 ‰, and the pseudo-salinity tracer is set to 37 (we have initialized the simulations with these values to save a little computing time on the machine)".

Line 199: there exist additional/newer in-situ observations in the buffer zone west of the Strait of Gibraltar and one additional station in the Alboran Sea:

- Voelker, A.H.L., Colman, A., Olack, G., Waniek, J.J., Hodell, D., 2015. Oxygen and hydrogen isotope signatures of Northeast Atlantic water masses. Deep Sea Research Part II: Topical Studies in Oceanography 116, 89-106, doi: 1016/j.dsr2.2014.11.006.

- With the raw data available in Pangaea, e.g. Voelker, Antje H L; Colman, Albert Smith; Olack, Gerard; Waniek, Joanna J; Hodell, David A (2015): Oxygen and hydrogen isotopes measured on water bottle samples during EUROFLEETS cruise Iberia-Forams. PANGAEA, https://doi.org/10.1594/PANGAEA.831462

- Benetti, M., Reverdin, G., Aloisi, G., Sveinbjörnsdóttir, Á., 2017. Stable isotopes in surface waters of the Atlantic Ocean: Indicators of ocean-atmosphere water fluxes and oceanic mixing processes. Journal of Geophysical Research: Oceans 122, 4723-4742, doi: 1002/2017JC012712.

- With the data included in Reverdin, G., Waelbroeck, C., Pierre, C., et al., 2022. The CISE-LOCEAN seawater isotopic database (1998–2021). Earth Syst. Sci. Data 14, 2721-2735, doi: 10.5194/essd-14-2721-2022. https://www.seanoe.org/data/00600/71186/

- Voelker, A.H., 2023. Seawater oxygen and hydrogen stable isotope data from the upper water column in the North Atlantic Ocean (unpublished data). Interdisciplinary Earth Data Alliance (IEDA), doi: https://doi.org/10.26022/IEDA/112743

- and in the Alboran Sea itself: Voelker, Antje H L (2017): Seawater oxygen isotopes for Station POS334-73, Alboran Sea. Instituto Portugues do Mar e da Atmosfera: Lisboa, Portugal, PANGAEA, https://doi.org/10.1594/PANGAEA.878063

We appreciate the references and new data. The data from the Mediterranean Sea has been added to the model evaluation figures and will aid in future analysis. The data from the Strait

of Gibraltar (i.e. in Voelker., et al 2015, 2023) will be valuable for setting boundary conditions in upcoming simulations.

We have incorporated all the data from Benetti et al. (2017) and Reverdin et al. (2022) database, primarily situated in the Western Mediterranean (refer to new Fig 2a, below new data are shown in green in Fig. 2d), as well as the data from Voelker et al. (2017) localized in the Alboran basin (in bleu).

[Figure]

*Figure 1 The model outputs against in-situ data for the present-day situation. a) $\delta^{18}O_w$ (in ‰) distribution in the surface water (50 m depth). b) E-W vertical section of $\delta^{18}O_w$ (in ‰) in the western Mediterranean basin d) Zonal mean comparison of $\delta^{18}O_w$ (in ‰) average vertical profiles in the western basin presenting model results against in-situ data. c) and e) the same as b) and d) but for the eastern basin. Colour-filled dots represent in-situ observations from (Epstein and Mayeda, 1953; Stahl and Rinow, 1973; Pierre et al., 1986; Gat et al., 1996; Pierre, 1999, Voelker et al. 2017, Reverdin et al. 2022). Both model and in-situ data use the same colour scale.*

Line 251: Voelker et al. (2015, DSR II) obtained a lower slope of 0.32 for surface waters in the NE Atlantic with a strong bias towards subtropical waters (see their figure 11a). Craig and Gordon (1965) also observed a slope of 0.22 for the Atlantic's subtropical to tropical waters. So, your MedSea slopes fit well to those observations.

Thank you for alerting us to this. Indeed, Voelker et al. (2015) provided a thorough analysis of the $\delta^{18}O$-Salinity relationship. We have integrated the slope value calculated by Voelker et al. (2015) into our discussion and have included extra sentences in the revised manuscript's discussion section.
See section 3.2, lines 269-273 in the track changes version.
"The lower slopes reflect the impact of the evaporation surplus in the EMed (Voelker et al., 2015). High values of the slope are simulated in the western basin (> 0.5, Fig. 5a), especially in the Alboran basin which is influenced by Atlantic water characterized by a $\delta^{18}O_w$-S slope of 0.48 (Laube-

Lenfant, 1996; Pierre, 1999), and 0.32 obtained by Voelker et al. (2015) in the North East Atlantic with a strong bias towards subtropical waters."

Line 275: you could check the model's performance in the buffer zone west of the Strait of Gibraltar as much of the new data listed above include dD measurements.

The NEMO-MED12 grid covers the entire Mediterranean Sea and a small portion of the Atlantic Ocean to the west of Gibraltar, serving as a buffer zone for open boundary conditions. In this zone, 3D $\delta^{18}O_w$ and $\partial D_w$ are relaxed towards in-situ data fields (from Pierre, 1999; Craig and Cordon 1965), meaning that the tracer values are imposed as boundary conditions rather than being predicted by the model.

The comparison between the new $\partial D_w$ measurements and our imposed boundary conditions in the buffer zone demonstrates good consistency, as shown in the figure below.

[Figure]

*Figure 2 Comparison between simulated and observed dD in the Buffer zone (west of Gibraltar strait)*

Line 284: in a general sense, you could compare d-excess trends with Benetti, M., Reverdin, G., Pierre, C., Merlivat, L., Risi, C., Steen-Larsen, H.C., Vimeux, F., 2014. Deuteriu »m excess in marine water vapor: Dependency on relative humidity and surface wind speed during evaporation. Journal of Geophysical Research: Atmospheres 119, 2013JD020535,doi: 10.1002/2013JD020535. That reference might also fit in the discussion in line 377.

We appreciate your suggestion. We've now included a comparison between the simulated trends and the data published by Benetti et al. (2015). The following text has been incorporated into the revised manuscript:

*"In a more recent study, Benetti et al. (2015) observed a d-excess ranging from -1.56 to -1.72 ‰ in the surface waters of the eastern subtropical Atlantic. Their findings reveal a contrasting trend between increasing $\delta^{18}O_w$, $\delta D_w$, and decreasing d-excess, which corresponds closely with our simulated values. The authors suggest that d-excess variations are predominantly influenced by humidity and wind speed rather than mixing effects"*.

See changes at the end of section 3.3, lines 314-317, and in the discussion section (lines 416-421).

**Technical corrections:**
Line 5: define what sw in $\partial^{18}O_{sw}$ stands for: sea water or surface water? If sea water, the more common practice is to just use "w" for water.
$\partial^{18}O_{sw}$ stands for seawater. We agree with this suggestion and we change this abbreviation to $\partial^{18}O_w$.

Line 14: O is missing
Corrected

Line 19: correct spelling to "include"
Corrected

Line 48: replace input with inflow
Replaced

Line 49: replace into with in. Later in the sentence, correct the word order to Levantine Intermediate Water.
Done

Line 142: salinity is nowadays only given as a number (as correctly, done, for example in line 234); so, PSU should be deleted here.
Corrected

Line 201: I assume you mean eastern and not western basin as all the Gat et al. (1996) data are from the eastern basin.
Corrected. Thank you for pointing this out.

Line 207: EMed and WMed as acronyms should be defined.
Done, already defined in the introduction section (and in the new table 1)

Line 209: if you write western Mediterranean instead of WMed, Sea should be added behind Mediterranean.
Done

Lines 228-229: check the longitudes given for the eastern and western Med, respectively. If referring to the WMed, there should also not be a negative sign before the 6°E.
Corrected (section 3.1, lines 248-249).

Lines 365-366: add the article the before EMed/WMed, respectively.
Added

Figures: chosen color scheme: many of the figures include a red to green color range with symbols overlain in such colors. So, for color blind people it will be impossible to correctly read some of the figures. The author might want to check, if plotting in a different color range would be possible.
The second reviewer also highlighted this concern. In the updated version of the paper, we have modified the color palettes accordingly (see the new version of our ms).

**References**

Avnaim-Katav, S., Almogi-Labin, A., Schneider-Mor, A., Crouvi, O., Burke, A. A., Kremenetski, K. v., & MacDonald, G. M. (2019). A multi-proxy shallow marine record for Mid-to-Late Holocene climate variability, Thera eruptions and cultural change in the Eastern Mediterranean. *Quaternary Science Reviews*, *204*, 133–148. https://doi.org/10.1016/J.QUASCIREV.2018.12.001

Ayache, M., Dutay, J.-C., Arsouze, T., Révillon, S., Beuvier, J., & Jeandel, C. (2016). High-resolution neodymium characterization along the Mediterranean margins and modelling of Nd distribution in the Mediterranean basins. *Biogeosciences*, *13*(18). https://doi.org/10.5194/bg-13-5259-2016

Ayache, M., Dutay, J.-C., Jean-Baptiste, P., Beranger, K., Arsouze, T., Beuvier, J., Palmieri, J., Le-Vu, B., & Roether, W. (2015). Modelling of the anthropogenic tritium transient and its decay product helium-3 in the Mediterranean Sea using a high-resolution regional model. *Ocean Science*, *11*(3). https://doi.org/10.5194/os-11-323-2015

Ayache, M., Dutay, J.-C., Mouchet, A., Tisnérat-Laborde, N., Montagna, P., Tanhua, T., Siani, G., & Jean-Baptiste, P. (2017). High-resolution regional modelling of natural and anthropogenic radiocarbon in the Mediterranean Sea. *Biogeosciences*, *14*(5). https://doi.org/10.5194/bg-14-1197-2017

Bemis, B. E., Spero, H. J., Bijma, J., & Lea, D. W. (1998). Reevaluation of the oxygen isotopic composition of planktonic foraminifera: Experimental results and revised paleotemperature equations. *Paleoceanography*, *13*(2), 150–160. https://doi.org/10.1029/98PA00070

Benetti, M., Reverdin, G., Aloisi, G., Sveinbjörnsdóttir, Á., 2017. Stable isotopes in surface waters of the Atlantic Ocean: Indicators of ocean-atmosphere water fluxes and oceanic mixing processes. Journal of Geophysical Research: Oceans 122, 4723-4742, doi: 1002/2017JC012712.

Benetti, M., Reverdin, G., Pierre, C., Merlivat, L., Risi, C., Steen-Larsen, H. C., & Vimeux, F. (2014). Deuterium excess in marine water vapor: Dependency on relative humidity and surface wind speed during evaporation. *Journal of Geophysical Research: Atmospheres*, *119*(2), 584–593. https://doi.org/10.1002/2013JD020535

Beuvier, J., Béranger, K., Lebeaupin Brossier, C., Somot, S., Sevault, F., Drillet, Y., Bourdallé-Badie, R., Ferry, N., & Lyard, F. (2012). Spreading of the Western Mediterranean Deep Water after winter 2005: Time scales and deep cyclone transport. *Journal of Geophysical Research*, *117*(C7), C07022. https://doi.org/10.1029/2011JC007679

Craig, H., & Gordon, L. (1965). *Deuterium and oxygen 18 variations in the ocean and the marine atmosphere, in: Proc. Stable Isotopes in Oceanographic Studies and Paleotemperatures* (E. Tongiogi & F. Lishi, Eds.; pp. 9–130).

Dansgaard, W. (1964). Stable isotopes in precipitation. *Tellus*, *16*, 468–468.
de Castro Coppa, M. G. , M. Z. M. , P. B. , S. F. and T. R. E. ,. (1980). Distributione stagionale e verticale dei foraminiferi planctonici del golfo di Napoli. *Bol. Soc. Nat*, *89*, 1–25.

Delaygue, G., Bard, E., Rollion, C., Jouzel, J., Stiévenard, M., Duplessy, J. C., & Ganssen, G. (2001). Oxygen isotope/salinity relationship in the northern Indian Ocean. *Journal of Geophysical Research: Oceans*, *106*(C3), 4565–4574. https://doi.org/10.1029/1999JC000061

Delaygue, G., Jouzel, J., & Dutay, J. C. (2000). Oxygen 18-salinity relationship simulated by an oceanic general circulation model. *Earth and Planetary Science Letters*, *178*(1–2), 113–123. https://doi.org/10.1016/S0012-821X(00)00073-X

Drobinski, P., Anav, A., Lebeaupin Brossier, C., Samson, G., Stéfanon, M., Bastin, S., Baklouti, M., Béranger, K., Beuvier, J., Bourdallé-Badie, R., Coquart, L., D'Andrea, F., de Noblet-Ducoudré, N., Diaz, F., Dutay, J.-C., Ethe, C., Foujols, M.-A., Khvorostyanov, D., Madec, G., … Viovy, N. (2012). Model of the Regional Coupled Earth system (MORCE): Application to process and climate studies in vulnerable regions. In *Environmental Modelling & Software*. https://doi.org/10.1016/j.envsoft.2012.01.017
http://dx.doi.org/10.1016/j.envsoft.2012.01.017

ElElla, A. (1993). *Preliminary studies on the geochemistry of the Nile river basin, Egypt*.

Gat, R. (1996). Oxygen and Hydrogen Isotopes in the Hydrologic Cycle. *AREPS*, *24*, 225–262. https://doi.org/10.1146/ANNUREV.EARTH.24.1.225

Kallel, N., & Labeyrie, L. (1997). Enhanced rainfall in the Mediterranean region during the last Sapropel Event. *Oceanologica Acta*, *20*(5), 697–7712. https://www.researchgate.net/publication/277157107

Kim, S. T., & O'Neil, J. R. (1997). Equilibrium and nonequilibrium oxygen isotope effects in synthetic carbonates. *Geochimica et Cosmochimica Acta*, *61*(16), 3461–3475. https://doi.org/10.1016/S0016-7037(97)00169-5

Laube-Lenfant, E. (1996). Utilisation des isotopes naturels #1#8o de l'eau et #1#3c du carbone inorganique dissous comme traceurs oceaniques dans les zones frontales et d'upwelling. Cas du pacifique equatorial et de la mer d'alboran [Paris 6]. In *http://www.theses.fr*. http://www.theses.fr/1996PA066229

Li, L., Bozec, A., Somot, S., Béranger, K., Bouruet-Aubertot, P., Sevault, F., & Crépon, M. (2006). Chapter 7 Regional atmospheric, marine processes and climate modelling. *Developments in Earth and Environmental Sciences*, *4*(C), 373–397. https://doi.org/10.1016/S1571-9197(06)80010-8

Lombard, F., Labeyrie, L., Michel, E., Bopp, L., Cortijo, E., Retailleau, S., Howa, H., & Jorissen, F. (2011). Modelling planktic foraminifer growth and distribution using an ecophysiological multi-species approach. *Biogeosciences*, *8*(4), 853–873. https://doi.org/10.5194/BG-8-853-2011

Ludwig, W., Dumont, E., Meybeck, M., & Heussner, S. (2009). River discharges of water and nutrients to the Mediterranean and Black Sea: Major drivers for ecosystem changes during past and future decades? *Progress in Oceanography*, *80*(3–4), 199–217. https://doi.org/10.1016/j.pocean.2009.02.001

Nixon, S. W. (2003). Replacing the Nile: Are Anthropogenic Nutrients Providing the Fertility Once Brought to the Mediterranean by a Great River? *AMBIO: A Journal of the Human Environment*, *32*(1), 30–39. https://doi.org/10.1579/0044-7447-32.1.30

Palmiéri, J., Orr, J. C., Dutay, J. C., Béranger, K., Schneider, A., Beuvier, J., & Somot, S. (2015). Simulated anthropogenic $CO_2$ storage and acidification of the Mediterranean Sea. *Biogeosciences*, *12*(3), 781–802. https://doi.org/10.5194/BG-12-781-2015

Pierre, C. (1999). The oxygen and carbon isotope distribution in the Mediterranean water masses. *Marine Geology*, *153*(1–4), 41–55. https://doi.org/10.1016/S0025-3227(98)00090-5

Rebotim, A., Helga Luise Voelker, A., Jonkers, L., Waniek, J. J., Schulz, M., & Kucera, M. (2019). Calcification depth of deep-dwelling planktonic foraminifera from the eastern North Atlantic constrained by stable oxygen isotope ratios of shells from stratified plankton tows. *Journal of Micropalaeontology*, *38*(2), 113–131. https://doi.org/10.5194/JM-38-113-2019

Reverdin, G., Waelbroeck, C., Pierre, C., et al., 2022. The CISE-LOCEAN seawater isotopic database (1998–2021). Earth Syst. Sci. Data 14, 2721-2735, doi: 10.5194/essd-14-2721-2022. https://www.seanoe.org/data/00600/71186/

Rigual-Hernández, A. S., Sierro, F. J., Bárcena, M. A., Flores, J. A., & Heussner, S. (2012). Seasonal and interannual changes of planktic foraminiferal fluxes in the Gulf of Lions (NW Mediterranean) and their implications for paleoceanographic studies: Two 12-year sediment trap records. *Deep Sea Research Part I: Oceanographic Research Papers*, *66*, 26–40. https://doi.org/10.1016/J.DSR.2012.03.011

Risi, C., Bony, S., Vimeux, F., & Jouzel, J. (2010). Water-stable isotopes in the LMDZ4 general circulation model: Model evaluation for present-day and past climates and applications to climatic interpretations of tropical isotopic records. *Journal of Geophysical Research: Atmospheres*, *115*(D12), 12118. https://doi.org/10.1029/2009JD013255

Risi, C., Bony, S., Vimeux, F., Chongd, M., & Descroixe, L. (2010). Evolution of the stable water isotopic composition of the rain sampled along Sahelian squall lines. *Quarterly Journal of the Royal Meteorological Society*, *136*(S1), 227–242. https://doi.org/10.1002/QJ.485

Risi, C., Noone, D., Worden, J., Frankenberg, C., Stiller, G., Kiefer, M., Funke, B., Walker, K., Bernath, P., Schneider, M., Bony, S., Lee, J., Brown, D., & Sturm, C. (2012). Process-evaluation of tropospheric humidity simulated by general circulation models using water vapor isotopic observations: 2. Using isotopic diagnostics to understand the mid and upper tropospheric moist bias in the tropics and subtropics. *Journal of Geophysical Research: Atmospheres*, *117*(D5), 5304. https://doi.org/10.1029/2011JD016623

Risi, C., Ogée, J., Bony, S., Bariac, T., Raz-Yaseef, N., Wingate, L., Welker, J., Knohl, A., Kurz-Besson, C., Leclerc, M., Zhang, G., Buchmann, N., Santrucek, J., Hronkova, M., David, T., Peylin, P., & Guglielmo, F. (2016). The water isotopic version of the land-surface model ORCHIDEE: implementation, evaluation, sensitivity to hydrological parameters. *Hydrol Current Res*, *7*, 4. https://doi.org/10.4172/2157-7587.1000258

Roche, D., Paillard, D., Ganopolski, A., & Hoffmann, G. (2004). Oceanic oxygen-18 at the present day and LGM: equilibrium simulations with a coupled climate model of intermediate complexity. *Earth and Planetary Science Letters*, *218*(3–4), 317–330. https://doi.org/10.1016/S0012-821X(03)00700-3

Roether, W., Muennich, K. O., & Schoch, H. (2006). On the C-14 to tritium relationship in the North Atlantic Ocean. In *Radiocarbon* (Vol. 22, Issue 3, pp. 636–646). https://doi.org/10.2458/azu_js_rc.22.653

Sachse, D., Billault, I., Bowen, G. J., Chikaraishi, Y., Dawson, T. E., Feakins, S. J., Freeman, K. H., Magill, C. R., McInerney, F. A., van der Meer, M. T. J., Polissar, P., Robins, R. J., Sachs, J. P., Schmidt, H.-L., Sessions, A. L., White, J. W. C., West, J. B., Kahmen, A., Sachse, D., … Kahmen, A. (2012). Molecular Paleohydrology: Interpreting the Hydrogen-Isotopic Composition of Lipid Biomarkers from Photosynthesizing Organisms. *AREPS*, *40*(1), 221–249. https://doi.org/10.1146/ANNUREV-EARTH-042711-105535

Schmidt, G. A. (1998). Oxygen-18 variations in a global ocean model. *Geophysical Research Letters*, *25*(8), 1201–1204. https://doi.org/10.1029/98GL50866

Schmidt, G. A. (1999). Forward modeling of carbonate proxy data from planktonic foraminifera using oxygen isotope tracers in a global ocean model. *Paleoceanography*, *14*(4), 482–497. https://doi.org/10.1029/1999PA900025

Schroeder, K., Ribotti, a., Borghini, M., Sorgente, R., Perilli, a., & Gasparini, G. P. (2008). An extensive western Mediterranean deep water renewal between 2004 and 2006. *Geophysical Research Letters*, *35*, 1–7. https://doi.org/10.1029/2008GL035146

Vadsaria, T., Li, L., Ramstein, G., & Dutay, J. C. (2020). Development of a sequential tool, LMDZ-NEMO-med-V1, to conduct global-to-regional past climate simulation for the Mediterranean

basin: an Early Holocene case study. *Geoscientific Model Development*, *13*(5), 2337–2354. https://doi.org/10.5194/GMD-13-2337-2020

van Breukelen, M. R., Vonhof, H. B., Hellstrom, J. C., Wester, W. C. G., & Kroon, D. (2008). Fossil dripwater in stalagmites reveals Holocene temperature and rainfall variation in Amazonia. *Earth and Planetary Science Letters*, *275*(1–2), 54–60. https://doi.org/10.1016/J.EPSL.2008.07.060

Vergnaud Grazzini, C. , G. C. , P. C. , P. C. , and U. M. J. (1986). Foraminifères planctoniques de Méditerranée en fin d'été. Relations avec les structures hydrologiques,. *Mem. Soc. Geol. Ital*, *36*, 175–188.
Voelker, A. H. L., Colman, A., Olack, G., Waniek, J. J., & Hodell, D. (2015). Oxygen and hydrogen isotope signatures of Northeast Atlantic water masses. *Deep Sea Research Part II: Topical Studies in Oceanography*, *116*, 89–106. https://doi.org/10.1016/J.DSR2.2014.11.006

Voelker, A.H., 2023. Seawater oxygen and hydrogen stable isotope data from the upper water column in the North Atlantic Ocean (unpublished data). Interdisciplinary Earth Data Alliance (IEDA), doi: https://doi.org/10.26022/IEDA/112743

Voelker, A.H.L., Colman, A., Olack, G., Waniek, J.J., Hodell, D., 2015. Oxygen and hydrogen isotope signatures of Northeast Atlantic water masses. Deep Sea Research Part II: Topical Studies in Oceanography 116, 89-106, doi: 1016/j.dsr2.2014.11.006.

Voelker, Antje H L (2017): Seawater oxygen isotopes for Station POS334-73, Alboran Sea. Instituto Portugues do Mar e da Atmosfera: Lisboa, Portugal, PANGAEA, https://doi.org/10.1594/PANGAEA.878063

Voelker, Antje H L; Colman, Albert Smith; Olack, Gerard; Waniek, Joanna J; Hodell, David A (2015): Oxygen and hydrogen isotopes measured on water bottle samples during EUROFLEETS cruise Iberia-Forams. PANGAEA, https://doi.org/10.1594/PANGAEA.831462

Vörösmarty, C. J., Fekete, B. M., & Tucker, B. A. (1996). Global River Discharge Database (RivDIS V1.0), International Hydrological Program. *Global Hydrological Archive and Analysis Systems, UNESCO, Paris*.

---

## Author Comment (AC2)

**Manuscript " Modelling the water isotopes distribution in the Mediterranean Sea using a high-resolution oceanic model (NEMO-MED12-watiso-v1.0): Evaluation of model results against in-situ observations"**

Mohamed Ayache[1], Jean-Claude Dutay[1], Anne Mouchet[2], Kazuyo Tachikawa[3], Camille Risi[4], and Gilles Ramstein[1]

[1]Laboratoire des Sciences du Climat et de l'Environnement, CEA-CNRS-Université Paris Saclay, 91191, Gif-sur-Yvette, France
[2]Freshwater and OCeanic science Unit of reSearch (FOCUS), Université de Liège, B-4000 Liège
[3]Aix Marseille Univ, CNRS, IRD, INRAE, Coll France, CEREGE, 13545, Aix-en-Provence, France
[4]Laboratoire de Météorologie Dynamique, IPSL, CNRS, Sorbonne Université, Paris, France Correspondence: Mohamed Ayache (mohamed.ayache@lsce.ipsl.fr)

**Reply to reviewers' comments**

Dear Pr. I., Andrew Yool,

We would like to thank you for providing us the opportunity to revise our manuscript, and we are extremely grateful to Pr. Antje Voelker, Pr. Allegra N. LeGrande and the anonymous reviewer for their careful reading and comments that helped to improve our manuscript significantly.

We have revised our manuscript and provided a detailed response to each reviewer's comment and request below.

**Color code**
Reviewer comments
Authors response
The modifications performed in the manuscript appear in red above and in the revised manuscript with Changes Marked.

**3: Review by Allegra N. LeGrande:**

Water isotope tracers are indeed a useful way to track the water cycle, and this study seeks to provide for high resolution insight into the Mediterranean ocean.
The authors of this study include expert isotope modelers, so the work is on the whole very solid.
I have Mostly few questions about the specifics of implementation and the write up.
1) for someone who is *not* a water isotope modeler, the casual inclusion of shorthand / jargon without explanation needs to be expressly defined. I.E., $\delta^{18}O_{sw}$ or $\delta D$ (also—shouldn't you write $\delta D_{sw}$ to be consistent?) or $CaCO_3$ or $\delta^{18}O_c$ all need to be defined – what does the delta mean. What do the subscripts mean. Some of the equation rendering has broken down maybe on the author's side, maybe on the Copernicus side.

We completely agree with Dr. Allegra N. LeGrande on this point, and we regret this lack of information, which is necessary for a better understanding of our manuscript. The same point was raised by Dr. Antje Voelker. We have added all the missing information in the revised version. $\partial^{18}O_{sw}$ stands for seawater, we change this abbreviation to $\partial^{18}O_w$ (use "w" for water). $\partial^{18}O_c$ we use c for calcite, and $\partial D_w$ for deuterium. A table containing all abbreviations used in this manuscript has been added to the revised manuscript (ms)

See new Table 1

2) In the write up of previous work in the med for isotopes, the authors may (not?) be aware that there is almost certainly a mistake in the δD values of Gat as they vary much much less than $\delta^{18}O_{sw}$ – probably the original source should be sought out for that validation.

Thank you for bringing to our attention the discrepancy from the data of Gat et al. (1996). After verification, we have identified a problem with the sources used in the previous version of our manuscript. The shift has been corrected in the new version of our paper, as shown in the figure below (corrected data are plotted in green in panel e). We have also added new data in the western basin from Reverdin et al., (2022).

[Figure]

Figure 1 The model outputs against in-situ data for the present-day situation. a) $\delta D_w$ (in ‰) distribution in the surface water (50 m depth). b) E-W vertical section of $\delta D_w$ (in ‰) in the western Mediterranean basin d) Zonal mean comparison of $\delta D_w$ (in ‰) average vertical profiles in the western basin presenting model results against in-situ data. c) and e) the same as b) and d) but for the eastern basin. Colour filled

dots represent in-situ observations from (Gat et al., 1996; Reverdin et al., 2022). Both model and in-situ data use the same colour scale.

3) When it is said that 'we use fluxes' from LMDZiso – that is surface water isotope fluxes? How are fluxes from rivers handled? Do you use observed isotope values or simulated ones? (Do the simulated river values closely approximate the measured ones?) If no measurements are available, what was done instead?

Ideally, the simulation of surface water isotope fluxes should be carried out using the land surface model ORCHIDEE. Isotopes are incorporated into the river discharge of ORCHIDEE, as described by Risi et al. (2016). However, the isotopic version of ORCHIDEE is outdated and cannot be coupled with the current version of LMDZ-iso. A joint project is currently underway to reintroduce water isotopes in the new versions of ORCHIDEE and to couple with LMDZ-iso.

In this scenario, we adopt an alternative solution proposed by Delaygue et al. (2000) to represent the isotopic flux carried by rivers to the ocean: this flux is calculated as $^{18}R_{river}= {^{18}R_{precipLMDZiso}} \times R_{runoff}$, where R is the ratio $^{18}O/O$, $R_{runoff}$ is the same freshwater forcing as that used in the dynamical simulation (Beuvier et al., 2012; Palmiéri et al., 2015), and $^{18}R_{precipLMDZiso}$ is the isotopic ratio in precipitation at the same time and location. Monthly $^{18}R_{precipLMDZiso}$ runoff values of the 33 main river mouths covering the entire Mediterranean draining basin were computed using the climatological mean of the interannual dataset of Ludwig et al. (2009) and the RivDis dataset from Vörösmarty et al. (1996). This alternative approach has shown effectiveness both in the results presented in this paper and globally, as demonstrated by Delaygue et al. (2000). The advantage of this approach lies in its reproducibility across different timescales and locations, as well as its applicability to paleoclimate studies where observed isotope values from rivers are very limited.

Following the reviewer's suggestion, we conducted additional sensitivity simulations to better evaluate the impact of $\partial^{18}O_{river}$ (please see the answer to question 9 below). A new section is added to the appendix to further elucidate this point (see Appendix E).

4) On page 5, they say "it is common to transport the isotopic ratio rather than the individual isotope…" then later "and pseudo-salinity fluxes". I don't know NEMO that well, but I am going to guess they are saying in a round about way that this ocean model has a rigid lid instead of a free surface. They should say either way. Because most isotope models do *not* in fact transport around concentrations of isotopes, they transport around mass. Sure – some models do not actually conserve mass – they are forever having to reimplement water isotopes in their code because they have virtual moisture or salt fluxes. Anyhow, those who *can* do indeed transport around mass not concentration. The per mil isotopic composition is determined on post-processing. Why? This is done so that the isotope / tracer code can have an exact replica of 'water' from the non-tracer code and this tracer can be 1:1 compared throughout the entire model to made sure mass isn't being gained/lost anywhere spuriously. Isotopic composition comes into play because SMOW is defined and fractionation at phase changes is defined. This is, in general, simpler for an ocean model where the mass of water is simply (MO – S), but if

you have a rigid lid, then you have virtual mass fluxes of isotopes. Clarity for this point is required.

It is important to note that NEMO (and OGCMs in general) have representations of concentration/dilution processes that depend on the context:

− In this study, we used the off-line uncoupled mode of NEMO (pre-calculated dynamics): in this case, we use the linear free surface (fixed volume) with explicit fluxes of evaporation, precipitation, and runoff (calculated according to Delaygue et al. 2000, see our answer to point 3 and 9 of Dr. Allegra N. LeGrande). In offline mode, the model-intrinsic evaporation and precipitation fluxes have to be switched off, since the tracers are already influenced by freshwater fluxes in the forcing.

− It's possible to use the online coupled mode of NEMO to calculate the dynamic variables (circulation fields U, V, and W) in real-time. The sea surface elevation and model layer thicknesses are modified by the freshwater flux (E-P-R), which in turn affects the model volume. It is crucial that the total volume variations precisely follow the E-P forcing used to drive the isotopic module to ensure the perfect conservation of tracer content.

An important issue when modeling isotopes is that of conservation. Since the ocean is not coupled to the atmosphere the tracer cycle is not closed. In consequence, drift occurs. A global correction must be applied based either on the instantaneous or yearly averaged imbalance of surface fluxes for each tracer. The drift due to the linearized free-surface equation and, if relevant, the Asselin filter, are corrected using a specified routine in NEMO. Technical aspects relative to the conservation of tracers in NEMO are not addressed here; they may be found in the NEMO engine webpage (https://www.nemo-ocean.eu/).

The boundary conditions at the ocean-atmosphere interface are provided by an atmospheric GCM with a comprehensive representation of water isotopes (LMDZiso GCM; Risi et al.,2010). They consist of climatological gross fluxes of evaporation and precipitation with their isotopic composition.

The isotopic composition is determined on post-processing because here we transport the isotopic ratio (see equation 1), this allows us to carry a single tracer "$^{18}R$" instead of two tracers "$^{18}O$ and $^{16}O$", which saves computing time on the machine, this point is very important for model performance when using this water isotope package in the coupled model and in very long paleo simulations. It is common practice too to transport the isotopic ratio rather than the individual species. e.g., radiocarbon distribution "$^{14}C/C$" in the Mediterranean Sea (Ayache et al., 2017) and the isotopic composition of water vapor in the advection scheme of LMDZ (Risi et al., 2010b).

Water isotopes behave like conservative tracers in the ocean; they are only modified by fluxes through open boundaries (Craig and Gordon, 1965; Schmidt, 1998; Delaygue et al., 2000; Roche et al., 2004). Isotopic fluxes in and out of the ocean are associated with water transfer at the ocean-atmosphere and land-ocean boundaries.

5) The 'interpolated to 20 min time step'—does this mean that actual rainfall and weather systems otherwise are regressed and then passed to the model at this finer time step, or is the daily value

simply applied/scaled at the 20 minute interval. I would guess that if you are using some sort of nudged version of LMDZiso that there is useful information at a finer timescale (i.e., if its been nudged at 3 hour timesteps, why not interpolate from 3hr->20min) – otherwise you'll miss the finer temporal resolution features. You wouldn't need to store *all* of LMDZiso values at that timestep—just those in your domain.

In numerical modeling, a time step refers to the discrete increment of time over which the model's equations are solved. The choice of time step is crucial as it can impact the accuracy and stability of the model's simulations

In this study the fields of physics variables are read and interpolated at each model time step, i.e., the circulation fields (U, V, W) previously computed by the dynamical model are read daily and interpolated to give values for each 20 min time step. NEMO-related forcings are provided at a day-frequency while isotopic-related fluxes are given on a monthly basis.

We chose a lower frequency of atmospheric forcing compared to NEMO forcing to evaluate model performance in the current climate state against in-situ data randomly observed between 1982 and 2022. Also, the high-frequency coupling could only be performed using an on-line coupled model (which is not currently possible). Consequently, we chose to use the climatological mean of the LMDZ-iso 1990-2020 simulation as boundary conditions. This choice aims to minimize the warming trend and to ensure an average state of precipitation and evaporation, thus reducing high-frequency variability.

Additional details have been incorporated into the revised manuscript to further clarify this aspect:

See section 2.2 lines 140-144

"The physical forcing fields are readed and interpolated at each model time step, i.e., the circulation fields (U, V, W) previously computed by the dynamical model are read daily and interpolated to give values for each 20 min time step. NEMO-related forcings are provided at a day frequency while isotopic-related fluxes are given monthly (see below for the atmospheric forcing)."

And section 2.3 lines 166-171

"The aim is to assess the model's performance in the present climate and against in-situ data observed randomly over the historical period. Therefore, we have opted to use the climatological mean of the LMDZ-iso 1990-2020 simulation as boundary conditions. This choice was made to minimize the warming trend during this period and to ensure that the precipitation and evaporation simulated by the LMDZ-iso model for the current climate situation are as close to the average state as possible, with minimal impact from inter-annual variability."

6) I'm still confused about the pseudo-salinity tracer. Please explain

The water fluxes from the stand-alone (non-coupled) experiments with LMDZiso are not identical to those constraining NEMO-Med12. Hence $\delta^{18}O_w$ or $\delta D_w$ computed with the water fluxes obtained with LMDZiso would not be consistent with the salinity predicted by NEMO-MED12. For this reason, we compute a "pseudo salinity" $S_w$ (Delaygue et al., 2000; Roche et al., 2004). This additional passive tracer does not affect the ocean dynamics. Its sole purpose is

to allow a coherent assessment of the relation of the isotopic fields predicted by the model with salinity since they are computed with the same fresh-water forcing.

The evolution equation for $S_w$ is given by:

$$\rho_0 K \bigtriangledown S_w|_{z=\eta} = (\mathcal{E} - \mathcal{P} - \mathcal{R} + |\mathcal{I}|)S_w - (^{S_w}\mathcal{I}).$$

with the further assumption that the salinity associated with evaporation, precipitation, and run-off is zero (no effect of freezing/melting on the concentration/dilution of pseudo-salinity in the Mediterranean Sea), the boundary condition for salinity reads.

$$\rho_0 K \bigtriangledown S_w|_{z=\eta} = (\mathcal{E} - \mathcal{P} - \mathcal{R})S_w$$

The basic understanding of these atmospheric fluxes is that evaporation tends to increase the surface salinity, and the 18O/O ratio, in contrast to precipitation and runoff.

Below, we have plotted the anomaly in salinity-pseudo-salinity to assess the correspondence between pseudo-salinity results and standard modeled salinity. The well-known east-west gradient is effectively captured by recalculated pseudo-salinity, showing very similar values to those of standard salinity. Minor deviations are noticed in the Gulf of Lions and the Algerian Basin, attributed to overlooked mesoscale activity impacts in the global LMDZiso simulation. Overall, the pseudo-salinity globally yields values highly comparable to standard simulated salinity.

[Figure]

*Figure 2 a) Standard simulated salinity from NEMO-MED12 in the surface model. b) Pseudo-salinity simulated in the surface water. c) the anomaly a) - b)*

This figure is included in the appendix of the revised manuscript, accompanied by additional details to provide a clearer explanation of the pseudo-salinity concept (see Appendix D).

7) Page 6: the present day values seem awfully low. CO2 of 348ppm – I rarely encounter PhD students anymore born in a world with CO2 this low.

We agree with the point made. It is evident that the value of 348 ppm used is significantly lower than the current value of 421 ppm. We have used this value because here we evaluate model performance against in-situ data observed at different times between 1982 and 2022.

8) NEMO-MED12 grid is jargon that I don't understand.

The term "NEMO-MED12 grid" refers to the specific configuration of the NEMO model that is used for the Mediterranean Sea (Beuvier et al., 2012). The "MED12" part of "NEMO-MED12" indicates that this configuration of the model has a resolution of $1/12°$.
The NEMO-MED12 grid is an extraction from the global ORCA-$1/12°$ grid. This corresponds to a grid cell size between 6 to 7.5km from 46°N to 30°N and represents a grid size of $567 \times 264$ points. NEMO-MED12 covers the whole Mediterranean Sea plus a buffer zone including a part of the near Atlantic Ocean, from 30°N to 47°N, and from 11°W to 36°E. The Black Sea is not represented. Clarified in the revised ms (see section 2.1, lines 87-88).

"The NEMO-MED12 grid is an extraction from the global ORCA-$1/12°$ grid. This corresponds to a grid cell size between 6 to 7.5km from 46°N to 30°N and represents a grid size of $567 \times 264$ points."

9) Still confused on L165-170 how the isotopic composition for the rivers was determined. It sounds like you are saying that the isotopic composition of river discharge = local grid box precipitation isotopic composition (which would be wrong of course). Can't you use observations *or* use d18Oriver from LMDZiso (or another isotope enabled model). Since you have already established that the Med is an evaporative basin, you might expect that d18Oriver to be a bit enriched compared to d18Oprec… (Places downriver or downhill in a P>E location you would expect d18Oriver to be a bit depleted compared to d18Oprec…) But the Med, and particular places like the Nile, you definitely should expect some evaporation to strip out the light isotopes of the river.

Thank you for your analysis and suggestions regarding the isotopic composition of the runoffs. In addition to our response to question 3, here are some key points to clarify:

– In response to the reviewer's suggestion, we conducted sensitivity simulations to assess the impact of computing the isotopic composition of rivers based on the isotopic composition of precipitation. Two new experiments (EXP1 and EXP2) were conducted using output from an earlier version of LMDZiso coupled to ORCHIDEE-iso (cf. Risi et al., 2016) at a lower resolution of R96x71.

- EXP1: Employed the approach described in our submitted paper, where $^{18}R_{river} = {}^{18}R_{precipLMDZiso} \times R_{runoff}$

- EXP2: Integrated the simulated $\delta^{18}O$ of rivers from the older version of LMDZiso at R96x71 resolution, where $^{18}R_{river} = {}^{18}R_{river} \times R_{runoff}$

- Here, R is the ratio $^{18}O/^{16}O$, $^{18}R_{precip}$ and $^{18}R_{riverLMDZiso}$ are derived from LMDZiso (R96x71, Risi et al., 2016), while $R_{runoff}$ is from the interannual dataset of Ludwig et al. (2009) and the RivDis dataset from Vörösmarty et al. (1996).

- The results of these sensitivity simulations are shown in the figure 9 below. In EXP1, the model reproduces a reasonable east-west gradient similar to our results using a higher version of LMDZiso (R96), as shown in Fig. 2a of the submitted paper. In EXP2, the addition of the $\delta^{18}O$ of rivers simulated by LMDZiso reveals a more enriched $\delta^{18}O_{river}$ compared to $\delta^{18}O_{precip}$, as predicted by the reviewer. Indeed, evaporation can enrich heavier isotopes in remaining water, including rivers, which is particularly evident for the Po river, exhibiting a clear positive anomaly around 0.5‰ near the coast and dispersed over the Adriatic Sea. The impact of other main rivers (e.g., Rhone and Po) remains very close to the coast, rapidly dispersed by circulation.
- However, the impact of the Nile significantly influences the $\delta^{18}O_w$ signal simulated in EXP2, highlighting a well-known issue in ORCHIDEE concerning the simulation of Nile discharge, where ORCHIDEE tends to largely overestimate the discharge, as depicted in figure 9 below.
- Consequently, we opted not to utilize the global version of LMDZiso due to the complex hydrology of the Mediterranean region. Instead, we employed a combination of model outputs and in-situ data to estimate the runoffs entering the Mediterranean Sea. For the isotopic composition, we adopted the same approach used by Delaygue et al. (2000).
- In conclusion, these sensitivity simulations (EXP1 and EXP2) showed an enrichment of $\delta^{18}O$ in the rivers due to evaporation, especially for the Po. The influence of the Nile significantly affects the signals, which has prevented the use of this version of LMDZiso (R71) and we are unable to couple this old version of ORCHIDEE (outdated) with the current version of LMDZiso. Therefore, the approach of Delaygue et al. 2000 was chosen over the data for its reproducibility and usability in paleo simulations.
- A new section is added to the appendix to further elucidate this point (see Appendix E). and we have mentioned this limitation in the conclusion of our paper lines 476-479:

"Here we calculate the isotopic composition of rivers based on the isotopic composition of precipitation, which means that the enriched $\delta^{18}O$ in rivers due to evaporation is not included in our simulation. It is recommended that a future study better represents the $\delta^{18}O_{river}$ (see Appendix E)."

[Figure]

*Figure 3 a) EXP1: we use the same approach as described in our submitted paper, i.e., 18Rriver= 18Rprecip × Rrunoff. b) EXP2: we added the d18Oriver simulated by the old version of LMDZiso at lower resolution R96x71. 18Rriver=18Rriver × Rrunoff. C) the deference EXP2 − EXP1.*

10) Can you write up the E-W surface d18Osw context from obs ? Maybe putting observed d18Oriver would make for a better gradient. (The baseline composition is set by your SMOW definition—I'd worry less about that.)

 For this study, we've opted not to rely on $\delta^{18}O_{river}$ observations and utilize the framework outlined by Delaygue et al. (2000), but we agree that this is a limitation of this study, and we now stated this limitation in the conclusion of our paper, lines 476-479:

"Here we calculate the isotopic composition of rivers based on the isotopic composition of precipitation, which means that the enriched $\delta^{18}O$ in rivers due to evaporation is not included in our simulation. It is recommended that a future study better represents the $\delta^{18}O_{river}$ (see Appendix E).".

In the future, the ongoing project at IPSL, aimed at updating and integrating various components of the IPSL model (LMDZiso, ORCHIDEEiso, and NEMOiso), will undoubtedly enhance the representation of $\delta^{18}O_{river}$ in future studies.

11) For deriving d18O-S relationships – can you put yours in context of the LMDZiso? Would you expect NEMOiso to differ that much given that you are prescribing your end member from the coupled model? Is this a useful section?

I'm not sure if I've understood this question correctly !

We have prescribed the end members from E and P of LMDZiso and not from the IPSL coupled model, it's important to note a distinction between the global model and NEMO-MED12iso. As explained in the response to question 6, NEMO-MED12 operates as an eddy-permitting model, which is clearly shown in Fig.8a and Fig.8b of simulated sea surface salinity (please refer to answer 6 above).

12) For section 3.3 – can you please check the Gat96 comparison. Does it make sense?

Corrected, see answer 2 above. Again, we apologize for the delay in the Gat et al. (1996) data.

13) For the d18Ocalcite discussion, what is the correlation between d18Oc and temperature temporally and spatially. For interannual variability, does the inclusion of d18Osw confound the correlation. Also—you are presuming surface dwelling foraminifera. Maybe its interesting to look at species specifiic d18Oc.

Calcite $\delta^{18}O_c$ is widely used in paleoclimate research. Understanding its seasonal variability is crucial for reconstructing past climates. The influence of seasonal temperature variability on $\delta^{18}O_c$ (equation 6) is important, particularly in the Mediterranean Sea because of marked seasonal thermal contrast. The $\delta^{18}O_c$ values are determined by both $\delta^{18}O_w$ and the seawater temperature at the calcification depth. For planktonic foraminifera such as *Globigerinoides ruber* and *Globigerina bulloides*, the calcification depth typically ranges from 0 to 100 meters, though variations exist depending on the basin (De Castro Coppa et al., 1980; Grazzini et al., 1986). The season of maximal foraminiferal production can be estimated by data from sediment traps. For instance, *G. ruber* and *G. bulloides* have been associated with calcification seasons in October-November and April-May according to Kallel et al. (1997), while others suggest January-March (Avnaim-Katav et al., 2019) and February-April (Rigual-Hernandez et al., 2012).

In this context, we used our model results to explore the relationship between $\delta^{18}O_c$ and temperature. We employed a paleotemperature equation for inorganic calcite by Kim and O'Neil (1997), which was modified by Bemis et al. (1998), as shown in Fig. 10. Our simulations indicate that the highest $\delta^{18}O_c$ values occur during winter (February, March), while the lowest values are observed during summer/autumn. Although the available observational data do not cover all months of the year, our results align with existing data, highlighting the significant influence of temperature on $\delta^{18}O_c$ in the Mediterranean Sea. Nonetheless, a dedicated study should be conducted to further elucidate the seasonal aspect.

In the revised version of our paper, we have included additional sentences to provide clarity on the seasonality aspect of $\delta^{18}O_c$ (see section 4 lines 427-441).

"Calcite $\delta^{18}O_c$ is widely used in paleoclimate research. Understanding its seasonal variability is crucial for reconstructing past climates. The influence of seasonal temperature variability on $\delta^{18}O_c$ (equation 6) is important, particularly in the Mediterranean Sea because of marked seasonal thermal contrast. The $\delta^{18}O_c$ values are determined by both $\delta^{18}O_w$ and the seawater temperature at the calcification depth. For planktonic foraminifera such as *Globigerinoides ruber* and *Globigerina bulloides*, the calcification depth typically ranges from 0 to 100 meters, though variations exist depending on the basin (De Castro Coppa et al., 1980; Grazzini et al., 1986). The season of maximal foraminiferal production can be estimated by data from sediment traps. For instance, *G. ruber* and *G. bulloides* have been associated with calcification seasons in October-November and April-May according to Kallel et al. (1997), while others suggest January-March (Avnaim-Katav et al., 2019) and February-April (Rigual-Hernandez et al., 2012). In this context, we used our model results to explore the relationship between the $\delta^{18}O_c$ and temperature. We employed a paleotemperature equation for inorganic calcite by Kim and O'Neil (1997), modified by Bemis et al. (1998), as shown in Fig. 10. Our simulations indicate that the highest $\delta^{18}O_c$ values occur during winter (February, March), while the lowest values are observed during summer/autumn. Although the available observational data do not cover all months of the year, our results align with existing data, highlighting the significant influence of temperature on $\delta^{18}O_c$ in the Mediterranean Sea. Nonetheless, a dedicated study should be conducted to further elucidate the seasonal aspect."

For inter-annual variability, the inclusion of $\delta^{18}O_w$ can indeed confound the correlation with $\delta^{18}O_c$. This is because $\delta^{18}O_w$ is influenced by factors such as evaporation, precipitation, and runoff, which can vary on interannual timescales. However, we did not delve into interannual variability in this paper. It should be examined in a separate study. We now discuss this issue in the article: ". The aim is to assess the model's performance in the present climate and against in-situ data observed randomly over the historical period. Therefore, we have opted to use the climatological mean of the LMDZ-iso 1990-2020 simulation as boundary conditions. This choice was made to minimize the warming trend during this period and to ensure that the precipitation and evaporation simulated by the LMDZ-iso model for the current climate situation are as close to the average state as possible, with minimal impact from inter-annual variability".
See section 2.3 and lines 166-171 in the track changes version.

Regarding the presumption of surface-dwelling foraminifera, it's true that different species of foraminifera calcify at different depths in the water column (e.g Rebotim et al., 2019). Therefore, the $\delta^{18}O_c$ values can vary between species, reflecting the different environmental conditions at their respective depths (Rebotim et al., 2019). In our forthcoming paper, which focuses on paleo events known as sapropels, we are currently implementing a module (developed by A. Mouchet, University of Liege) to facilitate a direct comparison of the model with proxy data (species-dependent). This module operates under the assumption that each planktonic foraminiferal species prefers a specific range of depth and temperature, similar to the approach used by Schmidt (1999). At each time step and geographical location, the possibility of occurrence of a particular foram species is evaluated on the basis of its preferred temperature and depth ranges. This process allows us to determine the mean $\delta^{18}O_c$ along with the mean $\delta^{18}O_w$ and temperature experienced by foraminifera during their life cycle. Currently,

the module considers four planktonic foraminifera species: *G. ruber*, *Neogloboquadrina pachyderma*, *Neogloboquadrin incompta*, and *G. bulloides* (Schmidt, 1999; Lombard et al., 2011).

Additional details have been incorporated into the revised manuscript to further clarify this aspect (see section 4, line 427-441).

13) There are *some* existing SWING comparisons of different isotopic compositions for different groups. Maybe for your next paper you could pull those in, but for this one, you should at least mention and speculate if it would be useful.

We have added a sentence in the conclusion to mention the SWING2 project (Risi et al 2012): "It would be interesting to compare how NEMO-MED12 responds to inputs from different isotope-enabled atmospheric GCMs, as documented in SWING2 (Risi et al., 2012). In addition, an intercomparison of results from different coupled models could be valuable as an extension of SWING2."

See section 4 lines 479-481

**References**

Avnaim-Katav, S., Almogi-Labin, A., Schneider-Mor, A., Crouvi, O., Burke, A. A., Kremenetski, K. v., & MacDonald, G. M. (2019). A multi-proxy shallow marine record for Mid-to-Late Holocene climate variability, Thera eruptions and cultural change in the Eastern Mediterranean. *Quaternary Science Reviews*, *204*, 133–148. https://doi.org/10.1016/J.QUASCIREV.2018.12.001

Ayache, M., Dutay, J.-C., Arsouze, T., Révillon, S., Beuvier, J., & Jeandel, C. (2016). High-resolution neodymium characterization along the Mediterranean margins and modelling of Nd distribution in the Mediterranean basins. *Biogeosciences*, *13*(18). https://doi.org/10.5194/bg-13-5259-2016

Ayache, M., Dutay, J.-C., Jean-Baptiste, P., Beranger, K., Arsouze, T., Beuvier, J., Palmieri, J., Le-Vu, B., & Roether, W. (2015). Modelling of the anthropogenic tritium transient and its decay product helium-3 in the Mediterranean Sea using a high-resolution regional model. *Ocean Science*, *11*(3). https://doi.org/10.5194/os-11-323-2015

Ayache, M., Dutay, J.-C., Mouchet, A., Tisnérat-Laborde, N., Montagna, P., Tanhua, T., Siani, G., & Jean-Baptiste, P. (2017). High-resolution regional modelling of natural and anthropogenic radiocarbon in the Mediterranean Sea. *Biogeosciences*, *14*(5). https://doi.org/10.5194/bg-14-1197-2017

Bemis, B. E., Spero, H. J., Bijma, J., & Lea, D. W. (1998). Reevaluation of the oxygen isotopic composition of planktonic foraminifera: Experimental results and revised paleotemperature equations. *Paleoceanography*, *13*(2), 150–160. https://doi.org/10.1029/98PA00070

Benetti, M., Reverdin, G., Aloisi, G., Sveinbjörnsdóttir, Á., 2017. Stable isotopes in surface waters of the Atlantic Ocean: Indicators of ocean-atmosphere water fluxes and oceanic mixing processes. Journal of Geophysical Research: Oceans 122, 4723-4742, doi: 1002/2017JC012712.

Benetti, M., Reverdin, G., Pierre, C., Merlivat, L., Risi, C., Steen-Larsen, H. C., & Vimeux, F. (2014). Deuterium excess in marine water vapor: Dependency on relative humidity and surface wind speed during evaporation. *Journal of Geophysical Research: Atmospheres*, *119*(2), 584–593. https://doi.org/10.1002/2013JD020535

Beuvier, J., Béranger, K., Lebeaupin Brossier, C., Somot, S., Sevault, F., Drillet, Y., Bourdallé-Badie, R., Ferry, N., & Lyard, F. (2012). Spreading of the Western Mediterranean Deep Water after winter 2005: Time scales and deep cyclone transport. *Journal of Geophysical Research*, *117*(C7), C07022. https://doi.org/10.1029/2011JC007679

Craig, H., & Gordon, L. (1965). *Deuterium and oxygen 18 variations in the ocean and the marine atmosphere, in: Proc. Stable Isotopes in Oceanographic Studies and Paleotemperatures* (E. Tongiogi & F. Lishi, Eds.; pp. 9–130).

Dansgaard, W. (1964). Stable isotopes in precipitation. *Tellus*, *16*, 468–468.

de Castro Coppa, M. G. , M. Z. M. , P. B. , S. F. and T. R. E. ,. (1980). Distributione stagionale e verticale dei foraminiferi planctonici del golfo di Napoli. *Bol. Soc. Nat*, *89*, 1–25.

Delaygue, G., Bard, E., Rollion, C., Jouzel, J., Stiévenard, M., Duplessy, J. C., & Ganssen, G. (2001). Oxygen isotope/salinity relationship in the northern Indian Ocean. *Journal of Geophysical Research: Oceans*, *106*(C3), 4565–4574. https://doi.org/10.1029/1999JC000061

Delaygue, G., Jouzel, J., & Dutay, J. C. (2000). Oxygen 18-salinity relationship simulated by an oceanic general circulation model. *Earth and Planetary Science Letters*, *178*(1–2), 113–123. https://doi.org/10.1016/S0012-821X(00)00073-X

Drobinski, P., Anav, A., Lebeaupin Brossier, C., Samson, G., Stéfanon, M., Bastin, S., Baklouti, M., Béranger, K., Beuvier, J., Bourdallé-Badie, R., Coquart, L., D'Andrea, F., de Noblet-Ducoudré, N., Diaz, F., Dutay, J.-C., Ethe, C., Foujols, M.-A., Khvorostyanov, D., Madec, G., … Viovy, N. (2012). Model of the Regional Coupled Earth system (MORCE): Application to process and climate studies in vulnerable regions. In *Environmental Modelling & Software*. https://doi.org/10.1016/j.envsoft.2012.01.017 http://dx.doi.org/10.1016/j.envsoft.2012.01.017

ElElla, A. (1993). *Preliminary studies on the geochemistry of the Nile river basin, Egypt*.

Gat, R. (1996). Oxygen and Hydrogen Isotopes in the Hydrologic Cycle. *AREPS*, *24*, 225–262. https://doi.org/10.1146/ANNUREV.EARTH.24.1.225

Kallel, N., & Labeyrie, L. (1997). Enhanced rainfall in the Mediterranean region during the last Sapropel Event. *Oceanologica Acta*, *20*(5), 697–7712. https://www.researchgate.net/publication/277157107

Kim, S. T., & O'Neil, J. R. (1997). Equilibrium and nonequilibrium oxygen isotope effects in synthetic carbonates. *Geochimica et Cosmochimica Acta*, *61*(16), 3461–3475. https://doi.org/10.1016/S0016-7037(97)00169-5

Laube-Lenfant, E. (1996). Utilisation des isotopes naturels #1#8o de l'eau et #1#3c du carbone inorganique dissous comme traceurs oceaniques dans les zones frontales et d'upwelling. Cas du pacifique equatorial et de la mer d'alboran [Paris 6]. In *http://www.theses.fr*. http://www.theses.fr/1996PA066229

Li, L., Bozec, A., Somot, S., Béranger, K., Bouruet-Aubertot, P., Sevault, F., & Crépon, M. (2006). Chapter 7 Regional atmospheric, marine processes and climate modelling. *Developments in Earth and Environmental Sciences*, *4*(C), 373–397. https://doi.org/10.1016/S1571-9197(06)80010-8

Lombard, F., Labeyrie, L., Michel, E., Bopp, L., Cortijo, E., Retailleau, S., Howa, H., & Jorissen, F. (2011). Modelling planktic foraminifer growth and distribution using an ecophysiological multi-species approach. *Biogeosciences*, *8*(4), 853–873. https://doi.org/10.5194/BG-8-853-2011

Ludwig, W., Dumont, E., Meybeck, M., & Heussner, S. (2009). River discharges of water and nutrients to the Mediterranean and Black Sea: Major drivers for ecosystem changes during past and future decades? *Progress in Oceanography*, *80*(3–4), 199–217. https://doi.org/10.1016/j.pocean.2009.02.001

Nixon, S. W. (2003). Replacing the Nile: Are Anthropogenic Nutrients Providing the Fertility Once Brought to the Mediterranean by a Great River? *AMBIO: A Journal of the Human Environment*, *32*(1), 30–39. https://doi.org/10.1579/0044-7447-32.1.30

Palmiéri, J., Orr, J. C., Dutay, J. C., Béranger, K., Schneider, A., Beuvier, J., & Somot, S. (2015). Simulated anthropogenic CO2 storage and acidification of the Mediterranean Sea. *Biogeosciences*, *12*(3), 781–802. https://doi.org/10.5194/BG-12-781-2015

Pierre, C. (1999). The oxygen and carbon isotope distribution in the Mediterranean water masses. *Marine Geology*, *153*(1–4), 41–55. https://doi.org/10.1016/S0025-3227(98)00090-5

Rebotim, A., Helga Luise Voelker, A., Jonkers, L., Waniek, J. J., Schulz, M., & Kucera, M. (2019). Calcification depth of deep-dwelling planktonic foraminifera from the eastern North Atlantic constrained by stable oxygen isotope ratios of shells from stratified plankton tows. *Journal of Micropalaeontology*, *38*(2), 113–131. https://doi.org/10.5194/JM-38-113-2019

Reverdin, G., Waelbroeck, C., Pierre, C., et al., 2022. The CISE-LOCEAN seawater isotopic database (1998–2021). Earth Syst. Sci. Data 14, 2721-2735, doi: 10.5194/essd-14-2721-2022. https://www.seanoe.org/data/00600/71186/

Rigual-Hernández, A. S., Sierro, F. J., Bárcena, M. A., Flores, J. A., & Heussner, S. (2012). Seasonal and interannual changes of planktic foraminiferal fluxes in the Gulf of Lions (NW Mediterranean) and their implications for paleoceanographic studies: Two 12-year sediment trap records. *Deep Sea Research Part I: Oceanographic Research Papers*, *66*, 26–40. https://doi.org/10.1016/J.DSR.2012.03.011

Risi, C., Bony, S., Vimeux, F., & Jouzel, J. (2010). Water-stable isotopes in the LMDZ4 general circulation model: Model evaluation for present-day and past climates and applications to climatic interpretations of tropical isotopic records. *Journal of Geophysical Research: Atmospheres*, *115*(D12), 12118. https://doi.org/10.1029/2009JD013255

Risi, C., Bony, S., Vimeux, F., Chongd, M., & Descroixe, L. (2010). Evolution of the stable water isotopic composition of the rain sampled along Sahelian squall lines. *Quarterly Journal of the Royal Meteorological Society*, *136*(S1), 227–242. https://doi.org/10.1002/QJ.485

Risi, C., Noone, D., Worden, J., Frankenberg, C., Stiller, G., Kiefer, M., Funke, B., Walker, K., Bernath, P., Schneider, M., Bony, S., Lee, J., Brown, D., & Sturm, C. (2012). Process-evaluation of tropospheric humidity simulated by general circulation models using water vapor isotopic observations: 2. Using isotopic diagnostics to understand the mid and upper tropospheric moist bias in the tropics and subtropics. *Journal of Geophysical Research: Atmospheres*, *117*(D5), 5304. https://doi.org/10.1029/2011JD016623

Risi, C., Ogée, J., Bony, S., Bariac, T., Raz-Yaseef, N., Wingate, L., Welker, J., Knohl, A., Kurz-Besson, C., Leclerc, M., Zhang, G., Buchmann, N., Santrucek, J., Hronkova, M., David, T., Peylin, P., & Guglielmo, F. (2016). The water isotopic version of the land-surface model ORCHIDEE: implementation, evaluation, sensitivity to hydrological parameters. *Hydrol Current Res*, *7*, 4. https://doi.org/10.4172/2157-7587.1000258

Roche, D., Paillard, D., Ganopolski, A., & Hoffmann, G. (2004). Oceanic oxygen-18 at the present day and LGM: equilibrium simulations with a coupled climate model of intermediate complexity. *Earth and Planetary Science Letters*, *218*(3–4), 317–330. https://doi.org/10.1016/S0012-821X(03)00700-3

Roether, W., Muennich, K. O., & Schoch, H. (2006). On the C-14 to tritium relationship in the North Atlantic Ocean. In *Radiocarbon* (Vol. 22, Issue 3, pp. 636–646). https://doi.org/10.2458/azu_js_rc.22.653

Sachse, D., Billault, I., Bowen, G. J., Chikaraishi, Y., Dawson, T. E., Feakins, S. J., Freeman, K. H., Magill, C. R., McInerney, F. A., van der Meer, M. T. J., Polissar, P., Robins, R. J., Sachs, J. P., Schmidt, H.-L., Sessions, A. L., White, J. W. C., West, J. B., Kahmen, A., Sachse, D., … Kahmen, A. (2012). Molecular Paleohydrology: Interpreting the Hydrogen-Isotopic Composition of Lipid Biomarkers from Photosynthesizing Organisms. *AREPS*, *40*(1), 221–249. https://doi.org/10.1146/ANNUREV-EARTH-042711-105535

Schmidt, G. A. (1998). Oxygen-18 variations in a global ocean model. *Geophysical Research Letters*, *25*(8), 1201–1204. https://doi.org/10.1029/98GL50866

Schmidt, G. A. (1999). Forward modeling of carbonate proxy data from planktonic foraminifera using oxygen isotope tracers in a global ocean model. *Paleoceanography*, *14*(4), 482–497. https://doi.org/10.1029/1999PA900025

Schroeder, K., Ribotti, a., Borghini, M., Sorgente, R., Perilli, a., & Gasparini, G. P. (2008). An extensive western Mediterranean deep water renewal between 2004 and 2006. *Geophysical Research Letters*, *35*, 1–7. https://doi.org/10.1029/2008GL035146

Vadsaria, T., Li, L., Ramstein, G., & Dutay, J. C. (2020). Development of a sequential tool, LMDZ-NEMO-med-V1, to conduct global-to-regional past climate simulation for the Mediterranean basin: an Early Holocene case study. *Geoscientific Model Development*, *13*(5), 2337–2354. https://doi.org/10.5194/GMD-13-2337-2020

van Breukelen, M. R., Vonhof, H. B., Hellstrom, J. C., Wester, W. C. G., & Kroon, D. (2008). Fossil dripwater in stalagmites reveals Holocene temperature and rainfall variation in Amazonia. *Earth and Planetary Science Letters*, *275*(1–2), 54–60. https://doi.org/10.1016/J.EPSL.2008.07.060

Vergnaud Grazzini, C. , G. C. , P. C. , P. C. , and U. M. J. (1986). Foraminifères planctoniques de Méditerranée en fin d'été. Relations avec les structures hydrologiques,. *Mem. Soc. Geol. Ital*, *36*, 175–188.
Voelker, A. H. L., Colman, A., Olack, G., Waniek, J. J., & Hodell, D. (2015). Oxygen and hydrogen isotope signatures of Northeast Atlantic water masses. *Deep Sea Research Part II: Topical Studies in Oceanography*, *116*, 89–106. https://doi.org/10.1016/J.DSR2.2014.11.006

Voelker, A.H., 2023. Seawater oxygen and hydrogen stable isotope data from the upper water column in the North Atlantic Ocean (unpublished data). Interdisciplinary Earth Data Alliance (IEDA), doi: https://doi.org/10.26022/IEDA/112743

Voelker, A.H.L., Colman, A., Olack, G., Waniek, J.J., Hodell, D., 2015. Oxygen and hydrogen isotope signatures of Northeast Atlantic water masses. Deep Sea Research Part II: Topical Studies in Oceanography 116, 89-106, doi: 1016/j.dsr2.2014.11.006.

Voelker, Antje H L (2017): Seawater oxygen isotopes for Station POS334-73, Alboran Sea. Instituto Portugues do Mar e da Atmosfera: Lisboa, Portugal, PANGAEA, https://doi.org/10.1594/PANGAEA.878063

Voelker, Antje H L; Colman, Albert Smith; Olack, Gerard; Waniek, Joanna J; Hodell, David A (2015): Oxygen and hydrogen isotopes measured on water bottle samples during EUROFLEETS cruise Iberia-Forams. PANGAEA, https://doi.org/10.1594/PANGAEA.831462

Vörösmarty, C. J., Fekete, B. M., & Tucker, B. A. (1996). Global River Discharge Database (RivDIS V1.0), International Hydrological Program. *Global Hydrological Archive and Analysis Systems, UNESCO, Paris*.

---

## Author Comment (AC3)

**Manuscript " Modelling the water isotopes distribution in the Mediterranean Sea using a high-resolution oceanic model (NEMO-MED12-watiso-v1.0): Evaluation of model results against in-situ observations"**

Mohamed Ayache[1], Jean-Claude Dutay[1], Anne Mouchet[2], Kazuyo Tachikawa[3], Camille Risi[4], and Gilles Ramstein[1]

[1]Laboratoire des Sciences du Climat et de l'Environnement, CEA-CNRS-Université Paris Saclay, 91191, Gif-sur-Yvette, France
[2]Freshwater and OCeanic science Unit of reSearch (FOCUS), Université de Liège, B-4000 Liège
[3]Aix Marseille Univ, CNRS, IRD, INRAE, Coll France, CEREGE, 13545, Aix-en-Provence, France
[4]Laboratoire de Météorologie Dynamique, IPSL, CNRS, Sorbonne Université, Paris, France Correspondence: Mohamed Ayache (mohamed.ayache@lsce.ipsl.fr)

**Reply to reviewers' comments**

Dear Pr. I., Andrew Yool

We would like to thank you for providing us the opportunity to revise our manuscript, and we are extremely grateful to Pr. Antje Voelker, Pr. Allegra N. LeGrande and the anonymous reviewer for their careful reading and comments that helped to improve our manuscript significantly.

We have revised our manuscript and provided a detailed response to each reviewer's comment and request below.

**Color code**
Reviewer comments
Authors response
The modifications performed in the manuscript appear in red above and in the revised manuscript with Changes Marked.

**Anonymous Referee #2**
**General comment**
Ayache et al. present the implementation of stable water isotopes (d18O and dD) in the high-resolution regional ocean model NEMO-MED12. The simulation of such isotope proxies in climate models is very useful for past climate reconstruction and to better understand climate processes recorded in the water cycle. Ayache et al. performed a simulation for present-day conditions and evaluate their results with available isotopic observations in seawater and marine calcite. They also investigate the relationship of isotopes with salinity. There are not so many studies on isotope modeling in the ocean, even more in a regional model. Moreover, the Mediterranean Sea is interesting in several points of view: many data, a strong east-west contrast in oceanic evaporation, a relatively short residence time… The article is easy to follow, and the analyses are sound. The figures could be improved, especially the used color scales, and some details on the description of the simulation are missing. Moreover, the discussion section is not really a discussion, yet, but more

a summary of the results. After addressing these minor points, detailed below, the article of Ayache et al. could be published in GMD.

We extend our thanks to Reviewer #2 for their valuable comments and suggestions, which have contributed to clarifying the manuscript, reinforcing our arguments, and enhancing the main message we aim to convey. The majority of the comments have been incorporated into the revised version.

**Specific comments (rather minor revisions)**

- Some details on the simulation are missing. Especially, what is spinup time? How was it performed? On line 154, it is said that LMDZ-iso simulation outputs for the period 1990-2020 were used as isotope boundary conditions? What does it mean exactly? That the authors performed a simulation 30 years between 1990 and 2020 with the forcings of the corresponding year? Or that the authors used a climatological average of the LMDz-iso 1990-2020 simulation as boundary conditions, in order to perform a simulation of several decades (so with the same conditions all along the simulation)? What does it involve in terms of bias in isotopic modeled results compared to the observations?

Thank you for pointing this out to us, and we agreed on the importance of clarifying our experimental design.

Here, we used the offline coupling mode. In this method, the physical variables i.e., the circulation fields (U, V, W) and mixing coefficients (Kz) are previously computed by the NEMO-MED12 dynamical model for the 1958–2013 period (Beuvier et al., 2012a) and used to propagate the passive tracers in the ocean.

The simulation was run during 30 years after 44 years of spin-up (1958–1980 repeated two times) allowing us to stabilize the model state (for more than 75 of run). The hydrodynamic forcing has been built from a random draw of year among the historical period (1958–2013 period, Beuvier et al., 2012a) to minimize the impact of the intense events of variability like the EMT or the WMT (Roether et al., 2006; Schroeder et al., 2008). The spin-up strategy was adapted in our previous passive tracer simulations (e.g. tritium and neodymium: Ayache et 2015a, 2016).

The isotopic simulation is performed using outputs from the global atmospheric model LMDZ-iso (Risi et al., 2010b) with an AMIP (Atmospheric Model Intercomparison Project) simulation from 1990 to 2020. The aim is to assess the model's performance in the present climate and against in-situ data observed randomly over the historical period. Therefore, we have opted to use the climatological mean of the LMDZ-iso 1990-2020 simulation as boundary conditions. This decision was made to minimize the warming trend during this period and to ensure that the precipitation and evaporation simulated by the LMDZ-iso model for the current climate situation are as close to the average state as possible, with minimal impact from inter-annual variability.

These points are clarified in the revised manuscript (see section 2.2, lines 149-154).

"The simulation was run for 30 years after 44 years of spin-up (1958–1980 repeated two times) allowing us to stabilize the model state (for more than 75 of run). The hydrodynamic forcing has been built from a random draw of the year among the historical period (1958–2013 period, Beuvier et al. (2012a) to minimize the impact of the intense events of variability like the EMT

or the WMT (Roether et al., 2006; Schroeder et al., 2008). The spin-up strategy was adapted in our previous passive tracer simulations (e.g. neodymium and tritium Ayache et al. (2015a, 2016))"

- Still about the experimental design, one important aspect for the calculation of d18Ocalcite from modeled d18Osw is the forcing for surface temperature conditions. As said by the authors, the surface temperature conditions do not come from LMDZ-iso but from an ERA-40 relaxation term applied to the ARPERA heat flux. It means that inconsistencies between d18O of freshwater fluxes and temperature are possible. Could the authors elaborate on this aspect? Could they evaluate the potential biases on the ocean temperature and so on the modeled d18Ocalcite?

Thank you for pointing this out. The reviewer is correct, the forcing of surface temperature used in the calculation of $\delta^{18}O_c$ does not come from LMDZ-iso but from an ERA-40 relaxation term applied to the ARPERA heat flux. This is certainly among the limitations of the OFFLINE coupling mode with the use of a pre-calculated dynamical field.

We presented horizontal temperature maps used in calculating $\delta^{18}O_c$ (refer to Fig. 10c). We checked that this simulation produced reasonable temperature patterns in the Mediterranean Sea against observations. A notable difference arises when comparing the $\delta^{18}O_c$ calculated with high-resolution simulated temperatures (cf. Fig. 3a and 3b below) to that derived from a global model temperature from a global version forced by LMDZ (cf. Fig. 3c and 3d). The global model shows a significant bias in $\delta^{18}O_c$ as a consequence of low temperatures simulated in the Mediterranean Sea (cf. Fig. 3c and 3d).

Additionally, in this simulation, we employed the same freshwater forcing (from Ludwig et al., 2009, and the RivDis dataset, Vörösmarty et al. 1996) as that used in the dynamical simulation (in Beuvier et al., 2012) where the temperature was simulated, ensuring complete consistency between freshwater flux and temperature. This validates our decision to utilize temperatures simulated by the MED12 model and forced by ERA5 rather than the global LMDZ model (see Fig.3 below). However, this inconsistency requires further investigation within a fully coupled ocean-atmosphere model to ensure consistent simulation of changes across various model components.

These points are clarified in the revised manuscript. see Appendix B and the following text (lines 341-343 in the track changes version).
"In this study, we analysed the impact of temperature on $\delta^{18}O_c$ calculations, both in a global model and at high regional resolution. Please refer to Appendix B for further details."

[Figure]

*Figure 3 comparing the $\delta^{18}O_c$ calculated with high-resolution simulated temperature (cf. panel a and b) to that derived from a global model using temperature data from LMDZ (c and d).*

- The discussion section is not really a discussion but more a summary of the results at the current state of the paper. Here are some topics the authors can discuss: How are the results NEMO-MED-wiso compared to global ocean models or coupled models? Are they improved thanks to the high resolution of NEMO-MED-wiso? The authors talk about coupling as a perspective, but what is possible to do with this model given that it is a regional model, not a global one? How can it bring new useful insights for paleoclimate applications except by putting as boundary forcings the atmospheric fields from paleoclimate global simulations (i.e., offline)? Can this model be used to improve global climate models? The seasonality aspect on d18Ocalcite is interesting, could you elaborate more on this aspect?

Thank you! We think that these points raised by the reviewer are very important to enrich the discussion and the perspectives of this work. We have included these various points in the new version of our paper:

**How are the results NEMO-MED-wiso compared to global ocean models or coupled models?**

The model's high resolution presents a unique opportunity to represent a realistic thermohaline circulation in the Mediterranean basin, thus enabling a better understanding of the processes governing water isotopic distribution within this intercontinental basin.

We initially discussed the potential impact of model resolution in the submitted version of our paper (refer to lines 337-341). Additionally, within our team, we utilize a low-resolution global NEMO model (ORCA 1° and 2° horizontal resolution). Figure 4 (below) provides a comparison

between the results of the global model (ORCA2) and NEMO-MED12 model, using the same water isotopes modeling approach and forced by the same atmospheric model LMDZiso (R96).

The figure demonstrates that the global model produces unrealistically high values of $\delta^{18}O_w$ in the Mediterranean Sea, particularly in the eastern basin ($\delta^{18}O_w > 2$, max 3.3), whereas in-situ data show maximum values of around 2.1 (Gat et al., 1996). This comparison with the global model has been incorporated in the revised version to complement the discussion on the high-resolution impact, as suggested by the reviewer (Added in Appendix C).

"Sensitivity tests were performed to investigate the effect of changing the resolution of the LMDZiso atmospheric model (between R96 and R144) and the oceanic model (between ORCA2 and NEMO-MED12), the results of which are presented in the supplementary material of this paper (see Appendix C)"

See section 3.4 lines 376-389.

[Figure]

*Figure 4 Comparison between the δ18Ow results of the global model (ORCA2 ~2° of resolution) and NEMO-MED12 model, using the same water isotopes modeling approach and forced by the same atmospheric model LMDZiso (R96).*

**The authors talk about coupling as a perspective, but what is possible to do with this model given that it is a regional model, not a global one?, #Can this model be used to improve global climate models?**

So far, water isotopes have been implemented separately in all components of the IPSL general circulation model (the atmospheric "LMDZiso", soil-vegetation "ORCHIDEEiso" and oceanic "NEMOiso"), but a fully-coupled, isotope-enabled version of the IPSL-GCM is still lacking. A

fully coupled simulation will allow us to better understand the feedback and non-linear aspects of the evolution of the water cycle, and hence provide a unique tool for better constraining the past climates simulated in climate models. There is currently a project at IPSL to update the water isotope code in the different components to prepare the isotope-enabled fully coupled version.

Using these models in the Mediterranean region provides a great opportunity to test this water isotope package in a basin where evaporation varies significantly from east to west with a relatively short residence time and much available data. Furthermore, there is no effect of sea ice formation or melting (i.e. no freshwater input from ice sheets during the recent "present situation" period) which is currently not well represented in models. This allows a better understanding of the relative roles of the different parameters within the model and provides a unique opportunity to understand better the spatial and temporal variations of water isotopes for which strict conservation is desirable. Additionally, the water isotope modeling package presented in this study can be utilized in coupled regional configurations of the Mediterranean region, such as regIPSL (refer to https://sourcesup.renater.fr/wiki/morcemed/), which will undoubtedly aid in the preparation of a global-scale coupled version.

We are currently implementing the same water isotopes package, as presented in section 2, into the new global version of NEMO (NEMOv4.2 at ORCA 2° and ORCA 1° of horizontal resolution). This work is aiding us significantly in refining the parameterization of the NEMO global model, particularly in representing runoff forcing. The Mediterranean Sea offers more constrained runoff data/models compared to the global scale, providing insights into the impact of surface runoff. Our sensitivity tests on the influence of the Po River in the Mediterranean Sea, including distributing the Po water discharge across the first vertical levels of the model to prevent numerical instability, have enhanced our understanding. This experience has also enabled us to improve our representation of the Amazon River's discharge in the global version. It's also important to note that the implementation and effectiveness of such a coupling would likely require further research and validation.

**How can it bring new useful insights for paleoclimate applications except by putting as boundary forcings the atmospheric fields from paleoclimate global simulations (i.e., offline)?**

In paleoclimate studies, one major problem with the simulation of past climate changes is that forcings/boundary conditions are not available from observations or data reconstruction to drive high-resolution regional models.

The coupled configuration will make it possible to study past climate for a wide range of periods (i.e. transient simulations) with good confidence, to characterise quantitatively past variations in the isotopic composition of water, and to allow direct comparison between isotopic signals obtained from models and various archives (ice cores, speleothems, oceanic sediment cores, etc.), which is not possible using the offline coupling mode.

Regional climate models can bridge the gap between the coarse resolution of global climate models and the regional-to-local scales. They can provide a more realistic representation of physical processes and climate feedback compared to global climate models. This is particularly true for the Mediterranean region with complex geology. In particular, atmospheric circulation (high wind gusts in winter) and oceanic circulation (deep convection) are better represented in regional models (Ludwig et al., 2019). Also, a numerical platform of global-to-regional modeling has been developed by Vadsaria et al., (2020). This sequential platform may be applied to a large number of paleoclimate contexts from the Quaternary to the Pliocene with

regional model forced by a global model. This can be useful for paleoclimate applications, as it can help to answer fundamental paleoclimate research questions and may be key to advancing a meaningful joint interpretation of climate model and proxy data (Ludwig et al., 2019).

We have included additional sentences to better clarify this point in the revised manuscript (see section 4, lines 453-461)

"Regional climate models can bridge the gap between the coarse resolution of global climate models and the regional-to-local scales. They provide a more realistic representation of physical processes and climate feedback compared to global climate models. This is especially true for the Mediterranean region with its complex geology (Li et al., 2006). The water isotope modelling package presented in this study can be used in coupled regional configurations, such as regIPSL (Drobinski et al. (2012), which may assist in the preparation of a global-scale coupled version. Additionally, a sequential architecture of a global-regional modelling platform has been developed by Vadsaria et al., (2020) using the same dynamical model NEMO-MED. This platform can be used sequentially in a wide range of paleoclimate contexts, from the Quaternary to the Pliocene, with a regional model that is forced by a global model."

**The seasonality aspect on d18Ocalcite is interesting, could you elaborate more on this aspect?**

Calcite $\delta^{18}O_c$ is widely used in paleoclimate research. Understanding its seasonal variability is crucial for reconstructing past climates. The influence of seasonal temperature variability on $\delta^{18}O_c$ (equation 6) is important, particularly in the Mediterranean Sea because of marked seasonal thermal contrast. The $\delta^{18}O_c$ values are determined by both $\delta^{18}O_w$ and the seawater temperature at the calcification depth. For planktonic foraminifera such as *Globigerinoides ruber* and *Globigerina bulloides*, the calcification depth typically ranges from 0 to 100 meters, though variations exist depending on the basin (De Castro Coppa et al., 1980; Grazzini et al., 1986). The season of maximal foraminiferal production can be estimated by data from sediment traps. For instance, *G. ruber* and *G. bulloides* have been associated with calcification seasons in October-November and April-May according to Kallel et al. (1997), while others suggest January-March (Avnaim-Katav et al., 2019) and February-April (Rigual-Hernandez et al., 2012).

In this context, we used our model results to explore the relationship between $\delta^{18}O_c$ and temperature. We employed a paleotemperature equation for inorganic calcite by Kim and O'Neil (1997), which was modified by Bemis et al. (1998), as shown in Fig. 10. Our simulations indicate that the highest $\delta^{18}O_c$ values occur during winter (February, March), while the lowest values are observed during summer/autumn. Although the available observational data do not cover all months of the year, our results align with existing data, highlighting the significant influence of temperature on $\delta^{18}Oc$ in the Mediterranean Sea. Nonetheless, a dedicated study should be conducted to further elucidate the seasonal aspect.

In the revised version of our paper, we have included additional sentences to provide clarity on the seasonality aspect of $\delta^{18}O_c$ (see section 4 lines 427-441).

"Calcite $\delta^{18}Oc$ is widely used in paleoclimate research. Understanding its seasonal variability is crucial for reconstructing past climates. The influence of seasonal temperature variability on

δ^18^O_c (equation 6) is important, particularly in the Mediterranean Sea because of marked seasonal thermal contrast. The $\delta^{18}O_c$ values are determined by both $\delta^{18}O_w$ and the seawater temperature at the calcification depth. For planktonic foraminifera such as *Globigerinoides ruber* and *Globigerina bulloides*, the calcification depth typically ranges from 0 to 100 meters, though variations exist depending on the basin (De Castro Coppa et al., 1980; Grazzini et al., 1986). The season of maximal foraminiferal production can be estimated by data from sediment traps. For instance, *G. ruber* and *G. bulloides* have been associated with calcification seasons in October-November and April-May according to Kallel et al. (1997), while others suggest January-March (Avnaim-Katav et al., 2019) and February-April (Rigual-Hernandez et al., 2012).

In this context, we used our model results to explore the relationship between the $\delta^{18}O_c$ and temperature. We employed a paleotemperature equation for inorganic calcite by Kim and O'Neil (1997), modified by Bemis et al. (1998), as shown in Fig. 10. Our simulations indicate that the highest $\delta^{18}O_c$ values occur during winter (February, March), while the lowest values are observed during summer/autumn. Although the available observational data do not cover all months of the year, our results align with existing data, highlighting the significant influence of temperature on $\delta^{18}Oc$ in the Mediterranean Sea. Nonetheless, a dedicated study should be conducted to further elucidate the seasonal aspect."

- The green-to-red colormap used in several figures is not appropriate for colorblind people and should be changed.

Thank you for pointing this out. The same point has been raised by the first reviewer. In the new version of the paper, the colour palettes have been changed.

- The d18Ocalcite dataset is not described in the method section (section 2.5).

The $\delta^{18}O_c$ data were recalculated employing present-day $\delta^{18}O_w$ and temperature data cited in this paper (in section 2.5). We utilized the same paleotemperature equation applied to model outputs, as described by Kim (1997) and further refined by Bemis (1998) for inorganic calcite. Comparison with real paleo data was not conducted as our simulations and their associated forcings were designed for the present-day situation; a specific paleo simulation was not undertaken in this study.

Clarified in the revised ms (see section 3.4, line 325)

"The equation was applied to both the model output and the available in-situ data, as presented in Section 2.5"

- The difference between R96 and R144 is described very briefly. To show the difference map between R144 and R96 for both d18Osw and applied isotope freshwater fluxes could help to understand better what does (not) happen. Could the remapping from LMDZ-iso grid to NEMO-MED-wiso one partly explain this non-diff erence? See technical comments below.

We have performed some sensitivity tests of the results by changing the horizontal resolution of LMDZ-iso between R96 and R144. The results are very close to each other as shown in Fig. 6. It is possible that there is a certain threshold of spatial resolution below which the simulation

is improved by a finer resolution. Vadsaria et al. (2020) showed that high resolution (~ 30 km of the atmospheric model with a more realistic wind pattern and hydrological cycle) is critical to accurately capture the synoptic variability needed to initiate the formation of the intermediate and deep waters of the Mediterranean thermohaline circulation (Li et al., 2006). Therefore, we decided to work with the R96 resolution which is the least expensive.

Following the suggestions made by the reviewers, we have included a figure 5 below showing the $\delta^{18}O_w$ anomaly map between the two simulations R144 and R96. The difference between these simulations is minimal, ranging between -0.2 and +0.2 ‰. One possible explanation for this slight difference lies in the runoff forcing utilized. As explained in the manuscript, the runoff forcing is derived from data by Ludwig et al. (2009) rather than from LMDZiso. This is because the water flows simulated by LMDZiso are unrealistic in the Mediterranean basin (e.g., LMDZiso significantly overestimates the Nile river discharge).

To sum up, the change of horizontal resolution between R144 and R96 is not sufficient to generate drastic changes in evaporation and precipitation (as suggested by Vadsaria et al., 2020). The fact that the same runoff forcing was used in both the R96 and R144 simulations explains the small difference between these two simulations.

We have moved Figure 6 to the supplementary materials because this change in resolution does not significantly impact our results and could potentially dilute our main message, and the following text was added in the revised version

"Sensitivity tests were performed to investigate the effect of changing the resolution of the LMDZiso atmospheric model (between R96 and R144) and the oceanic model (between ORCA2 and NEMO-MED12), the results of which are presented in the supplementary material of this paper (see Appendix C)."

 (see lines 376-379 and appendix C).

[Figure]

[Figure]

*Figure 5 Distribution of $\delta^{18}O_w$ (in per mil) in surface water (at a depth of 50 m) from the R96 simulation. Colored dots represent in-situ observations compiled from Epstein and Mayeda (1953), Stahl and Rinow (1973), Pierre et al. (1986), Gat et al. (1996), and Pierre (1999). Panel d) presents a multi-scatter plot comparing simulated $\delta^{18}O_w$ (averaged over the last 30 years of the simulation) from the R96 simulation with in-situ data from the mentioned sources across the entire basin. The color code indicates the latitudes of the data in degrees east. Panels b) and e) depict the same as panels a) and d), respectively, but from the R144 simulation. Panel c) illustrates the $\delta^{18}O_w$ anomaly map between the R144 and R96 simulations in surface water*

**Minor technical comments:**

- Line 14: O is missing in d18O.

  Corrected

- Line 16: (d18O-S relationship) can be removed.

  Done

- Line 40: "high resolution regional ocean model, yet.".

  Added

- Line 57: Replace that by which.

  Replaced

- Line 67: remove "as an oceanographic tracer".

  Removed

- Line 76: We use isotope fluxes from…

Changed

- Line 110: The term isotopologue should be used at the beginning of the paper (line 14). Then you can say you use the term isotope instead.

Agreed

- Line 110: high-resolution

Corrected

- Line 121: replace bouquin AIEA by the appropriate IAEA reference.

Changed

- Lines 142-143: Table S2 are…

Corrected

- Section 2.3: see major comment about spin-up time and simulation length.

In the revised version of the paper, more information has been added about our experimental design (see new section 2.2).

- Line 157: remove Risi et al., 2010b.

Done

- Section 2.5: see major comment about the description of d18Ocalcite dataset.

The information was added in the revised version

- Line 230: pseudo-salinity results or standard modeled salinity?

Pseudo-salinity (corrected)

- Section 2.3: please change salinity by pseudo-salinity where needed to avoid misunderstanding between the modeled standard salinity of NEMO-MED and the pseudo-salinity described in this paper. Change salinity by pseudo-salinity in the title too.

Thank you for pointing this out. Changed in the revised version.

- Line 238: spatial slope?

Indeed. Added

- Line 241: between observed salinity and d18Osw…

Added

- Lines 259-265: this part has nothing to do with the d18O-pseudo salinity relationship. It should be removed, except if you can show a change in the relationship when using R96 or R144 LMDz-iso fields.

Thank! We have added a comparison between standard simulated salinity and pseudo-salinity (see new Appendix D)

Overall, the pseudo-salinity globally yields values highly comparable to standard simulated salinity. Minor deviations are noticed in the Gulf of Lions and the Algerian

Basin, attributed to overlooked mesoscale activity impacts in the global LMDZiso simulation.

- Section 3.3: I think the part on dD can be removed. It's similar to d18o and there are not so many data. Figure 7 should be removed too.

We agree with the reviewer that $\delta D_w$ and $\delta^{18}O_w$ tendencies are similar since identical boundary fluxes (precipitation, evaporation, and river runoff) drive both $\delta^{18}O_w$ and $\delta D_w$ isotopes in the surface water. However, as this is a development paper, we have included the deuterium results to show that the code exists and can be used by the scientific community. Besides, simulating and evaluating both $\delta D_w$ and $\delta^{18}O_w$ is necessary for the perspective of the future coupling of NEMO with the other components of the IPSL model, which will be used for paleoclimate applications involving both $\delta D$ and $\delta^{18}O$ of natural archives. In particular, $\delta D$ in leaf waxes (Sachse et al 2012) and speleothem fluid inclusions (van Breukelen et al 2008) are useful for paleoclimate reconstructions.

We have clarified this point in the article: "Simulating both $\delta D_w$ and $\delta^{18}O_w$ is useful for paleoclimate applications involving both $\delta D$ and $\delta 18O$ of natural archives, particularly when using this modelling approach in a fully coupled configuration. Notably, $\delta D$ in leaf waxes (Sachse et al., 2012) and speleothem fluid inclusions (van Breukelen et al., 2008) are useful for paleoclimate reconstructions."

See section 4, lines 418-420 in track changes version

- Lines 266-277: can be removed.

We prefer keeping the $\delta D$ result, because of its relevance for paleoclimate reconstructions as explained above.

- Line 278: remove (d-excess= $\delta D$ - 8* $\delta 18Osw$, Dansgaard, 1964) as you already said it at the beginning of the paper.

Done

- For the Figure 9, it could be interesting to see the depth profile of d-excess too. Are there some data to compare with in EMed (according to Figure 7)? Then you could maybe elaborate a little bit more for the section 3.3.

Thanks to the suggestion of the first reviewer we have found new dD and d-excess data in the western basin (Reverdin et al., 2022) and we have added these data to Fig. 7. The data have allowed us to replot Fig. 9 and enrich the discussion of the section 3.3.

See the new figure below and the new section 3.3 in the revised version.

[Figure]

*Figure 6 The model outputs against in-situ data for the present-day situation. a) d-excess (in ‰) distribution in the surface water (50 m depth). b) E-W vertical section of d-excess (in ‰) in the western Mediterranean basin d) Zonal mean comparison of d-excess (in ‰) average vertical profiles in the western basin presenting model results against in-situ data. c) and e) the same as b) and d) but for the eastern basin. Colour filled dots represent in-situ observations from (Gat et al., 1996; Reverdin et al., 2022). Both model and in-situ data use the same colour scale.*

- Figure 1: typo in "Precipitation" in plots d, e, and f. Also in the legend of the figure.

  Done

- Figure 6: could you show the difference R144-R96 in the d18Osw, but also in the applied isotope freshwater fluxes. It could help to understand the little difference between the two simulations and to elaborate a little bit more (for now, the results are described in 6 lines at the wrong place (lines 259-265).

  In the revised version of our paper, the text (lines 259-265) and Figure 6 has been moved to the supplementary materials (Appendix C) as this change in the resolution doesn't significantly affect our results and could potentially obscure the main message (see Figure below).

  For an explanation of the slight variance between R144 and R96, please refer to our answer to specific comments of the reviewer #2.

- Figure A1: Apply different scales for EMed and WMed d18Osw.

  Done

- Legend of figure A2: average vertical profiles.

Done

**References**

Avnaim-Katav, S., Almogi-Labin, A., Schneider-Mor, A., Crouvi, O., Burke, A. A., Kremenetski, K. v., & MacDonald, G. M. (2019). A multi-proxy shallow marine record for Mid-to-Late Holocene climate variability, Thera eruptions and cultural change in the Eastern Mediterranean. *Quaternary Science Reviews*, *204*, 133–148. https://doi.org/10.1016/J.QUASCIREV.2018.12.001

Ayache, M., Dutay, J.-C., Arsouze, T., Révillon, S., Beuvier, J., & Jeandel, C. (2016). High-resolution neodymium characterization along the Mediterranean margins and modelling of Nd distribution in the Mediterranean basins. *Biogeosciences*, *13*(18). https://doi.org/10.5194/bg-13-5259-2016

Ayache, M., Dutay, J.-C., Jean-Baptiste, P., Beranger, K., Arsouze, T., Beuvier, J., Palmieri, J., Le-Vu, B., & Roether, W. (2015). Modelling of the anthropogenic tritium transient and its decay product helium-3 in the Mediterranean Sea using a high-resolution regional model. *Ocean Science*, *11*(3). https://doi.org/10.5194/os-11-323-2015

Ayache, M., Dutay, J.-C., Mouchet, A., Tisnérat-Laborde, N., Montagna, P., Tanhua, T., Siani, G., & Jean-Baptiste, P. (2017). High-resolution regional modelling of natural and anthropogenic radiocarbon in the Mediterranean Sea. *Biogeosciences*, *14*(5). https://doi.org/10.5194/bg-14-1197-2017

Bemis, B. E., Spero, H. J., Bijma, J., & Lea, D. W. (1998). Reevaluation of the oxygen isotopic composition of planktonic foraminifera: Experimental results and revised paleotemperature equations. *Paleoceanography*, *13*(2), 150–160. https://doi.org/10.1029/98PA00070

Benetti, M., Reverdin, G., Aloisi, G., Sveinbjörnsdóttir, Á., 2017. Stable isotopes in surface waters of the Atlantic Ocean: Indicators of ocean-atmosphere water fluxes and oceanic mixing processes. Journal of Geophysical Research: Oceans 122, 4723-4742, doi: 1002/2017JC012712.

Benetti, M., Reverdin, G., Pierre, C., Merlivat, L., Risi, C., Steen-Larsen, H. C., & Vimeux, F. (2014). Deuterium excess in marine water vapor: Dependency on relative humidity and surface wind speed during evaporation. *Journal of Geophysical Research: Atmospheres*, *119*(2), 584–593. https://doi.org/10.1002/2013JD020535

Beuvier, J., Béranger, K., Lebeaupin Brossier, C., Somot, S., Sevault, F., Drillet, Y., Bourdallé-Badie, R., Ferry, N., & Lyard, F. (2012). Spreading of the Western Mediterranean Deep Water after winter 2005: Time scales and deep cyclone transport. *Journal of Geophysical Research*, *117*(C7), C07022. https://doi.org/10.1029/2011JC007679

Craig, H., & Gordon, L. (1965). *Deuterium and oxygen 18 variations in the ocean and the marine atmosphere, in: Proc. Stable Isotopes in Oceanographic Studies and Paleotemperatures* (E. Tongiogi & F. Lishi, Eds.; pp. 9–130).

Dansgaard, W. (1964). Stable isotopes in precipitation. *Tellus*, *16*, 468–468.

de Castro Coppa, M. G. , M. Z. M. , P. B. , S. F. and T. R. E. ,. (1980). Distributione stagionale e verticale dei foraminiferi planctonici del golfo di Napoli. *Bol. Soc. Nat*, *89*, 1–25.

Delaygue, G., Bard, E., Rollion, C., Jouzel, J., Stiévenard, M., Duplessy, J. C., & Ganssen, G. (2001). Oxygen isotope/salinity relationship in the northern Indian Ocean. *Journal of Geophysical Research: Oceans*, *106*(C3), 4565–4574. https://doi.org/10.1029/1999JC000061

Delaygue, G., Jouzel, J., & Dutay, J. C. (2000). Oxygen 18-salinity relationship simulated by an oceanic general circulation model. *Earth and Planetary Science Letters*, *178*(1–2), 113–123. https://doi.org/10.1016/S0012-821X(00)00073-X

Drobinski, P., Anav, A., Lebeaupin Brossier, C., Samson, G., Stéfanon, M., Bastin, S., Baklouti, M., Béranger, K., Beuvier, J., Bourdallé-Badie, R., Coquart, L., D'Andrea, F., de Noblet-Ducoudré, N., Diaz, F., Dutay, J.-C., Ethe, C., Foujols, M.-A., Khvorostyanov, D., Madec, G., … Viovy, N. (2012). Model of the Regional Coupled Earth system (MORCE): Application to process and climate studies in vulnerable regions. In *Environmental Modelling & Software*. https://doi.org/10.1016/j.envsoft.2012.01.017
http://dx.doi.org/10.1016/j.envsoft.2012.01.017

ElElla, A. (1993). *Preliminary studies on the geochemistry of the Nile river basin, Egypt*.

Gat, R. (1996). Oxygen and Hydrogen Isotopes in the Hydrologic Cycle. *AREPS*, *24*, 225–262. https://doi.org/10.1146/ANNUREV.EARTH.24.1.225

Kallel, N., & Labeyrie, L. (1997). Enhanced rainfall in the Mediterranean region during the last Sapropel Event. *Oceanologica Acta*, *20*(5), 697–7712. https://www.researchgate.net/publication/277157107

Kim, S. T., & O'Neil, J. R. (1997). Equilibrium and nonequilibrium oxygen isotope effects in synthetic carbonates. *Geochimica et Cosmochimica Acta*, *61*(16), 3461–3475. https://doi.org/10.1016/S0016-7037(97)00169-5

Laube-Lenfant, E. (1996). Utilisation des isotopes naturels #1#8o de l'eau et #1#3c du carbone inorganique dissous comme traceurs oceaniques dans les zones frontales et d'upwelling. Cas du pacifique equatorial et de la mer d'alboran [Paris 6]. In *http://www.theses.fr*. http://www.theses.fr/1996PA066229

Li, L., Bozec, A., Somot, S., Béranger, K., Bouruet-Aubertot, P., Sevault, F., & Crépon, M. (2006). Chapter 7 Regional atmospheric, marine processes and climate modelling. *Developments in Earth and Environmental Sciences*, *4*(C), 373–397. https://doi.org/10.1016/S1571-9197(06)80010-8

Lombard, F., Labeyrie, L., Michel, E., Bopp, L., Cortijo, E., Retailleau, S., Howa, H., & Jorissen, F. (2011). Modelling planktic foraminifer growth and distribution using an ecophysiological multi-species approach. *Biogeosciences*, *8*(4), 853–873. https://doi.org/10.5194/BG-8-853-2011

Ludwig, W., Dumont, E., Meybeck, M., & Heussner, S. (2009). River discharges of water and nutrients to the Mediterranean and Black Sea: Major drivers for ecosystem changes during past and future decades? *Progress in Oceanography*, *80*(3–4), 199–217. https://doi.org/10.1016/j.pocean.2009.02.001

Nixon, S. W. (2003). Replacing the Nile: Are Anthropogenic Nutrients Providing the Fertility Once Brought to the Mediterranean by a Great River? *AMBIO: A Journal of the Human Environment*, *32*(1), 30–39. https://doi.org/10.1579/0044-7447-32.1.30

Palmiéri, J., Orr, J. C., Dutay, J. C., Béranger, K., Schneider, A., Beuvier, J., & Somot, S. (2015). Simulated anthropogenic CO2 storage and acidification of the Mediterranean Sea. *Biogeosciences*, *12*(3), 781–802. https://doi.org/10.5194/BG-12-781-2015

Pierre, C. (1999). The oxygen and carbon isotope distribution in the Mediterranean water masses. *Marine Geology*, *153*(1–4), 41–55. https://doi.org/10.1016/S0025-3227(98)00090-5

Rebotim, A., Helga Luise Voelker, A., Jonkers, L., Waniek, J. J., Schulz, M., & Kucera, M. (2019). Calcification depth of deep-dwelling planktonic foraminifera from the eastern North Atlantic constrained by stable oxygen isotope ratios of shells from stratified plankton tows. *Journal of Micropalaeontology*, *38*(2), 113–131. https://doi.org/10.5194/JM-38-113-2019

Reverdin, G., Waelbroeck, C., Pierre, C., et al., 2022. The CISE-LOCEAN seawater isotopic database (1998–2021). Earth Syst. Sci. Data 14, 2721-2735, doi: 10.5194/essd-14-2721-2022. https://www.seanoe.org/data/00600/71186/

Rigual-Hernández, A. S., Sierro, F. J., Bárcena, M. A., Flores, J. A., & Heussner, S. (2012). Seasonal and interannual changes of planktic foraminiferal fluxes in the Gulf of Lions (NW Mediterranean) and their implications for paleoceanographic studies: Two 12-year sediment trap records. *Deep Sea Research Part I: Oceanographic Research Papers*, *66*, 26–40. https://doi.org/10.1016/J.DSR.2012.03.011

Risi, C., Bony, S., Vimeux, F., & Jouzel, J. (2010). Water-stable isotopes in the LMDZ4 general circulation model: Model evaluation for present-day and past climates and applications to climatic interpretations of tropical isotopic records. *Journal of Geophysical Research: Atmospheres*, *115*(D12), 12118. https://doi.org/10.1029/2009JD013255

Risi, C., Bony, S., Vimeux, F., Chongd, M., & Descroixe, L. (2010). Evolution of the stable water isotopic composition of the rain sampled along Sahelian squall lines. *Quarterly Journal of the Royal Meteorological Society*, *136*(S1), 227–242. https://doi.org/10.1002/QJ.485

Risi, C., Noone, D., Worden, J., Frankenberg, C., Stiller, G., Kiefer, M., Funke, B., Walker, K., Bernath, P., Schneider, M., Bony, S., Lee, J., Brown, D., & Sturm, C. (2012). Process-evaluation of tropospheric humidity simulated by general circulation models using water vapor isotopic observations: 2. Using isotopic diagnostics to understand the mid and upper tropospheric moist bias in the tropics and subtropics. *Journal of Geophysical Research: Atmospheres*, *117*(D5), 5304. https://doi.org/10.1029/2011JD016623

Risi, C., Ogée, J., Bony, S., Bariac, T., Raz-Yaseef, N., Wingate, L., Welker, J., Knohl, A., Kurz-Besson, C., Leclerc, M., Zhang, G., Buchmann, N., Santrucek, J., Hronkova, M., David, T., Peylin, P., & Guglielmo, F. (2016). The water isotopic version of the land-surface model ORCHIDEE: implementation, evaluation, sensitivity to hydrological parameters. *Hydrol Current Res*, *7*, 4. https://doi.org/10.4172/2157-7587.1000258

Roche, D., Paillard, D., Ganopolski, A., & Hoffmann, G. (2004). Oceanic oxygen-18 at the present day and LGM: equilibrium simulations with a coupled climate model of intermediate complexity. *Earth and Planetary Science Letters*, *218*(3–4), 317–330. https://doi.org/10.1016/S0012-821X(03)00700-3

Roether, W., Muennich, K. O., & Schoch, H. (2006). On the C-14 to tritium relationship in the North Atlantic Ocean. In *Radiocarbon* (Vol. 22, Issue 3, pp. 636–646). https://doi.org/10.2458/azu_js_rc.22.653

Sachse, D., Billault, I., Bowen, G. J., Chikaraishi, Y., Dawson, T. E., Feakins, S. J., Freeman, K. H., Magill, C. R., McInerney, F. A., van der Meer, M. T. J., Polissar, P., Robins, R. J., Sachs, J. P., Schmidt, H.-L., Sessions, A. L., White, J. W. C., West, J. B., Kahmen, A., Sachse, D., … Kahmen, A. (2012). Molecular Paleohydrology: Interpreting the Hydrogen-Isotopic Composition of Lipid Biomarkers from Photosynthesizing Organisms. *AREPS*, *40*(1), 221–249. https://doi.org/10.1146/ANNUREV-EARTH-042711-105535

Schmidt, G. A. (1998). Oxygen-18 variations in a global ocean model. *Geophysical Research Letters*, *25*(8), 1201–1204. https://doi.org/10.1029/98GL50866

Schmidt, G. A. (1999). Forward modeling of carbonate proxy data from planktonic foraminifera using oxygen isotope tracers in a global ocean model. *Paleoceanography*, *14*(4), 482–497. https://doi.org/10.1029/1999PA900025

Schroeder, K., Ribotti, a., Borghini, M., Sorgente, R., Perilli, a., & Gasparini, G. P. (2008). An extensive western Mediterranean deep water renewal between 2004 and 2006. *Geophysical Research Letters*, *35*, 1–7. https://doi.org/10.1029/2008GL035146

Vadsaria, T., Li, L., Ramstein, G., & Dutay, J. C. (2020). Development of a sequential tool, LMDZ-NEMO-med-V1, to conduct global-to-regional past climate simulation for the Mediterranean basin: an Early Holocene case study. *Geoscientific Model Development*, *13*(5), 2337–2354. https://doi.org/10.5194/GMD-13-2337-2020

van Breukelen, M. R., Vonhof, H. B., Hellstrom, J. C., Wester, W. C. G., & Kroon, D. (2008). Fossil dripwater in stalagmites reveals Holocene temperature and rainfall variation in Amazonia. *Earth and Planetary Science Letters*, *275*(1–2), 54–60. https://doi.org/10.1016/J.EPSL.2008.07.060

Vergnaud Grazzini, C. , G. C. , P. C. , P. C. , and U. M. J. (1986). Foraminifères planctoniques de Méditerranée en fin d'été. Relations avec les structures hydrologiques,. *Mem. Soc. Geol. Ital*, *36*, 175–188.

Voelker, A. H. L., Colman, A., Olack, G., Waniek, J. J., & Hodell, D. (2015). Oxygen and hydrogen isotope signatures of Northeast Atlantic water masses. *Deep Sea Research Part II: Topical Studies in Oceanography*, *116*, 89–106. https://doi.org/10.1016/J.DSR2.2014.11.006

Voelker, A.H., 2023. Seawater oxygen and hydrogen stable isotope data from the upper water column in the North Atlantic Ocean (unpublished data). Interdisciplinary Earth Data Alliance (IEDA), doi: https://doi.org/10.26022/IEDA/112743

Voelker, A.H.L., Colman, A., Olack, G., Waniek, J.J., Hodell, D., 2015. Oxygen and hydrogen isotope signatures of Northeast Atlantic water masses. Deep Sea Research Part II: Topical Studies in Oceanography 116, 89-106, doi: 1016/j.dsr2.2014.11.006.

Voelker, Antje H L (2017): Seawater oxygen isotopes for Station POS334-73, Alboran Sea. Instituto Portugues do Mar e da Atmosfera: Lisboa, Portugal, PANGAEA, https://doi.org/10.1594/PANGAEA.878063

Voelker, Antje H L; Colman, Albert Smith; Olack, Gerard; Waniek, Joanna J; Hodell, David A (2015): Oxygen and hydrogen isotopes measured on water bottle samples during EUROFLEETS cruise Iberia-Forams. PANGAEA, https://doi.org/10.1594/PANGAEA.831462

Vörösmarty, C. J., Fekete, B. M., & Tucker, B. A. (1996). Global River Discharge Database (RivDIS V1.0), International Hydrological Program. *Global Hydrological Archive and Analysis Systems, UNESCO, Paris*.

---

## Editor Decision (ED1)

**Oxygen-18**

b) Evaporation Oxygen-18 (mg/m2/s)

"bites"  "wings"

---

## Author Response (AR2)

**Manuscript " Modelling the water isotopes distribution in the Mediterranean Sea using a high-resolution oceanic model (NEMO-MED12-watiso-v1.0): Evaluation of model results against in-situ observations"**

Mohamed Ayache[1], Jean-Claude Dutay[1], Anne Mouchet[2], Kazuyo Tachikawa[3], Camille Risi[4], and Gilles Ramstein[1]

[1]Laboratoire des Sciences du Climat et de l'Environnement, CEA-CNRS-Université Paris Saclay, 91191, Gif-sur-Yvette, France
[2]Freshwater and OCeanic science Unit of reSearch (FOCUS), Université de Liège, B-4000 Liège
[3]Aix Marseille Univ, CNRS, IRD, INRAE, Coll France, CEREGE, 13545, Aix-en-Provence, France
[4]Laboratoire de Météorologie Dynamique, IPSL, CNRS, Sorbonne Université, Paris, France

Correspondence: Mohamed Ayache (mohamed.ayache@lsce.ipsl.fr)

**Reply to editor comments**

Dear Mr. Yool,

We would like to express our gratitude for the time you have dedicated to reading and evaluating our paper at each stage since its submission. Your comments have significantly enhanced the quality of our paper. We have incorporated your comments into our revisions, as detailed below.

**Color code**
Editor comments
Authors response
The modifications performed in the manuscript appear in red in the Revised Manuscript with Changes Marked.

In your response to the first comment (actually a set of questions) from referee #2, you provide an extensive answer to these points. However, the resulting text modifications are quite short in contrast – for instance, you mention LDMZ several times in your response, but this does not appear to occur in your modified text. If I have overlooked this, please let me know, but if relevant information has not been included in the manuscript can you please address this.

We would like to thank you for your comment. We have clarified and better presented our experimental protocol in the revised version of the paper as requested by reviewer #2. Section 2 has been largely rewritten in the revised version of the paper.

We have incorporated the following sentences on the dynamic forging (see lines 150-165 in the track changes version):

The simulation was conducted over 30 years following a 44-year spin-up period (1958–1980 repeated twice), ensuring model stability for over 75 years. The years of hydrodynamic forcing were randomly selected from precalculated circulation fields spanning 1958 to 2013 (Beuvier et al., 2012a). The objective of this method is to minimize the impact of extreme variability effects, such as the Eastern Mediterranean Transient (EMT) or the Western Mediterranean Transition (WMT), on the simulated circulation (Roether et al., 2006; Schroeder et al., 2008). The spin-up

strategy was adapted from previous passive tracer simulations, such as neodymium and tritium studies (Ayache et al., 2015a, 2016).

and these sentences, on atmospheric forcing (see lines 170-175 in the track changes version).

The aim is to assess the model's performance in the present climate and against in-situ data observed between 1982 and 2022. Therefore, we have opted to use the climatological mean of the LMDZ-iso 1990-2020 simulation as boundary conditions. This choice was made to minimize the warming trend during this period and to ensure that the precipitation and evaporation simulated by the LMDZ-iso model for the current climate situation are as close to the average state as possible, with minimal impact from inter-annual variability.

On the same point, there are a few minor issues with the added text:
- "a random draw" – what is meant by this? It's not clear to me at all

The term 'random draw' refers to the process of selecting years randomly from precalculated circulation fields of the Mediterranean Sea spanning 1958 to 2013 (Beuvier et al., 2012a). This approach is employed to simulate a consistent and random representation of present-day circulation. By incorporating these random selections into the historical period (1958–2013), the goal is to minimize the impact of intense variability events like the EMT or the WMT on the circulation simulations.

We have clarified in the revised manuscript, as shown below (see lines 150-165 in the track changes version):

The simulation was conducted over 30 years following a 44-year spin-up period (1958–1980 repeated twice), ensuring model stability for over 75 years. The years of hydrodynamic forcing were randomly selected from precalculated circulation fields spanning 1958 to 2013 (Beuvier et al., 2012a). The objective of this method is to minimize the impact of extreme variability effects, such as the Eastern Mediterranean Transient (EMT) or the Western Mediterranean Transition (WMT), on the simulated circulation (Roether et al., 2006; Schroeder et al., 2008). The spin-up strategy was adapted from previous passive tracer simulations, such as neodymium and tritium studies (Ayache et al., 2015a, 2016).

- "(for more than 75 of run)" – simulation years one assumes?

Corrected

- "The spin-up strategy was adapted FROM our previous passive tracer simulations"

Changed

In your response to point #4 from referee #3, you again provide an extensive and detailed response to the issues raised, but again it is unclear if or how the manuscript has been revised to account for this. Please make this clearer.

Agreed,

These sentences was added to the revised manuscript

(see lines 127-131 in the track changes version).

The isotopic composition is determined on post-processing because here we transport the isotopic ratio (see equation 1), which allows us to carry a single tracer "$^{18}R$" instead of two tracers "18O and 16O". This reduces the computation time on the machine, which is a crucial factor in the performance of the model, especially in a very long palaeo-simulation. It is a common practice

to transport the isotopic ratio rather than the individual species. For example, radiocarbon distribution (14C/C) in the Mediterranean Sea (Ayache et al., 2017) and $^{18}O/^{16}O$ of precipitation (Risi et al., 2010b).

and these sentences (see lines 209-214 in the track changes version).

In our study, we utilized the off-line uncoupled mode of NEMO, which employs pre-calculated dynamics. This mode operates with a fixed volume and explicit fluxes of evaporation, precipitation, and runoff. Alternatively, the online coupled mode of NEMO can be employed to compute dynamic variables (such as circulation fields U, V, and W) in real time. The sea surface elevation and model layer thicknesses are adjusted by the freshwater flux (E-P-R), consequently affecting the model volume. It is essential to ensure that total volume variations accurately correspond to the E-P forcing used to drive the isotopic module, thus maintaining the perfect conservation of tracer content.

Your response to point #7 from referee #3 on the value of pCO2 used is both ambiguous and unclear on whether it has led to any manuscript amendment for clarification. I appreciate that the 348 ppm value is that from around year 1982, but go on to confuse things by referring to comparison time points up to year 2022. Can you please make clear – and do so in your manuscript – why the value was chosen, and whether it's constant regardless of simulation time, which it what's implied in your response.

We fully agree with this remark. The LMDZ-iso Atmospheric simulation was conducted following the Atmospheric Model Intercomparison Project (AMIP) protocol, as presented in Risi et al. (2010b), utilizing prescribed monthly and interannually varying SST and sea ice, in addition to a constant CO2 value of 348 ppm for the present-day situation. The same values have been employed in this study because their impact is constrained on our results, given that the model has been evaluated against in-situ data sampled primarily in the 1980s (i.e. Pierre et al. 1999). While in-situ data from more recent periods 1998-2022 from Reverdin et al. (2022) are more limited and only localized in the northwestern Mediterranean Sea.

This point is clarified in the revise manuscript.

(see lines 166-170 in the track changes version)

The LMDZ-iso Atmospheric simulation was conducted following the Atmospheric Model Intercomparison Project (AMIP) protocol, as presented in Risi et al. (2010b), utilising prescribed monthly and interannually varying SST and sea ice, in addition to a constant CO2 value of 348 ppm for the present-day situation. The impact of these low pCO2 values in comparison to the current value of 421 ppm is constrained by the fact that the model has been evaluated against in-situ data sampled primarily in the 1980s.

Figure 1 uses a strange plotting style with large circular patches. As these patches overlap, it tends to have the result that the order in which the data is plotted matters, such that southernly data overlies more northerly data given a strange tessellating pattern. Is there any way that the panels could be replotted in a style that doesn't involve overlapping like this? It doesn't appear to be a problem in later figures.

Thank you for pointing this out. We've looked at the problem of judging between boxes in the current version of Figure 1 and found that the problem is due to saturation in the scale used to plot this figure. We've adjusted both the scale and the colour palette used (see the new version of Figure 1). We hope that these changes will resolve the problem that appeared in the previous version.

[Figure]

*Figure 1 Boundary conditions and input (evaporation and precipitation) maps applied to NEMO that originate from the LMDZ-iso atmospheric model (Risi et al., 2010a). a) Evaporation, b) Precipitation, {c) River runoff, J) Net surface flux (E - P - R) for $H_2O$, (b, e, h, k) the same but for $\delta^{18}O_w$, (c, f, i, l) for $\delta D_w$. The isotopic composition of river runoff is not available from the LMDZ-iso model: this flux is computed as $^{18}RP \times R$ where R is prepared from the data of Ludwig et al., (2009) and Vorosmartyet al., (1996) and $^{18}RP$ is the isotopic ratio in precipitations at the same time and location*

 Finally, regarding the use of colour in the figures, this was noted by both the referees and by our production team. The change between revisions seems to have been to move from one rainbow palette to a more garish one, and the revised colour map is arguably inferior to that at the previous revision as it "wraps-around" such that the final colour is very similar to the first colour. Please consider checking the colour scales of your figures using our recommended colour blindness simulator …

 https://www.color-blindness.com/coblis-color-blindness-simulator/

 You may also wish to consult the ColorBrewer website for colour maps that are appropriate for colour blindness …

 https://colorbrewer2.org/#type=sequential&scheme=BuGn&n=3

 Please advise me if you have any technical challenges on this point. However, from the changes between manuscript revisions, it appears that the software you use is flexible on this point.

 On more specific details, I note that Figure 1 has transitioned from a colourblind-friendly palette, Viridis / Parula, to one that makes use of red-green, while Figure 6 appears to show the old colour bar for a figure using the new colour scale.

We appreciate the feedback regarding the use of color in the figures. We understand the importance of ensuring that our visualizations are accessible to all readers. In response to your suggestions, we have replaced the colour palette with a new one, as shown in the revised version of our manuscript. We hope that this new palette will offer improved clarity. Furthermore, we have rectified the inconsistencies noted in Figures 1 and 6 to maintain coherence throughout the manuscript.

---

## Author Response (AR3)

**Manuscript " Modelling the water isotopes distribution in the Mediterranean Sea using a high-resolution oceanic model (NEMO-MED12-watiso-v1.0): Evaluation of model results against in-situ observations"**

Mohamed Ayache[1], Jean-Claude Dutay[1], Anne Mouchet[2], Kazuyo Tachikawa[3], Camille Risi[4], and Gilles Ramstein[1]

[1]Laboratoire des Sciences du Climat et de l'Environnement, CEA-CNRS-Université Paris Saclay, 91191, Gif-sur-Yvette, France
[2]Freshwater and OCeanic science Unit of reSearch (FOCUS), Université de Liège, B-4000 Liège
[3]Aix Marseille Univ, CNRS, IRD, INRAE, Coll France, CEREGE, 13545, Aix-en-Provence, France
[4]Laboratoire de Météorologie Dynamique, IPSL, CNRS, Sorbonne Université, Paris, France Correspondence: Mohamed Ayache (mohamed.ayache@lsce.ipsl.fr)

Dear Mr. Yool,

Thank you once again for your effort and advice on revising our paper. We apologize for the delay caused by the projection of Figure 1.

In the previous version, panels (a)–(f) were plotted directly from the output of the LMDZiso atmospheric model, whereas panels (j)–(l) were interpolated from the NEMO-MED12 grid (net surface flux calculated during simulation). In the new version, we have applied the same projection used in panels (j)–(l) to the entire Figure 1. We hope this has corrected the issues found in the previous version (see the new Fig.1 in the revised version, and track-changes file).

Regarding the color palette, we have found a new palette (batlow) recommended by Scientific Colour Maps (https://s-ink.org/scientific-colour-maps). Please see (below) a comparison between the mpl_Div_Spectral and batlow palette colors for Figure 1 and Figure 2.

We prefer using the mpl_Div_Spectral palette because the gradients are clearer compared to the batlow palette. However, if the production team prefers the batlow palette, we can easily change it for the published version.

We hope these changes resolve the problems that appeared in the previous version.

Best regards,

Mohamed Ayache

[Figure]

**Figure 1 (used color palette 'mpl_Div_Spectral' )** Boundary conditions and input (evaporation and precipitation) maps applied to NEMO that originate from the LMDZ-iso atmo spheric model (Risi et al., 2010b).a) Evaporation, b) Precipitation, c) River runoff, J) Net surface flux (E- P- R) for H2O, (b, e, h, k) the same but for $\delta^{18}O_w$, (c, f, i, l) for $\delta D_w$. The isotopic composition of river runoff is not available from the LMDZ-iso model: this flux is computed as $^{18}RP \times R$ where R is prepared from the data ofLudwig et al. (2009) and Vörösmarty et al. (1996) and 18RP is the isotopic ratio in precipitations at the same time and location

[Figure]

**Figure 1 (color palette 'batlow')** Boundary conditions and input (evaporation and precipitation) maps applied to NEMO that originate from the LMDZ-iso atmo spheric model (Risi et al., 2010b).a) Evaporation, b) Precipitation, c) River runoff, J) Net surface flux (E- P- R) for H2O, (b, e, h, k) the same but for δ18Ow, (c, f, i, l) for δDw. The isotopic composition of river runoff is not available from the LMDZ-iso model: this flux is computed as 18RP × R where R is prepared from the data ofLudwig et al. (2009) and Vörösmarty et al. (1996) and 18RP is the isotopic ratio in precipitations at the same time and location

[Figure]

**Figure 2.** (**used color palette 'mpl_Div_Spectral'**) **The model outputs against in-situ data for the present-day situation. a) δ18Ow (in ‰) distribution in the surface water (50 m depth). b) E-W vertical section of δ18Ow (in ‰) in the western Mediterranean basin d) Zonal mean comparison of δ18Ow (in ‰) average vertical profiles in the western basin presenting model results against in-situ data. c) and e) the same as b) and d) but for the eastern basin. Colour-filled dots represent in-situ observations from (Epstein and Mayeda, 1953; Stahl and Rinow, 1973; Pierre et al., 1986; Gat et al., 1996; Pierre, 1999; Voelker, 2017; Reverdin et al., 2022). Both model and in-situ data use the same color scale.**

[Figure]

**Figure 2.** **(color palette 'batlow')** The model outputs against in-situ data for the present-day situation. a) δ18Ow (in ‰) distribution in the surface water (50 m depth). b) E-W vertical section of δ18Ow (in ‰) in the western Mediterranean basin d) Zonal mean comparison of δ18Ow (in ‰) average vertical profiles in the western basin presenting model results against in-situ data. c) and e) the same as b) and d) but for the eastern basin. Colour-filled dots represent in-situ observations from (Epstein and Mayeda, 1953; Stahl and Rinow, 1973; Pierre et al., 1986; Gat et al., 1996; Pierre, 1999; Voelker, 2017; Reverdin et al., 2022). Both model and in-situ data use the same color scale.